# Selective inhibition of mitochondrial Kv1.3 prevents and alleviates multiple sclerosis in vivo

Beatrice Angi[1,9], Tatiana Varanita[1,9], Marco Puthenparampil[2], Valentina Scattolini[1], Michael Donadon[1], Mitra Tavakoli[1], Marta Favero[1], Maguie El Boustani[1], Matthias Soddemann[3], Lucia Biasutto[4], Diletta Arcidiacono[5], Alberto Ongaro[6], Andrea Mattarei[6], Livio Trentin[7], Gregory Wilson[8], Erich Gulbins[3,8✉], Paolo Gallo[2] & Ildiko Szabo[1✉]

## Abstract

Multiple sclerosis (MS) is characterized by invasion of the brain by effector memory T ($T_{EM}$) lymphocytes that have been activated by repeated auto-antigen stimulation. Existing therapies target these and other autoreactive lymphocytes but their side effects include general immunosuppression and toxicity. Because the Kv1.3 potassium channel is highly expressed by chronically activated autoreactive $T_{EM}$s, we investigated whether specific targeting of mitochondrial Kv1.3 using the pharmacological inhibitor PAPTP could selectively kill these $T_{EM}$s in patients and mice with MS. 1 µM PAPTP targeted and reduced the number of autoreactive $T_{EM}$s in blood samples from relapsing-remitting MS (RRMS) patients, leaving other T cell populations unaffected. Remarkably, pre-treatment of the entire T cell population with PAPTP during adoptive transfer of experimental autoimmune encephalomyelitis (EAE) killed $T_{EM}$s and completely prevented disease onset in this mouse model. Moreover, PAPTP selectively eliminated activated $T_{EM}$s and halted EAE progression when administered following disease onset. Our findings reveal the potential of PAPTP as an effective treatment for MS without adverse side effects.

Keywords Mitochondrial Kv1.3 Channel Inhibition; Multiple Sclerosis; Experimental Autoimmune Encephalomyelitis; Effector Memory T Cell
Subject Category Immunology

See also: S Pluchino and CM Willis

## Introduction

Multiple sclerosis (MS) is a disabling autoimmune disorder affecting more than 2.5 million people worldwide, in which inflammatory cells migrate from peripheral blood vessels to the central nervous system (CNS), where they cause demyelination and axonal degeneration (Klotz et al, 2023). Several lines of evidence support the role of myelin-specific autoreactive T cells in early disease progression (Attfield et al, 2022; Bronge et al, 2019; Cruciani et al, 2021). In particular, inflammatory responses mediated by these T cells against myelin proteins including myelin basic protein (MBP), myelin oligodendrocyte glycoprotein (MOG) and proteolipid protein (PLP) oligodendrocytes are crucial for MS pathogenesis. These cluster of differentiation (CD) 4-expressing autoreactive T cells interact with antigen-presenting cells and B cells in peripheral lymphoid organs, but upon activation, they become able to cross the blood–brain barrier. After reaching the CNS, their subsequent reactivation triggers the release of effector cytokines, leading to attraction and activation of microglia and macrophages, followed by neuronal demyelination and axonal injury (Bittner and Meuth, 2013; Sospedra and Martin, 2016). Antibody production by plasma cells and attack by cytotoxic T cells further contributes to this neurodegeneration (Attfield et al, 2022).

Inhibition of MOG-specific autoreactive $CD4^+$ T cells was shown to exert a strong protective effect in experimental autoimmune encephalomyelitis (EAE) animal model of multiple sclerosis (Kohm et al, 2002), however, selective targeting of the autoreactive $CD4^+$ T cell population while sparing other T cells in patients with MS remains a challenging task. Individuals expressing the HLA-DR15 gene product show a proportionate increase in autoreactivity of peripheral $CD4^+$ T helper ($T_H$) lymphocytes (Jelcic et al, 2018; Mohme et al, 2013), in agreement with the finding that HLA-DR15 haplotype confers significant genetic susceptibility to the development of MS (Attfield et al, 2022). These autoreactive $T_H$ cells, which are characterized by high proliferative propensity were defined as autoproliferative T cells (Jelcic et al, 2018; Mohme et al, 2013) and show a highly activated, T effector memory ($T_{EM}$) phenotype with characteristics compatible with pro-inflammatory T cells. These autoreactive/autoproliferative cells are likely activated by self-peptides presented by HLA class II molecules in MS patients (Wang et al, 2020).

[1]Department of Biology, University of Padova, Padova, Italy. [2]Department of Neurology, University Hospital of Padova, Padova, Italy. [3]Department of Molecular Biology, University Hospital Essen, Essen, Germany. [4]CNR Institute of Neurosciences, Padova, Italy. [5]Veneto Institute of Oncology IOV-IRCCS, Padova, Italy. [6]Department of Pharmacological Sciences, University of Padova, Padova, Italy. [7]Hematology Unit, University Hospital of Padova, Padova, Italy. [8]Department of Surgery, University of Cincinnati College of Medicine, Cincinnati, OH, USA. [9]These authors contributed equally: Beatrice Angi, Tatiana Varanita. ✉E-mail: erich.gulbins@uni-due.de; ildiko.szabo@unipd.it

Natalizumab (NAT) is a humanized monoclonal antibody that is used to treat MS (Rinaldi et al, 2012). It recognizes the α4 chain of the VLA4 integrin expressed on lymphocytes. By blocking the interaction between VLA4 and its endothelially-expressed ligand VCAM-1, NAT prevents migration of lymphocytes across the blood–brain barrier and into the brain parenchyma (Nielsen et al, 2017). However, patients treated with NAT display significantly higher lymphocyte autoproliferation than naive patients (Cruciani et al, 2021) indicating that the treatment merely traps pathogenic cells in the periphery rather than eliminating them. Furthermore, chronic NAT treatment is associated with sustained immunosuppression as well as toxicity (Edan and Le Page, 2023; Singer, 2017), specifically the development of progressive multifocal leukoencephalopathy (PML) leading to severe disability or death. New approaches are therefore needed for the treatment of MS. Because chronically activated $T_{EM}$s (CD45RO$^+$CCR7$^-$) are the major infiltrating cell type in MS brains (Rus et al, 2005), their selective removal would be a valuable strategy, as it could lead to their definitive elimination from the circulation of patients undergoing NAT treatment.

Given their essential role in immune cell function, ion channels represent attractive targets for autoimmune diseases (Manolios et al, 2023). In particular, the plasma membrane (PM) Kv1.3 potassium channel, which is highly expressed by chronically activated $T_{EM}$s, is crucial for T cell proliferation due to its role in maintaining the negative membrane potential required for $Ca^{2+}$ influx during cell activation (Cahalan and Chandy, 2009). Indeed, its inhibition causes persistent and specific suppression of $T_{EM}$ proliferation (Beeton et al, 2001b; Wulff et al, 2003). High levels of Kv1.3 expression are found in autoreactive and myelin-specific T cells from patients with MS (Beeton et al, 2006; Rus et al, 2005), but not in T cells from healthy controls. In addition, selective peptide toxin-based PM Kv1.3 blockers and a global Kv1.3 knockout have been shown to alleviate EAE symptoms in rodents (Gocke et al, 2012). An advantage of Kv1.3 inhibition is its independence of antigen specificity. However, although PM Kv1.3 blockers decrease proliferation (Varga et al, 2021), they do not kill pathogenic T cells (Lam and Wulff, 2011), in accordance with observations from our and other groups showing that PM Kv1.3 blockers such as *Stichodactyla* toxin ShK and Margatoxin do not trigger programmed cell death (apoptosis) of healthy or pathologic cells (e.g., (Leanza et al, 2012; Szabo et al, 2008)).

We and others have revealed the presence of functional Kv1.3 in the inner mitochondrial membrane (mitoKv1.3) and its positive correlation with Kv1.3 expression in the PM in various cell types, including human and mouse T lymphocytes (Capera et al, 2022; Szabo et al, 2008; Szabo et al, 2005). We have recently developed PAPTP (for structure see Fig. EV2A), a membrane-permeable derivative of the Kv1.3 inhibitor PAP-1, that specifically targets this mitochondrial form of Kv1.3 (Leanza et al, 2017). In cells that express elevated levels of Kv1.3, PAPTP induces apoptosis by blocking mitoKv1.3 with consequent hyperpolarization of the inner mitochondrial membrane and subsequent increase of mitochondrial ROS production and cytochrome c release (Severin et al, 2022), presenting a potential means to eliminate chronically activated $T_{EM}$s. The effect of PAPTP is specific for Kv1.3 and depends on the expression of the channel (e.g., (Leanza et al, 2017; Prosdocimi et al, 2024). PAPTP is stable in the blood, as it was quantitatively recovered unaltered after 4 h upon incubation in fresh mouse blood at 37 °C (Leanza et al, 2017) and was able to selectively trigger cytochrome c release and apoptosis in pathologic

Kv1.3$^{high}$ B lymphocytes in vivo upon intraperitoneal injection, leading to a significant amelioration of chronic B cell lymphocytic leukemia in a genetic model (Severin et al, 2022). Here we present evidence that PAPTP selectively eliminates autoreactive/autoproliferative $T_{EM}$s in the peripheral blood of MS patients and EAE mice. Remarkably, this treatment completely prevents disease onset and significantly reduces symptoms, including demyelination, neuronal degeneration, and motor deficit in murine EAE. These findings suggest that inhibition of mitoKv1.3 by PAPTP is a promising strategy to reduce the number of autoreactive lymphocytes, and thus prevents disease onset and progression, in MS.

# Results

## PAPTP induces apoptosis in $T_{EM}$s from MS patients

For PAPTP to induce apoptosis, cells must have enhanced basal ROS levels as well as high expression of Kv1.3 (Leanza et al, 2017; Severin et al, 2022). Although Kv1.3 is known to be highly expressed in $T_{EM}$s from MS patients (Beeton et al, 2006), and mitochondrial metabolism is altered in autoreactive T cells (De Biasi et al, 2019), the levels of ROS in these cells are unknown. We therefore measured mitochondrial ROS production in peripheral blood mononuclear cells (PBMCs) isolated from RRMS patients and undergoing treatment with NAT, as these patients display higher lymphocyte autoproliferation than those who did not receive NAT treatment. Mitochondrial ROS production was measured using MitoSox following seven days of T cell autoproliferation in vitro, carried out according to the protocol described by Jelcic and colleagues (Jelcic et al, 2018) using cells from patients undergoing NAT treatment. The Interleukin-2 receptor alpha chain also called CD25 was exploited as a highly expressed marker of canonically activated T lymphocytes (Adamczyk et al, 2023; Peng et al, 2023; Szabo et al, 2019). Activation of naive T cells (through the antigen-specific T cell receptor) drives differentiation into long-lived circulating central ($T_{CM}$s) and effector memory ($T_{EM}$s) T cells. Upon antigenic restimulation $T_{CM}$ lose the chemokine receptor CCR7 expression and differentiate into $T_{EM}$ and finally into $T_{EMRA}$, which are considered to be terminally differentiated. $T_{EMRA}$ (defined as terminally differentiated effector memory cells re-expressing CD45RA) lack both CCR7 and CD45RO, a short isoform of CD45, while re-express the longer isoform, CD45RA (Carrasco et al, 2006). Similarly to $T_{EM}$s, the number of CD4$^+$ $T_{EMRA}$s is also increased in MS patients compared to healthy controls (Hawke et al, 2020). $T_{EM}$s and $T_{EMRA}$s do not express the chemokine receptor CCR7 in contrast to $T_{CM}$s and naive $T_H$ cells. CD4$^+$ CD25$^+$CCR7$^-$ effector T cells ($T_{EM}$s and $T_{EMRA}$) exhibited 3-fold higher levels of basal mitochondrial ROS production than CD4$^+$ CD25$^+$CCR7$^+$ $T_{CM}$s and naive $T_H$ cells in all examined patients (Fig. 1A). This difference in ROS production was observed also following 30-min treatment with 1 μM PAPTP (Figs. 1A and EV1A), resulting in consistent ROS increase in CD4$^+$CCR7$^-$ cells compared to CD4$^+$CCR7$^+$ cells already at 1 μM drug concentration (Leanza et al, 2013) (see Fig. EV1N for gating strategy). We also observed significantly greater (2-fold higher) expression of Kv1.3 in activated (CD25$^+$) versus non-activated (CD25$^-$) $T_H$ cells (Figs. 1B and EV1B), as expected (Beeton et al, 2003). Total Kv1.3 expression was evaluated in whole cells using the

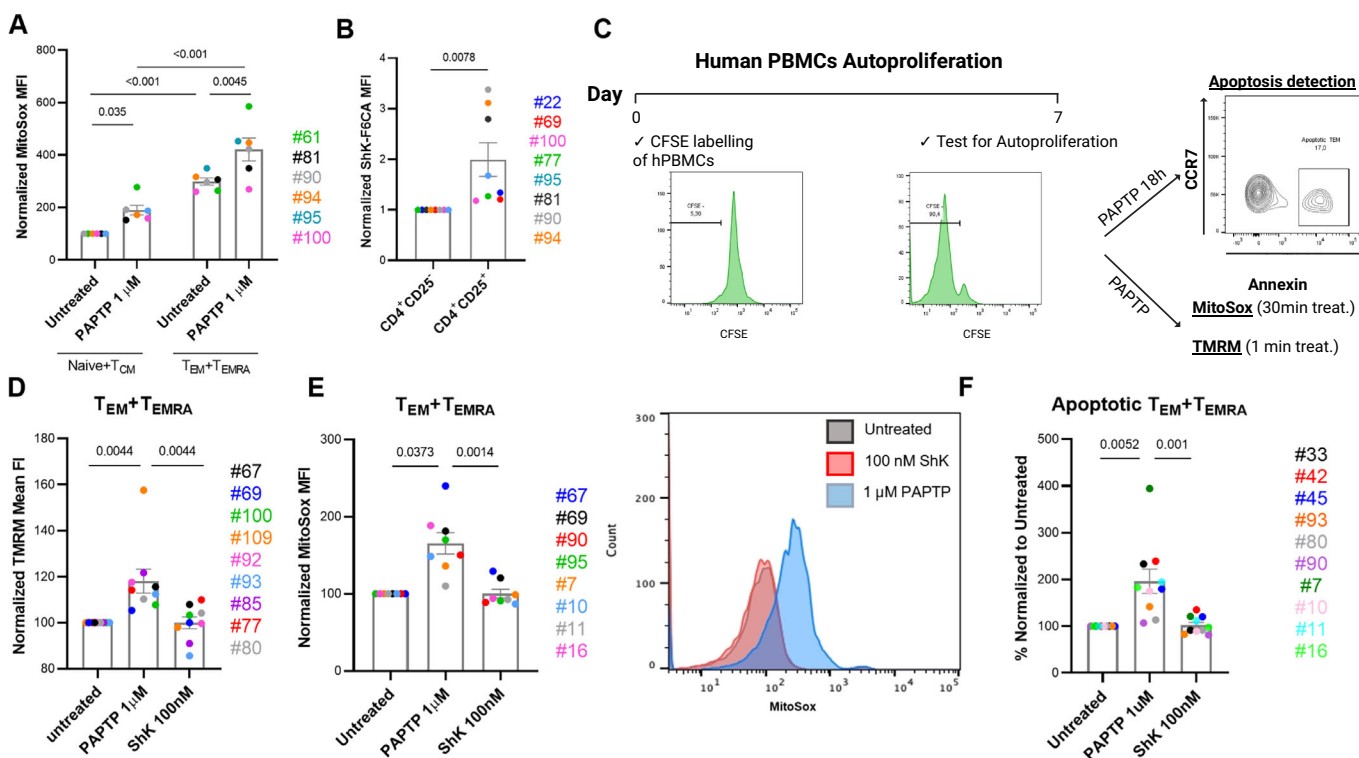

**Figure 1. Analysis of the effects of PAPTP on PBMCs from MS patients.**

(A) Quantitative results of the MitoSox Mean Fluorescence Intensity of CD4$^+$ CD25$^+$ CCR7$^+$ (Naive + T$_{CM}$) and CD4$^+$ CD25$^+$ CCR7$^-$ (T$_{EM}$ + T$_{EMRA}$) untreated hPBMCs or treated with 1 μM PAPTP for 30 min (cultured for 7 days in both cases). Data were normalized on the mean Fluorescence Intensity of untreated CD4$^+$ CD25$^+$ CCR7$^+$ (Naive + TCM) cells (n = 6 patients). P-values obtained from two-way ANOVA test are shown. (B) Mean Fluorescence Intensity of ShK-F6CA (fluorescent Kv1.3 inhibitor ShK) in CD4$^+$CD25$^-$ and CD4$^+$CD25$^+$ T cells from proliferative hPBMCs after 7 days of autoproliferation in vitro (n = 8 patients). For each patient, the mean Fluorescence intensity of CD4$^+$CD25$^+$ T cells was normalized on the mean fluorescence intensity of CD4$^+$CD25$^-$ T cells. Shown are p-values of Wilcoxon test. (C) Strategy for analyzing apoptosis in autoproliferative lymphocytes: hPBMCs were labeled with CFSE and incubated for 7 days at 37 °C. Following the incubation period, proliferation was assessed as mean of CFSE dilution. Subsequently, cells were subjected to treatment with PAPTP, or left untreated for different timepoints based on the experiment (18 h for Annexin, 30 min for MitoSox, and 1 min for TMRM). Apoptosis, mitochondrial ROS production and mitochondrial membrane hyperpolarization were evaluated via flow cytometry analysis. (D) Quantitative results of the TMRM Mean Fluorescence Intensity of CD4$^+$ CCR7$^-$ (T$_{EM}$ + T$_{EMRA}$) untreated hPBMCs or treated with 1 μM PAPTP or 100 nM ShK for 60 s (cultured for 7 days in both cases). Data were normalized on the Mean Fluorescence Intensity of untreated CD4$^+$ CCR7$^-$ (T$_{EM}$ + T$_{EMRA}$) cells (n = 9 patients). (E) Quantitative results of the MitoSox Mean Fluorescence Intensity of CD4$^+$ CD25$^+$ CCR7$^-$ (T$_{EM}$ + T$_{EMRA}$) untreated hPBMCs or treated with 1 μM PAPTP or 100 nM ShK for 30 min (cultured for 7 days in both cases). Data were normalized on the Mean Fluorescence Intensity of untreated CD4$^+$ CCR7$^-$ (T$_{EM}$ + T$_{EMRA}$) cells (n = 8 patients). Right: Representative quantitative results of the Mean Fluorescence Intensity of MitoSox in CD4$^+$ CD25$^+$ CCR7$^-$ (T$_{EM}$ + T$_{EMRA}$) lymphocytes either treated with 1 μM PAPTP or 100 nM ShK. (F) Normalized apoptotic levels of proliferative CD4$^+$CD25$^+$CCR7$^-$ T$_{EM}$ + T$_{EMRA}$ cells at 1 μM PAPTP or 100 nM ShK. For each patient, data were normalized based on the untreated sample (n = 10 for each group). (A, B, D–F) Data represent average ± SEM with superimposed individual data points for each patient. Each data point represents hPBMCs derived from a distinct patient. (D–F) p-values of Friedman test are shown. Source data are available online for this figure.

fluorescently labeled ShK toxin which binds to the pore region of the channel with high affinity (Beeton et al, 2003).

Having established that activated T cells from MS patients strongly upregulate Kv1.3 expression and pathological effector T cells are characterized by high basal mitochondrial ROS levels, we sought to determine their susceptibility to PAPTP-induced changes in mitochondrial parameters (changes in membrane potential and ROS release) and to apoptosis using flow cytometry (Fig. 1C). Naive T$_H$ cells, T$_{CM}$s, T$_{EM}$s, and T$_{EMRA}$s were identified from PBMCs isolated from HLA DRB1*15-positive RRMS patients, based on expression of CCR7 and CD45RO (an alternative isoform of CD45) that is expressed on activated memory T cells (see Fig. EV1C for gating strategy). To prove that PAPTP indeed affects mitochondrial membrane potential and ROS release in the autoproliferative human T$_{EM}$s, and T$_{EMRA}$s, as expected (Leanza et al, 2017), we measured a rapid hyperpolarization within 1 min after addition of PAPTP, in

accordance with the block of the influx of depolarizing positively charged K$^+$ ions into the matrix by the drug (Fig. 1D). Importantly, ShK, a membrane-impermeable Kv1.3 toxin inhibitor that is unable to reach the mitochondria, did not have this effect. Consistent with the observations that hyperpolarization triggers mitochondrial ROS release (Murphy, 2009), we only observed ROS production after treatment with PAPTP, but not with ShK (Fig. 1E). In contrast to PAPTP, which induces death through strong oxidative stress (Severin et al, 2022), ShK was not able to trigger apoptosis of these autoproliferative T cells from MS patients (Fig. 1F). Altogether, these data indicate high expression of Kv1.3 in activated autoproliferative T$_H$ cells and the presence of Kv1.3 in their mitochondria, as inhibition of mitoKv1.3 (but not of PM Kv1.3) triggers mitochondrial membrane potential changes and ROS release in lymphocytes.

Next, we dissected the effect of PAPTP on the different T cell subpopulations. We observed a significant decrease in the

percentage of autoproliferative $T_{EM}$s amongst total CD4$^+$CD25$^+$ autoproliferative T cells following treatment with 1 μM PAPTP (Fig. 2A), which was associated with a 2-fold increase of apoptotic $T_{EM}$s population (Fig. 2B). PAPTP also increased apoptosis in $T_{EMRA}$s (Fig. 2D) but did not change their overall percentage (Fig. 2C). When evaluating $T_{EM}$ and $T_{EMRA}$s together (the CCR7$^-$ population), a two-fold increase in Annexin-positive apoptotic cells could be detected (Fig. 2E). Other types of lymphocytes, including naive $T_H$ and $T_{CM}$ cells, were unaffected by 1 μM PAPTP (Fig. 2F–I). While 5 μM PAPTP did not further increase death of $T_{EM}$s and $T_{EMRA}$s, it triggered a slight increase of apoptosis in naive $T_H$ cells and $T_{CM}$s (Fig. EV1D–L). Unlike 1 μM PAPTP, the drug applied at 5 μM concentration triggered apoptosis also in T cells isolated from PBMC of healthy subjects (Fig. EV1M), suggesting an unspecific effect taking place at a concentration that is five times higher than the efficient dose against pathologic cells. Importantly, 1 μM PAPTP did not cause enhanced apoptosis in T cells of healthy subjects.

Finally, we observed that the majority of PBMCs from MS patients comprised CD3$^+$ T cells, and only a small percentage of CD19$^+$ B cells, following autoproliferation in vitro (Fig. 2J). 1 μM PAPTP did not induce significant changes in the relative proportions of immune subpopulations; T cells, B cells, CD4$^+$ $T_H$ cells, and cytotoxic T cells (CD8$^+$) (Fig. 2J). Together, these findings demonstrate that already at low concentration (1 μM) PAPTP induces apoptosis of activated, autoproliferative $T_{EM}$s and $T_{EMRA}$s in PBMCs isolated from HLA DRB1*15-positive RRMS patients without affecting other subpopulations of T and B cells.

## PAPTP prevents adoptive transfer of EAE

Given the selective effect of PAPTP on activated, autoproliferative pathologic T cells in vitro, we sought to investigate whether PAPTP acts on mitochondrial Kv1.3 also in murine T lymphocytes and whether it might prevent disease resulting from adoptive cell transfer of $T_{EM}$s. First, to investigate the subcellular localization of PAPTP in mouse T lymphocytes, a fluorescent analog (PAPTP-NBD; see Fig. EV2A) was synthesized by covalently attaching the fluorophore 7-nitrobenz-2-oxa-1,3-diazole (NBD) to PAPTP (named PAPTP-Fluor). NBD is commonly used in biochemistry and chemical biology due to its strong fluorescence in the visible range, low molecular weight, and lack of ionic functional groups, which minimizes interference with the pharmacodynamic properties of labeled compounds (Jiang et al, 2021). Details regarding the synthesis of PAPTP-NBD (PAPTP-Fluor) are provided in the Materials and methods section and in Fig. EV2A. As observable in Fig. 3A (see also Fig. EV2B for control), the PAPTP-Fluor co-localized with Mitotracker, a lipophilic cationic dye that is taken up by the mitochondria that do not form a typical network around the nucleus in T cells (Baixauli et al, 2011; Buck et al, 2016). We also obtained genetic evidence showing that expression of mitochondrial Kv1.3 is sufficient to trigger PAPTP-induced apoptosis in T lymphocytes: Kv1.3 was silenced using CRISPR-Cas9 in cells obtained from the spleen of MOG$_{33-55}$-immunized mice and the Kv1.3-expressing cells were separated from those silenced/negative for Kv1.3 (downregulation of Kv1.3 in cells treated with CRIPSR/Cas9 and Kv1.3 staining in sorted cells is shown in Fig. EV2C; see also Methods section). Next, the Kv1.3-negative cells were

transfected with mitochondria-targeted EYFP-Kv1.3 construct (Szabo et al, 2008) (see Fig. EV2C) and the cells were either left untreated or treated with 1 μM PAPTP (Fig. 3B). These experiments clearly show that in contrast to mitoKv1.3-negative cells, where PAPTP induced apoptosis only in 2% of the cells, in mitoKv1.3-positive cells cell apoptosis exceeded 75% as assessed by Annexin V staining and flow cytometry (Figs. 3C and EV2C). Altogether, these data further confirm the action of PAPTP on mitochondrial Kv1.3.

EAE can be transferred from one mouse to another using the adoptive transfer model of EAE (EAE-AT), in which T lymphocytes isolated from the spleens of EAE-induced mice are injected into healthy mice (Becker et al, 2017) (Fig. 3D). In rodents, the differential expression of the adhesion molecules CD62L (also called L-selectin) and CD44 (activation marker) are used to distinguish T cell subpopulations (Mahnke et al, 2013). The subpopulation of $T_{EMRA}$s is not distinguished but MOG-specific $T_{EM}$s are sufficient to induce the disease in the adoptive transfer model of EAE (Williams et al, 2011). Encouragingly, as low as 1 μM PAPTP treatment caused a marked reduction in the percentage of $T_{EM}$s in lymphocytes (close to 0%) isolated from the spleens of EAE mice (Fig. 3E), as well as increased overall level of apoptosis amongst the whole population of isolated lymphocytes as assessed by Annexin staining (Fig. 3F). Because neither naive $T_H$ cells nor $T_{CM}$s exhibited signs of apoptosis (Fig. 3G), this increase appeared to be solely due to induction of apoptosis in $T_{EM}$s. Indeed, 1 μM PAPTP selectively increased apoptosis of CD44$^+$CD62L$^-$ $T_{EM}$s by 60% (Fig. 3G).

We subsequently injected PAPTP-treated and untreated lymphocytes from EAE mice into healthy animals and monitored their clinical score daily. Recipient mice injected with untreated T cells developed severe EAE symptoms. However, the disease score remained at zero in mice injected with PAPTP-treated (1 μM) lymphocytes (Fig. 3H). In agreement with this result, PAPTP pre-treatment abolished the infiltration of $T_{EM}$s into the spinal cord, as determined by flow cytometry analysis of the whole spinal cord tissue lysate (Fig. 3I). Consistent with this result, demyelination and neuronal degeneration were largely prevented by PAPTP in longitudinal lumbar sections of the spinal cord, as assessed by Klüver Barrera Luxol Fast Blue (LFB) and Blieschowsky stainings, respectively (Fig. EV2D,E), following the procedure described in (Theotokis et al, 2022) (see also Methods section). $T_{CM}$s infiltration into the spinal cord occurred to four-fold less extent with respect to $T_{EM}$s and was also prevented by PAPTP pre-treatment (Fig. 3I). A result consistent with these findings was obtained by macroscopic visualization of infiltration of CD45$^+$ (CD45: leukocyte common antigen) cells into the brains of mice receiving adoptive EAE at 11 days after transfer immunization (Fig. 3J). The majority of these infiltrating cells were CD4$^+$CD44$^+$CDL62$^-$ $T_{EM}$s (Fig. EV2F) as determined by immunofluorescence of the pons region (upper part of brainstem, where CD4$^+$ T cell infiltration can be detected by the adoptive transfer of $T_{MBP}$GFP cells (e.g., (Flügel et al, 1999)). CD4$^+$CD44$^+$CDL62$^-$ cells were absent in the brain of PAPTP-treated AT-EAE (data not shown). Brain astrogliosis (astrocyte activation), detected by immunohistochemistry using anti-GFAP (intermediate filament glial fibrillary acidic protein) antibody (Brahmachari et al, 2006), was similar to that observed in healthy animals for the PAPTP-treated AT-EAE (Fig. EV2G). Altogether, these results confirm that PAPTP can prevent disease in an adoptive transfer model of EAE, underlining the importance of pathological

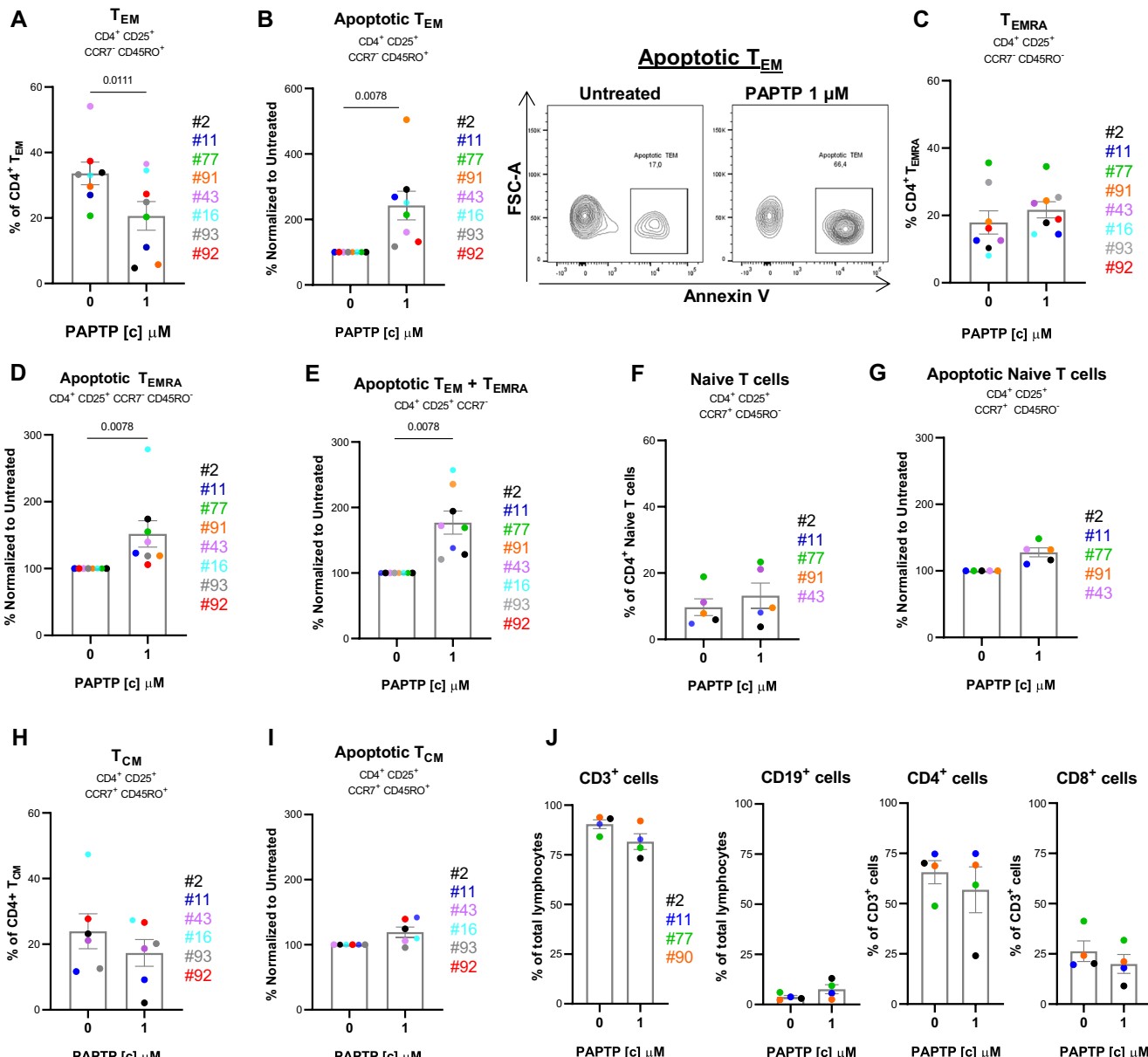

**Figure 2. PAPTP induces apoptosis in T_EM and T_EMRA cells from MS patients.**

(A) Percentage of CCR7⁻ CD45RO⁺ T_EM cells in hPBMCs from MS patients treated with 1 µM PAPTP. Cells were gated within CD4⁺ CD25⁺ CFSE⁻ autoproliferative lymphocytes (n = 8 for each group). p-value refers to Paired Student's T test. (B) Normalized apoptotic levels of CCR7⁻ CD45RO⁺ T_EM cells at specified PAPTP concentration (n = 8 for each group) ((B, D, E) for each patient, data were normalized to the untreated sample). On the right, representative density plot showing Annexin-V staining in CCR7⁻ CD45RO⁺ T_EM cells in untreated and 1 µM PAPTP treated hPBMCs. (C) Percentage of CCR7⁻ CD45RO⁻ T_EMRA cells in hPBMCs from MS patients treated with 1 µM PAPTP. Cells were gated within CD4⁺ CD25⁺ CFSE⁻ autoproliferative lymphocytes (n = 8 for each group). (D) Normalized apoptotic levels of CCR7⁻ CD45RO⁻ T_EMRA cells treated with 1 µM PAPTP (n = 8 for each group). (E) Normalized apoptotic levels of CCR7⁻ effector cells (T_EM + T_EMRA) at 1 µM PAPTP (n = 8 for each group). (F) Percentage of CCR7⁺ CD45RO⁻ naive T cells in hPBMCs from MS patients treated with 1 µM PAPTP. Cells were gated within CD4⁺ CD25⁺ CFSE⁻ autoproliferative lymphocytes (n = 5 for each group). (G) Normalized apoptotic levels of CCR7⁺ CD45RO⁻ naive T cells at 1 µM PAPTP. For each patient, data were normalized based on the untreated sample (n = 5 for each group). (H) Percentage of CCR7⁺ CD45RO⁺ T_CM cells in hPBMCs from MS patients treated with 1 µM PAPTP. Cells were gated within CD4⁺ CD25⁺ CFSE⁻ autoproliferative lymphocytes (n = 6 for each group). (I) Normalized apoptotic levels of CCR7⁺ CD45RO⁺ T_CM cells at 1 µM PAPTP. For each patient, data were normalized based on the untreated sample (n = 6 for each group). An outlier, determined using Origin6.1 algorithm, was removed from the graph. (J) Percentage of CD3⁺ T cells, CD19⁺ B cells, CD4⁺ T helper cells and CD8⁺ cytotoxic T cells in hPBMCs from MS patients treated with 1 µM PAPTP concentrations. CD3⁺ and CD19⁺ cells were gated on total lymphocytes; CD4⁺ and CD8⁺ cells were gated on CD3⁺ T cells (n = 4 for each group). (A–J) Data represent average ± SEM with superimposed individual data points for each patient. Each data point represents hPBMCs derived from a distinct patient. (B, D, E) p-values obtained from Wilcoxon test are indicated. Source data are available online for this figure.

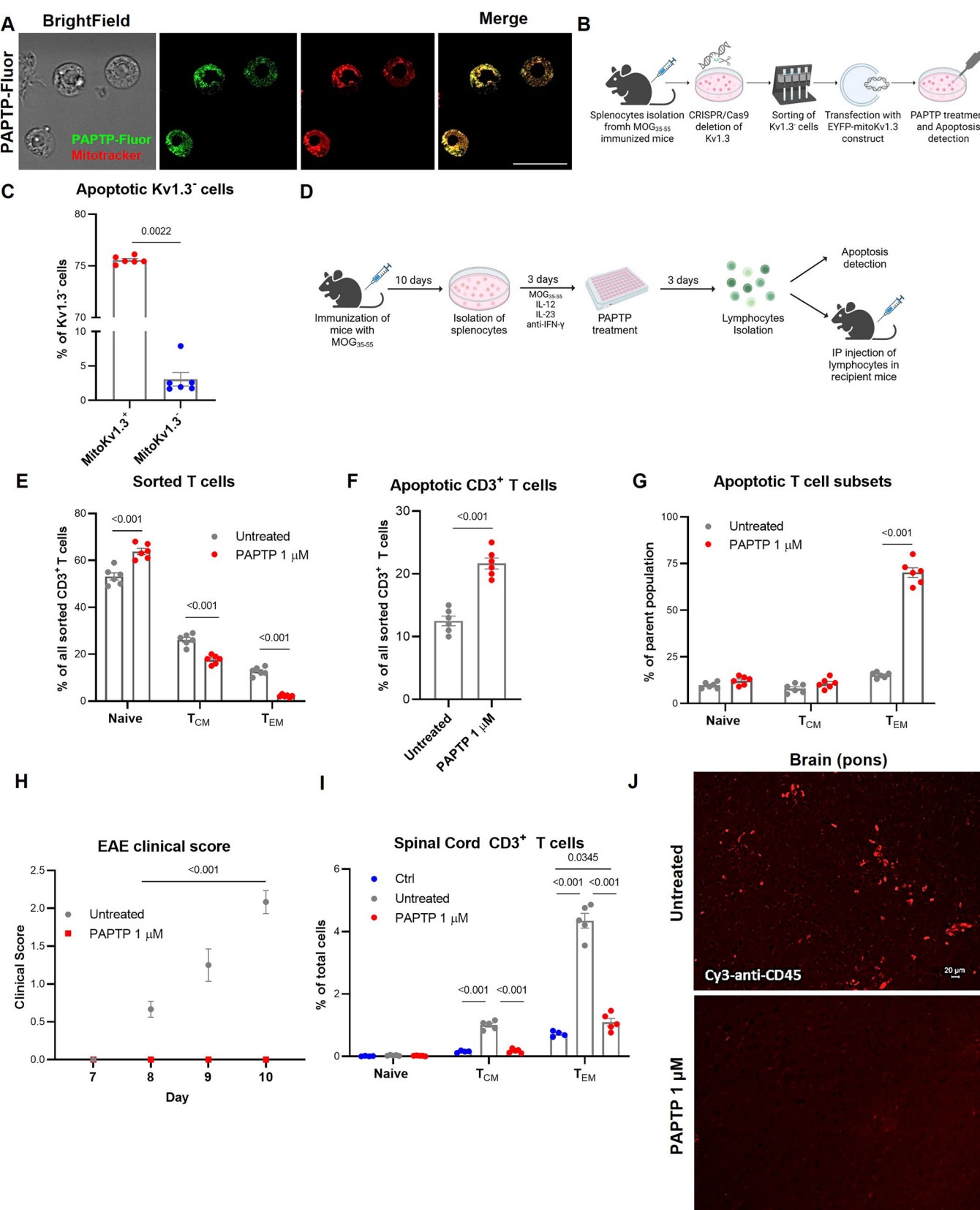

**Figure 3. PAPTP prevents EAE onset in the Adoptive Transfer Model.**

(A) Confocal microscopy image showing the accumulation of fluorescent PAPTP (PAPTP-fluor) in Mitotracker Red stained mitochondria of CD4$^+$CD25$^-$ Tconv cells isolated from healthy mice. T$_{conv}$ were treated with 100 nM PAPTP-Fluor for 30 min. The scale bar is 10 µm. The same cells shown in this representative image are also shown in Fig. EV2B at lower magnification. (B) Scheme showing the strategy for the generation of Kv1.3 knockout cells expressing the mitochondrial Kv1.3 form. Splenocytes were isolated from MOG$_{35-55}$ immunized mice and Kv1.3 was deleted using CRISPR/Cas9 technology. Cells were then sorted for Kv1.3 expression, removing splenocytes still expressing Kv1.3, and transfected with EYFP-mitoKv1.3 construct. Finally, cells were treated with 1 µM PAPTP for 48 h. Apoptosis was detected in Mito-Kv1.3$^+$ and MitoKv1.3$^-$ cells by flow cytometry. (C) Percentage of Annexin V$^+$ Mito-Kv1.3$^+$ and Annexin V$^+$ Mito-Kv1.3$^-$ cells after 1 µM PAPTP treatment for 48 h. Populations were gated on total cells (Kv1.3$^-$ cells) ($n = 6$ for each group). Data represent average ± SEM with superimposed individual data points for each animal. Indicated $p$-values refer to Mann-Whitney test. See also Fig. EV2C. (D) Scheme of the EAE Adoptive Transfer Model: Mice underwent immunization via subcutaneous injection of MOG$_{35-55}$ in complete Freund's adjuvant. Ten days post-immunization, spleens were harvested, and single-cell suspensions were prepared. These splenocytes were treated for 3 days with IL-12, IL-23, and anti-IFN-γ. Subsequently, the splenocytes were subjected to a 3-day treatment with 1 µM PAPTP, or left untreated. Antigen-specificity of the cells was confirmed by additional proliferation assays of isolated donor lymphocytes (see Methods). Lymphocytes were isolated, and the induction of apoptosis was assessed using flow cytometry. Additionally, these lymphocytes were transferred into wild-type recipient animals, and the animals' clinical scores were monitored daily for 10 days. Following the observation period, the animals were euthanized for further analysis. (E) Percentage of Naive, T$_{CM}$ and T$_{EM}$ cells of all sorted CD3$^+$ T lymphocytes ($n = 6$ for each group). (F) Percentage of apoptotic Annexin V$^+$ cells of all sorted CD3$^+$ T cells. The Trypan Blue staining gave the same result ($n = 6$ for each group). (G) Percentage of cell death in CD4$^+$ CD44$^-$CDL62$^+$ Naive, CD44$^+$CDL62$^+$ T$_{CM}$ and CD44$^+$CDL62$^-$ T$_{EM}$ subsets ($n = 6$ for each group). (H) Data represent average ± SEM of disease scores by daily scoring of mice receiving the indicated group of lymphocytes. Days 1–6: All mice had a score of 0 ($n = 6$ for each group). $p$-values from two-way ANOVA test are shown. (I) Data represent average ± SEM of the percentage of spinal cord infiltrated Naive, T$_{CM}$, and T$_{EM}$ lymphocytes (determined as in (G)) of all cells ($n = 4$ for healthy controls; $n = 5$ for mice receiving untreated and $n = 5$ for mice receiving 1 µM PAPTP-treated lymphocytes). $p$-values from two-way ANOVA test are shown. (J) Representative image (of 5 sections) of brains (upper part of the pons region) stained with PE-anti-CD45, from wild-type mice intraperitoneally injected with either untreated or PAPTP 1 µM treated, MOG$_{35-55}$ activated lymphocytes. (E-G) Data represent average ± SEM with superimposed individual data points for each animal. Indicated p-values from Unpaired Student's T test. Source data are available online for this figure.

T$_{EM}$s in the peripheral blood for immune cell infiltration into the brain and the subsequent development of EAE onset.

## PAPTP mitigates disease progression in EAE mice

Following the success of PAPTP in preventing adoptive transfer of EAE, we wondered whether the compound might also alleviate demyelination and symptoms in mice with pre-existing EAE. To test this, we treated EAE mice with a dose of PAPTP (4 nmol/g) that we had previously validated as non-toxic in various mouse models, as no apoptosis was observed in healthy tissues and there was no immune depletion or cardiotoxicity (Leanza et al, 2017; Severin et al, 2022). In a first set of experiments, after induction and onset of the first indication of EAE (limp tail), mice were treated every second day with either 4 nmol/g PAPTP (i.p.) or DMSO solvent (i.p.) for a total of 3 injections, and evaluated every day for disease symptoms (Fig. 4A). Disease developed in EAE mice as expected (Bittner et al, 2014), however, a significant alleviation of these symptoms was observed in the PAPTP-treated group (Fig. 4B). In agreement, inflammatory/demyelinating lesions (e.g., (Morales et al, 2006; Nam et al, 2021; Ucciferri et al, 2024)) of the lumbar sections of the spinal cord, assessed by Klüver-Barrera dual staining method and quantified following the protocol described in (Miyauchi et al, 2020), clearly observable in the untreated EAE, was largely reduced in the PAPTP-treated animals (Figs. 4C and EV2I). Neuronal degeneration examined using Blieschowsky staining (Fig. 4D) (see e.g., (Pyka-Fosciak et al, 2023)) and spinal cord astrogliosis were also significantly reduced (Fig. 4E). Regarding the molecular mechanism of action of PAPTP in vivo, we provided evidence that also in the case of the murine EAE model, basal mitochondrial ROS level was significantly higher in T$_{EM}$ with respect to T$_{CM}$ and naive T cells of the animals at the endpoint of the experiment shown in Fig. 4B. The latter group only responded with a significant ROS increase upon treatment with PAPTP (Fig. EV3A).

Next, we applied PAPTP 11 times in the same treatment regimen (Fig. 5A), in order to understand its long-term effects on EAE and to evaluate possible side effects on the immune system. Figure 5B reports the results of three independent series of experiments, showing that PAPTP quickly decreased the clinical score with respect to the untreated animals. Importantly, long-term treatment with PAPTP

(injected every second day for 20 days) did not lead to general immune suppression, nor depletion of red or white blood cells (including T cells and B cells) or platelets (Figs. 5C–E and EV3B,E). Neither did PAPTP change the percentage of activated macrophages or monocytes (Fig. EV3C,D). Moreover, the long-term treatment with PAPTP applied here did not cause toxicity or alter spleen and liver histology (Fig. EV3F) and the animals did not show signs of malaise and distress throughout the treatment. Hence, our findings indicate that PAPTP can relieve the symptoms of murine EAE without obvious side effects.

Strikingly, the rotarod test revealed significant amelioration of motor coordination following PAPTP treatment (Fig. 5F). In accordance, transmission electron microscopy showed that PAPTP reduced axon demyelination in the brain cortex and spinal cord after induction of EAE (Figs. 5G–I and EV3G,H) and restored the g-ratio (the ratio between the axon diameter and the myelinated fiber diameter (Wang et al, 2022)), in the spinal cord.

Myelin-activated T$_{EM}$s cells have been shown to migrate into the CNS (Rus et al, 2005), triggering inflammatory events that include microglia activation (Peruzzotti-Jametti et al, 2024). Since EAE progression-driving inflammation of the CNS leading to myelin sheaths damage is maintained by microglia (e.g., (Geladaris et al, 2024)), to evaluate whether PAPTP could ameliorate this phenomenon, we analyzed microglia morphology and activation, using IBA-1 (Ionized calcium-binding adapter molecule 1) as activation marker of microglia and of infiltrated macrophages (Walker et al, 2014). We focused on the number of IBA-1-positive cells and the length of branches, the number of endpoints, and the number of junctions in microglia (Hopperton et al, 2018). In EAE mice, microglia exhibit an amoeboid morphology with reduced ramification and an increased number of IBA-1 positive cells compared to control brains. Treatment with PAPTP reverted microglia morphology and activation, decreasing the number of activated microglial cells and restoring the number of branches, junctions, endpoints and the maximum branch length (Figs. 5J–L and EV3I). Consistent with this result is the prevention of brain astrogliosis by PAPTP (Fig. EV3J), as activated microglia can significantly contribute to the subsequent development of astrogliosis (Zhang et al, 2010).

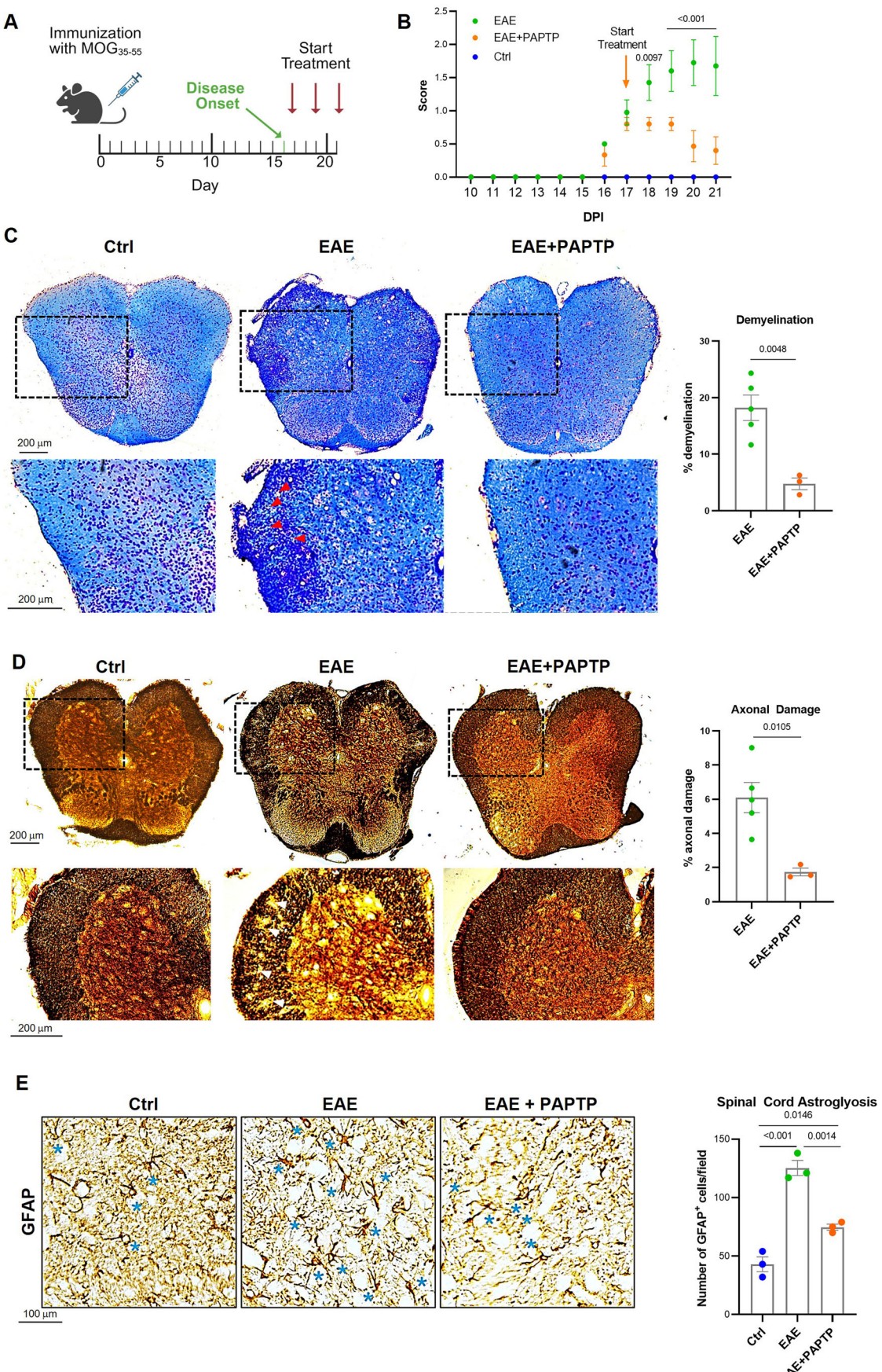

**Figure 4.  PAPTP rapidly ameliorates symptoms in the EAE mouse model.**

(A) Treatment scheme for wild-type mice immunized via subcutaneous injection of $MOG_{35-55}$: upon the manifestation of initial symptoms, treatment commenced. The treatment regimen was administered every 48 h, totaling 3 injections. Mice were euthanized a few hours after the last injection for subsequent analysis of brains and spinal cords. (B) Data represent average ± SEM of disease scores of mice of the indicated group ($n = 4$ for controls; $n = 4$ for EAE, $n = 3$ for EAE + PAPTP). PAPTP treatment was started after disease onset, when indicated. $p$-values of two-way ANOVA test are shown. (C) Representative images of transversal lumbar spinal-cord sections stained with luxol fast blue, from control ($n = 4$), EAE ($n = 4$), and EAE + PAPTP ($n = 3$) groups. Demyelinated/infiltrated areas are indicated by arrows. Scale bar, 200 μm. Right, the demyelinated area in the white matter was calculated in the EAE and EAE + PAPTP groups. (D) Representative images of transversal lumbar spinal-cord sections stained with Bielschowsky staining, from control ($n = 5$), EAE ($n = 5$), and EAE + PAPTP ($n = 3$) groups. Areas with axonal loss are indicated by arrows. Scale bar, 200 μm. Right, the axonal loss in the white matter was calculated in the EAE and EAE + PAPTP groups. (E) Average ± SEM of the number of GFAP+ cells per field in spinal cord slices of mice of the indicated groups ($n = 3$). At least 5 sections per animal were analyzed. On the left, representative immunohistochemical images of GFAP+ in spinal cord transversal slices from mice of the indicated groups. GFAP+ cells are indicated in the figure. The images were taken from the same region for each animal. The scale bar corresponds to 100 μm. $p$-values of one-way ANOVA are indicated. (C, D) Data represent average ± SEM with superimposed individual data points for each animal. $p$-values of Unpaired Student's T test are shown. Source data are available online for this figure.

## PAPTP selectively eliminates autoreactive $T_{EM}$s in EAE mice

PAPTP does not efficiently cross the BBB (Leanza et al, 2017), as we also confirmed in this study, where in the brain of EAE mice, we did not find PAPTP at detectable levels (i.e., its concentration, if any, was <0.1 nmoles/g tissue, data not shown) at the endpoint of the experiment. Therefore, we investigated whether its beneficial effects were due to elimination of autoreactive $T_{EM}$s or other T cell populations in the peripheral blood using flow cytometry analysis, keeping in mind that in EAE the most expanded CD4+ T cells are specific for the inducing myelin peptide $MOG_{35-55}$ (while clonally expanded CD8+ T cells are not specific to myelin peptides or proteins) (Saligrama et al, 2019). By applying anti-CD62L and anti-CD44 antibodies at the end of our in vivo EAE experiments, we categorized CD4+ $T_H$ cells into $T_{EM}$s (CD44+CD62L−), $T_{CM}$s (CD44+CD62L+), and naive $T_H$ cells (CD44−CD62L+) (Fig. EV4). PAPTP-treated animals exhibited a lower percentage of $T_{EM}$s in peripheral blood than EAE mice (Fig. 6A), likely due to increased levels of apoptosis in these cells (Fig. 6B). In contrast, the percentage of $T_{CM}$s and naive $T_H$ cells, as well as their levels of apoptosis, remained unchanged following PAPTP treatment (Fig. 6C–F). In agreement, the absolute number of $T_{CM}$s and naive $T_H$ cells did not significantly change in animals with EAE upon PAPTP treatment, while the absolute number of $T_{EM}$s decreased by about 50% (Fig. 6G). These data indicate that PAPTP targets $T_{EM}$s without affecting other $T_H$ populations in mice.

Altogether, these findings underscore the potential benefit of PAPTP treatment in EAE animals and suggest it can be attributed to specific targeting of $T_{EM}$s within peripheral blood and the subsequent induction of apoptosis in this immune cell population (Fig. 7).

## Discussion

We have evaluated the effect of PAPTP in the context of MS and demonstrated its ability to selectively eliminate specific subsets of autoreactive/autoproliferative T cells while preserving other non-pathogenic lymphocytes. Specifically, our findings reveal that PAPTP reduces the number of autoreactive/autoproliferative T cells in both PBMCs isolated from MS patients and in mice with MOG-induced EAE. This resulted in diminished symptoms, enhanced motor coordination, decreased microglia activation, decreased astrogliosis and decreased demyelination. Furthermore, the targeted effect of PAPTP on pathological $T_{EM}$s prevented disease onset in an adoptive

transfer model of EAE. These findings suggest that inhibition of mitochondrial Kv1.3 is a promising strategy to minimize autoreactive lymphocytes in MS. Notably, the effect of PAPTP, which does not cross the BBB, in MOG-induced chronic EAE and in EAE-AT is comparable to that observed in whole body Kv1.3 KO mice in the same experimental setting (Gocke et al, 2012). Although Kv1.3-specific toxin inhibitors (especially ShK and its variants) have previously been shown to reduce proliferation and activation of autoreactive T cells, PAPTP is the only drug that can trigger their death, as further confirmed in the present study. Consistent with this, ShK has been reported to be less effective than PAPTP in EAE-AT (Beeton et al, 2001b), whereas transient application of PAP-1, a BBB permeable Kv1.3 inhibitor, has been shown to enhance optic nerve axonal damage in the MOG-induced rat EAE model (Stokely et al, 2008). From a clinical perspective, although there are many disease modifying therapies that effectively prevent MS relapse, there is still a need for safer, more targeted therapy that will deplete pathogenic immune populations while sparing other subsets.

Death of $T_{EM}$s induced by PAPTP is due to its specific inhibition of mitochondrial Kv1.3, which triggers severe mitochondrial dysfunction and subsequent apoptosis. Not all $T_{EM}$s underwent apoptosis in our EAE model, but those remaining after treatment (around 50%) were insufficient to sustain pathology progression, suggesting a threshold effect. In agreement with this assumption, a two-fold increase in the CD4+CD45+ effector T cell absolute number in the lymph nodes was reported to increase three-fold the clinical score in an EAE model (Koutrolos et al, 2014). On the contrary, in a rat EAE model, a decrease in the number of CD4+CCR7− $T_{EM}$s by 12% in PBMC correlated with a two-fold reduction of the clinical score (Li et al, 2012). Likewise, IL-7Rα–blocking antibody reduced the clinical scores by 50% in an EAE model, by the partial depletion of peripheral effector memory and of proliferating naive T cells, without reducing the number of central memory T cells (Lee et al, 2011). In our experiments, PAPTP did not reduce the absolute number of naive and $T_{CM}$ cells in vivo in EAE, in contrast to $T_{EM}$s. The narrow target of PAPTP provides an opportunity to eliminate disease-causing lymphocytes without severe immunosuppression, toxicity, or other adverse events (Leanza et al, 2017; Severin et al, 2022), which are the major shortcomings of most clinically used MS drugs. Our findings showing that PAPTP can selectively kill autoproliferative patholo-gic $T_{EM}$s in PBMC isolated from HLA DRB1*15-positive RRMS patients, underlines the broad therapeutic potential of PAPTP, since HLA-DR15 is the strongest genetic risk factor for MS (e.g., (Drosu et al, 2024)).

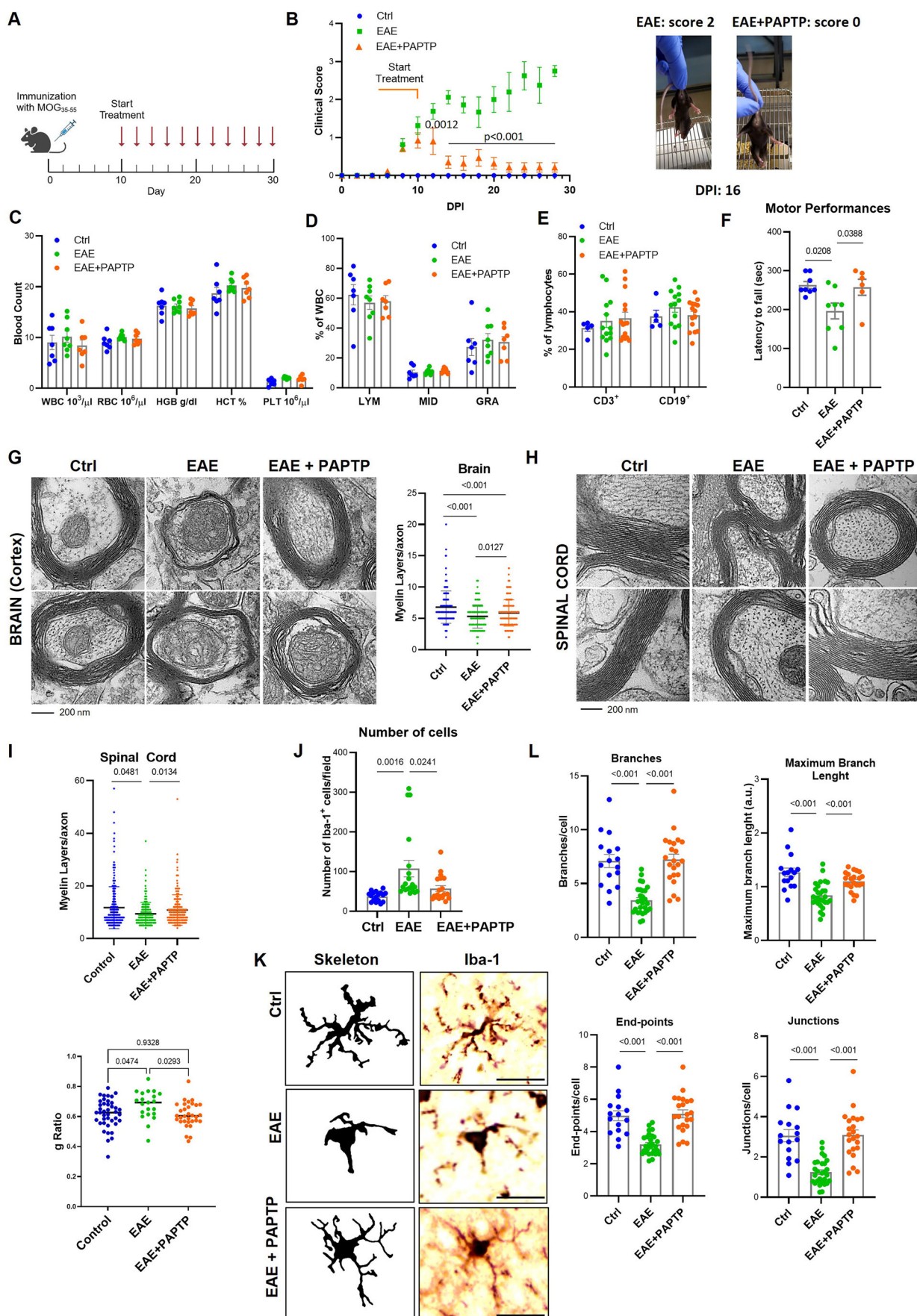

**Figure 5. Long-term PAPTP treatment ameliorates symptoms in the EAE mouse model.**

(A) Treatment scheme for wild-type mice immunized via subcutaneous injection of MOG$_{35-55}$: upon the manifestation of initial symptoms, typically around day 10 post-immunization, treatment commenced. The treatment regimen was administered every 48 h, totaling 11 injections. Mice were euthanized the day following the final injection for subsequent analysis of peripheral blood, brains, and spinal cords. (B) Data represent average ± SEM of disease scores of mice of the indicated group ($n = 9$ for controls; $n = 14$ for EAE, $n = 15$ for EAE + PAPTP). PAPTP treatment was started at day 10 post immunization (DPI). $p$-values of two-way ANOVA test are indicated. On the right representative photos taken at 16 DPI under the indicated conditions are shown. (C) Quantitative analysis of hematological parameters in the blood of mice from specified experimental groups at the endpoint of the experiment. The measurements include counts of white blood cells (WBC), red blood cells (RBC), platelets (PLT), hemoglobin and hematocrit levels. As an example, WBC count is around $8 \times 10^3/\mu l$, while RBC count is $10 \times 10^6/\mu l$. Data represent average ± SEM ($n = 7$ for control and EAE + PAPTP groups; $n = 8$ for EAE). (D) Percentages of lymphocytes (LYM), monocytes (MID), and granulocytes (GRA) in peripheral blood of mice of the indicated group at the endpoint of the experiment, evaluated using a blood counter. Data represent average ± SEM ($n = 7$ for control and EAE + PAPTP groups; $n = 8$ for EAE). (E) CD3$^+$ T cell and CD19$^+$ B cell percentages within the total lymphocyte population in peripheral blood samples collected from mice belonging to the indicated groups at the experimental endpoint ($n = 5$ for controls; $n = 13$ for EAE, and $n = 15$ for EAE + PAPTP groups). (F) Latency to fall (in seconds) of mice of the indicated group evaluated using the rotarod test ($n = 8$ for controls and EAE mice; $n = 6$ for EAE + PAPTP group). (G) Representative Transmission Electron Microscopy images showing neuronal myelination in brain cortexes of mice from the indicated groups at the experimental endpoint. Images were taken from the same region of the brain of each animal. Scale bar indicated in the figure. On the right, quantification of the average ± SEM of myelin layers per axon. Each point represents a different axon ($n = 3$ sections for controls, and $n = 3$ for EAE, and $n = 4$ for EAE + PAPTP groups). (H) Representative Transmission Electron Microscopy images showing neuronal myelination in spinal cords of mice from the indicated groups at the experimental endpoint. Scale bar indicated in the figure. (I) Upper panel: Quantification of the average ± SEM of myelin layers per axon. Each point represents a different axon ($n = 5$ sections for controls, and $n = 3$ for EAE, and $n = 4$ for EAE + PAPTP groups). Lower panel: g-ratio (axon diameter/diameter of myelinated fiber) as determined from TEM images for individual axons of the spinal cord, for the indicated groups. (J) Average ± SEM of the number of Iba-1$^+$ cells per field in brain slices of mice of the indicated groups ($n = 3$ for each group, 6 slices/animal were analyzed). (K) Representative binary images of individual microglia. The scale bar corresponds to 25 μm. (L) Average ± SEM of the number of branches, maximal branch length, number of junctions, and end-points in Iba-1$^+$ microglia cells in brain slices of mice of the indicated group. Each data point represents a single cell ($n = 3$ for each group). (C–F) Data represent average ± SEM with superimposed individual data points for each animal. (J, L) $p$-values obtained in one-way ANOVA test. (F, G, I) $p$-values of Kruskal–Wallis test or of Tukey's multiple comparison test (I lower panel). Source data are available online for this figure.

The identification of this novel tool adds specificity to the goal of eliminating disease-causing lymphocytes in MS. Our data show that PAPTP specifically kills human pathologic T$_{EM}$s and to some extent T$_{EMRA}$s, while sparing other T and B cell populations, due to being an inhibitor of the dominant functional K$^+$ channel in that cell type following chronic activation (Beeton and Chandy, 2005; Ribas et al, 2016). In rodents, dormant T cells, including unstimulated normal splenic and lymph node T cells, express only a few Kv1.3 channels per cell (Beeton et al, 2001b). However, acutely activated cells that have been stimulated once or twice, express approximately 200 Kv1.3 channels, and chronically activated cells that have been stimulated at least eight times express approximately 1500 Kv1.3 channels. The intermediate-conductance calcium-dependent K$^+$ channel KCa3.1 reduces concomitantly from around 325 channels per acutely activated cell to between 20 and 100 per chronically activated cell. This high Kv1.3:low KCa3.1 expression ratio is a functional marker of chronically activated rat and human T lymphocytes (Beeton et al, 2003). That being the case, high Kv1.3 expression is not sufficient for the specificity of mitochondrial Kv1.3-dependent apoptosis. High basal ROS production, typical of chronically activated T$_{EM}$s (Bantug et al, 2018), is also required as it renders cells vulnerable to oxidative stress (Wahl et al, 2012). Indeed, we have previously demonstrated that high expression of Kv1.3 along with oxidative stress is sufficient to trigger cell death in CCR7$^-$ cells (Leanza et al, 2017), but PAPTP does not cause a general immune depletion in mice injected 4 times (Leanza et al, 2017) or 10 times daily (Severin et al, 2022) (in both cases the drug was administered i.p., as in the present study). The results reported here showing a lack of general immune depletion and of alteration of organ histology is fully in agreement with our previous findings (Leanza et al, 2017; Li et al, 2022; Severin et al, 2022). Healthy T cells do not die after mitochondrial Kv1.3 inhibition as they have a lower initial ROS level and more efficient antioxidant systems than chronically activated T$_{EM}$s (Wahl et al, 2012). Mitochondrial and cell metabolism differ between resting, acutely activated and chronically activated disease-causing lymphocytes. Autoreactive lymphocytes are characterized by a depleted antioxidant pool and do not engage very

much in aerobic glycolysis, instead relying on oxidative phosphorylation (OXPHOS) (Wahl et al, 2012), which results in increased ROS production. Indeed, when compared to cells obtained from healthy individuals, T cells derived from patients with lupus demonstrate both mitochondrial membrane hyperpolarization and increased ROS (Kesarwani et al, 2013; Wahl et al, 2012). Bz-423, a small molecule that inhibits complex V, causes hyperpolarization and thus increases ROS in alloreactive T cells (Wahl et al, 2012). Although this compound did not induce generalized lymphocyte apoptosis, it triggered apoptosis in alloreactive lymphocytes, significantly reducing the number of disease-causing T cells and thus disease symptoms (Gatza et al, 2011). Likewise, Bz-423 was shown to attenuate lupus by selective killing of autoimmune lymphocytes (Bednarski et al, 2003; Sundberg et al, 2009). Because autoreactive T cells share metabolic characteristics with alloreactive T cells (Wahl et al, 2012), and display high Kv1.3 expression, hyperpolarized mitochondrial membrane potentials and elevated basal ROS levels, we predicted they would be selectively and efficiently eliminated by PAPTP. Please note that this phenomenon does not occur in activated microglia that are also characterized by upregulation of Kv1.3 expression (Sarkar et al, 2020) and is not expected to take place in the case of other Kv1.3 expressing brain-resident (or infiltrating) cells, as PAPTP does not pass the BBB. However, PAPTP is able to kill in the peripheral blood autoreactive T cells and thereby decreases their overall infiltration into the brain and spinal cord, as we observed in agreement with (i) the fact that EAE was induced by activation of myelin-specific CD4$^+$ T cells; (ii) the ability of activated CD4$^+$ effector T cells to invade CNS (Bartholomäus et al, 2009; Rus et al, 2005) and (iii) the presence of CD4$^+$ cells, producing exclusively T helper type 1 (Th1) cytokines (Williams et al, 2011), but the complete lack of CD8$^+$ T cells in the CNS, detected at 11 days after transfer immunization in an adoptive transfer model of EAE (Yura et al, 2001). In agreement with the negligible infiltration of autoreactive myelin-specific CD4$^+$ T cells into the spinal cord and the pons region of the brainstem, we observed upon pre-treatment of PBMC with PAPTP in the adoptive transfer model, the motor coordination of the animals with EAE significantly improved. In

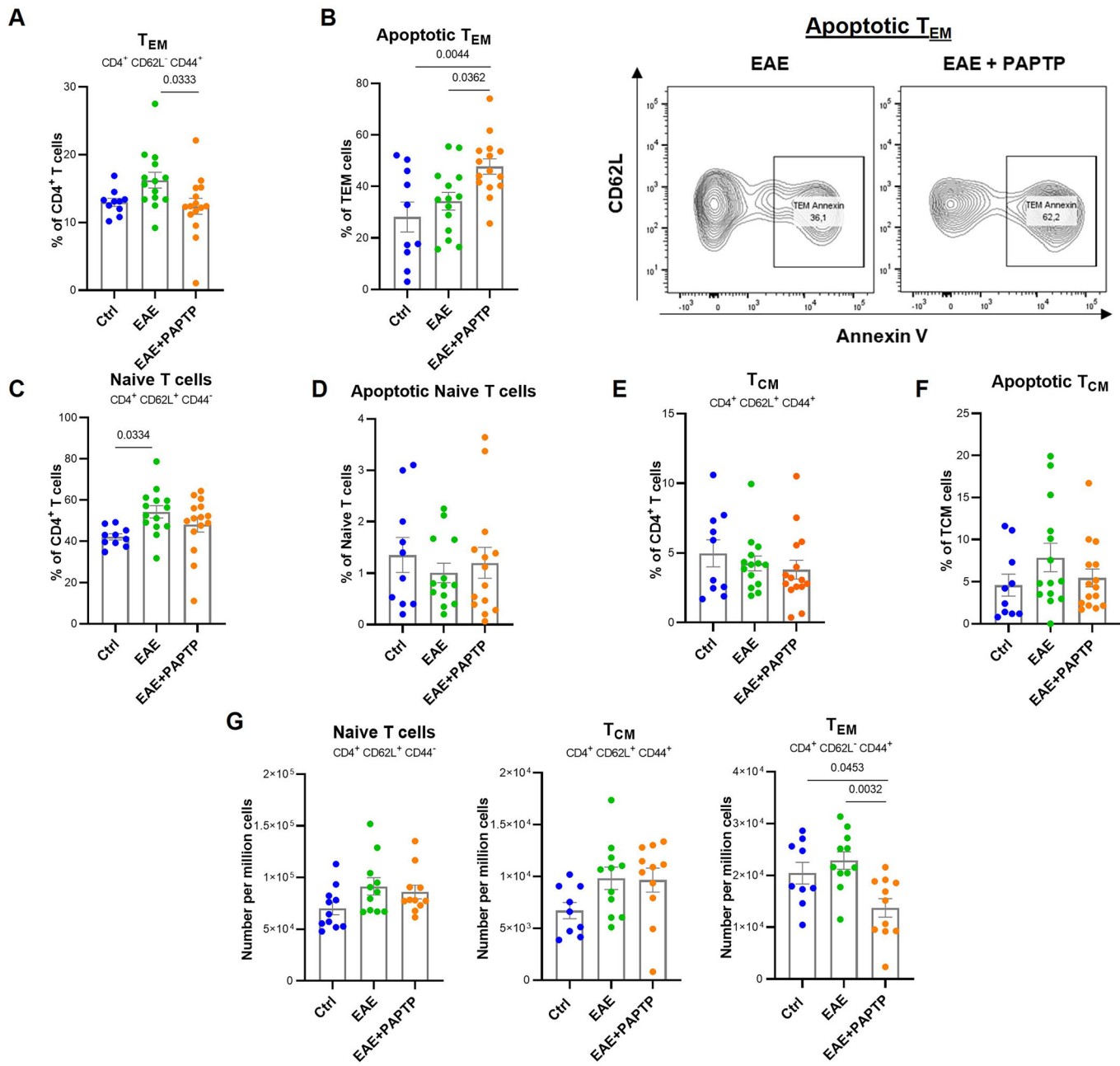

**Figure 6. PAPTP selectively eliminates autoreactive T_EM cells in peripheral blood of EAEmice.**

(A) CD62L⁻ CD44⁺ T_EM cells percentages within CD4⁺ lymphocytes in peripheral blood samples collected from mice belonging to the indicated groups at the experimental endpoint (in (A) to (F)): n = 10 for controls; n = 14 for EAE, and n = 15 for EAE + PAPTP group). (B) Apoptotic CD62L⁻ CD44⁺ T_EM cells percentages of CD4⁺ T_EM lymphocytes. For apoptosis determination, peripheral blood samples were collected from mice belonging to the indicated groups at the experimental endpoint and stained with Annexin V. On the right, representative dot plot showing Annexin V staining of CD62L⁻ CD44⁺ T_EM cells in mice of the indicated groups. (C) CD62L⁻ CD44⁻ naive T cell percentages within CD4⁺ lymphocytes in peripheral blood samples collected from mice belonging to the indicated groups at the experimental endpoint. (D) Apoptotic CD62L⁻ CD44⁻ naive T cell percentages of CD4⁺ naive T lymphocytes. Apoptosis was determined as in (B). (E) CD62L⁺ CD44⁺ T_CMs percentages within CD4⁺ lymphocytes in peripheral blood samples collected from mice belonging to the indicated groups at the experimental endpoint. (F) Apoptotic CD62L⁺ CD44⁺ T_CMs percentages of CD4⁺ T_CM lymphocytes. Apoptosis was determined as in (B). (G) Quantification of CD4⁺ T cell subsets in peripheral blood. The number of CD4⁺ CD62L⁺ CD44⁻ naive T cells, CD4⁺ CD62L⁺ CD44⁺ T_CM, and CD4⁺ CD62L⁻ CD44⁺ T_EM per million cells was measured at the experimental endpoint (n = 10 for controls, n = 11 for EAE and EAE + PAPTP groups). (A–G) Data represent average ± SEM with superimposed individual data points for each animal. p-values from one-way ANOVA test. Source data are available online for this figure.

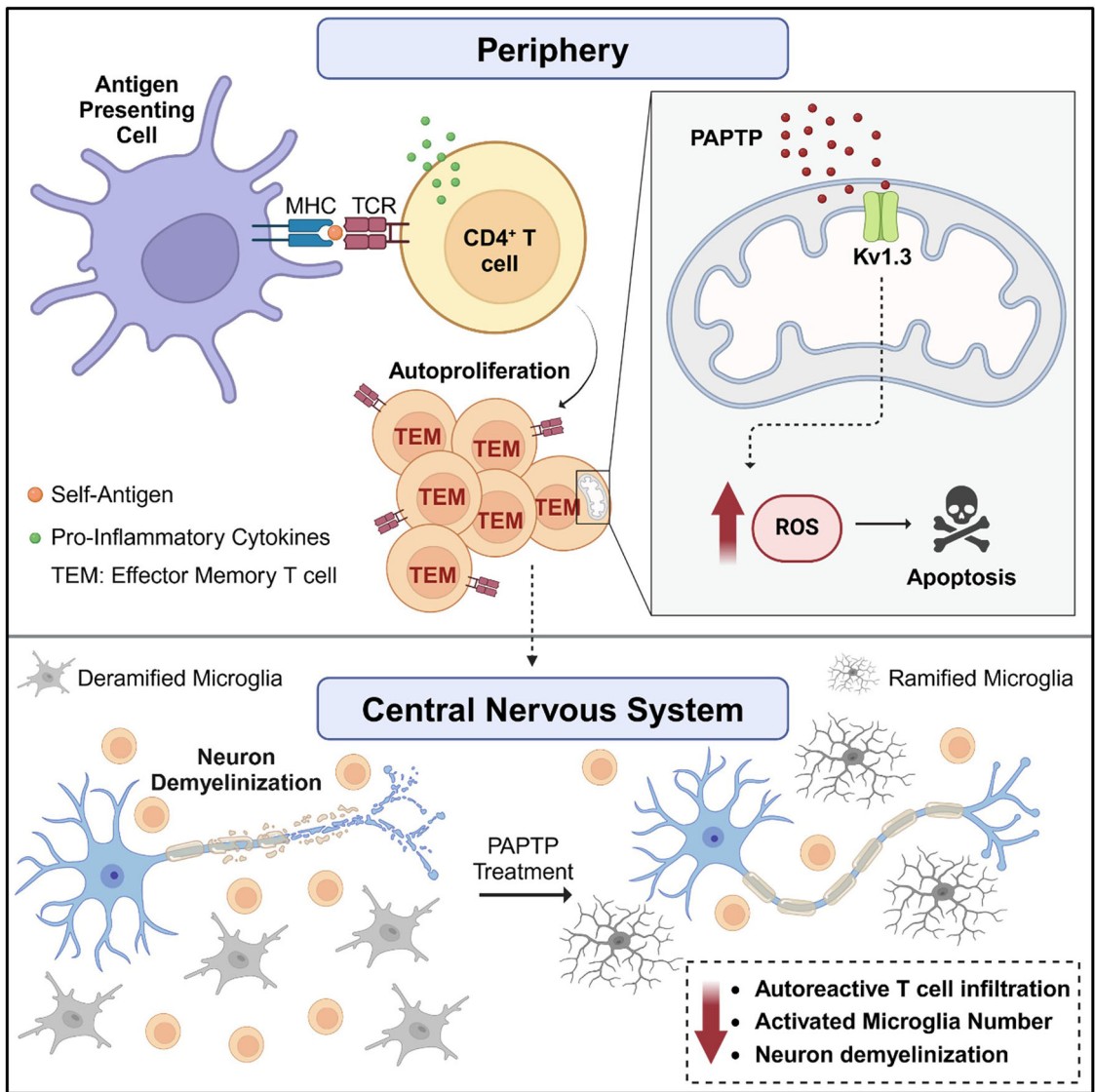

**Figure 7.  Proposed mechanism of action of the mitochondrial Kv1.3 inhibitor PAPTP against multiple sclerosis.**

In the periphery, the presentation of self-antigens by antigen-presenting cells (APCs) to T helper lymphocytes (CD4+ T cells) results in their autoproliferation and activation (top left). These self-antigen-specific lymphocytes, exhibiting an effector memory phenotype ($T_{EM}$), then migrate into the central nervous system (CNS). Within the CNS, they release inflammatory cytokines, initiating an inflammatory cascade that ultimately leads to neuron demyelination and axonal loss (bottom left). In this inflammatory milieu, microglia typically appear de-ramified with fewer processes and more round cell bodies. PAPTP, by inhibiting the mitochondrial Kv1.3 ion channel in autoreactive T cells, increases mitochondrial ROS production, ultimately inducing apoptosis. The induction of apoptosis in these autoreactive T cells reduces their migration into the CNS, thereby decreasing inflammation and ameliorating neuron demyelination. In this context, microglia exhibit a more ramified phenotype and their numbers decrease.

addition, demyelination, observed in the cortex region of the brain and in the spinal cord in the EAE model using transmission electron microscopy (as in e.g., (Dupree et al, 2022)) and Klüver-Barrera Luxol Fast Blue stain (as in (Miyauchi et al, 2020), was less evident. Neuronal degeneration was also decreased upon PAPTP treatment. Remyelination, whose efficiency decreases with disease progression and chronic inflammation, is the normal repair mechanism of demyelination. Although it is well recognized that the process of demyelination is orchestrated by a complex network of cells, microglia play a critical role in the maintenance of inflammation-related demyelination and neuroaxonal injury in active lesions (Häusler and Weber, 2024).

PAPTP likely limits the inflammatory response in EAE, as the treatment decreased the number of IBA1-positive, activated microglia, known to contribute to persistent neuroinflammation and demyelination (Klaver et al, 2013).

PM-specific non-permeable toxin Kv1.3 inhibitors, such as ShK, were shown to strongly decrease the clinical score in EAE-AT models (Beeton et al, 2001a; Yuan et al, 2018) and this effect was ascribed to their action on $T_{EM}$s. Our data are in agreement with these findings and show a selective decrease of $T_{EM}$s both in ex-vivo and in-vivo settings. However, an eventual contribution of additional targets of PAPTP in alleviating disease progression

cannot be fully excluded. The finding that the pathology onset and relevant $T_{EM}$s infiltration into the spinal cord was completely prevented by PAPTP in the EAE-AT model strongly suggest a crucial role for $T_{EM}$s Kv1.3, however, the relevance, if any, of the slight effect of PAPTP on $T_{CM}$s in EAE-AT, remains to be established. Interestingly, active $CD8^+$ memory T cells highly expressing Kv1.3 were found to be infiltrated in the kidney of patients suffering from another autoimmune disease, lupus nephritis (LN) (Khodoun et al, 2020). Strikingly, selective down-regulation of Kv1.3 expression in patients' PBMC using siRNA against Kv1.3 loaded into nanoparticles coated with anti-CD45RO prolonged survival of humanized LN mice when applied prior to engraftment. This result suggests that the same strategy might also be useful against $CD4^+$ memory T cells in the context of multiple sclerosis, although such a hypothesis remains to be tested (Cañas et al, 2022). Likewise, the effect of the promising, highly Kv1.3-selective natural immunosuppressive peptide, Vm24, shown to reduce delayed-type hypersensitivity reactions in rats in vivo (Varga et al, 2012), remains to be investigated in EAE.

Other channels, including KCa3.1 and two-pore potassium channels such as TASK-1 have been proposed as potential targets in T lymphocytes following in vivo experiments in knockout mice and from pre-clinical pharmacological studies (Bittner and Meuth, 2013). A293, an aromatic carbonamide high affinity TASK-1 inhibitor, showed benefit in EAE model, especially as prophylactic treatment (Bittner et al, 2012) but potential side effects were not studied. The relative importance of Kv1.3 and KCa3.1 as pharmacological targets appears to vary based on the specific subset of lymphocytes and their level of activation. KCa3.1 appears to play a notable role in activated T cells, $T_{CM}$s, and $T_H1/2$ cells. However, Kv1.3 and KCa3.1 have been proposed to regulate antigen-specific memory T cell functions in a cooperative and compensatory manner in Kv1.3 KO rats (Chiang et al, 2017). Additionally, calcium release-activated calcium (CRAC) channels in T cells are crucial for autoimmune inflammation. Their regulators, STIM1/2, play an essential role in the pro-inflammatory function of both $T_H1$ and $T_H17$ cells, including in the context of MOG peptide-induced EAE (Feske et al, 2005; Feske et al, 2015). In agreement with our findings, $CD4^+$-specific deletion of STIM1/2 prevented autoreactive T cell function, demyelination and infiltration of $CD45^+$ cells into the CNS, and disease onset (Ma et al, 2010). However, the small molecule inhibitors of store-operated calcium entry mediated by CRAC may cause the development of colorectal cancer in the long term (Letizia et al, 2022). PAPTP instead, by killing Kv1.3-expressing cancer cells, exerts a potent anti-tumoral effect (Leanza et al, 2017; Li et al, 2022; Severin et al, 2022).

Although NAT prevents migration of pathogenic T cells into the brain, and therefore exerts a therapeutic effect in RRMS patients, its long-term use causes lymphocytosis (Callegari et al, 2021). Moreover, $CD4^+$ $T_H1$ cells can accumulate in the peripheral blood of patients treated with NAT (Jelcic et al, 2018), and those expressing high levels of Kv1.3 are associated with chronic inflammatory bowel diseases (Unterweger et al, 2021). Reducing the number of pathogenic lymphocytes in the peripheral blood by inhibiting Kv1.3 is thus a better strategy to prevent disease onset and progression. Dimethyl fumarate (DMF), an FDA approved drug for the treatment of relapsing-remitting multiple sclerosis (RRMS), was recently shown to reduce the proportion and absolute

number of $T_{EM}$, $T_{CM}$ and $T_{EMRA}$ while increasing the proportion of naive T cells in DMF-treated MS patients, likely due to the ability of the drug to induce apoptosis in vitro (Wu et al, 2017). DMF rarely gives rise to a side-effect, namely PML, a polyoma virus infection-triggered disease, likely linked to the DMF-induced depletion of $CD8^+$ T cells and $T_{EMRA}$s. It remains to be established if PAPTP, by reducing $T_{EMRA}$s in the blood, might have a similar adverse effect on long-term treatment. However, the mice in our study remained disease-free during the entire course of the PAPTP treatment, similarly to rhesus macaques treated with PAP-1 (the precursor of PAPTP) where $T_{CM}$ function required for antiviral response had been preserved (Pereira et al, 2007).

# Methods

**Reagents and tools table**

| Reagent/Resource | Reference or Source | Identifier or Catalog Number |
| --- | --- | --- |
| **Experimental models** | | |
| C57BL/6N (*M. Musculus*) | Charles River | 027 |
| **Recombinant DNA** | | |
| pJK-EYFP-mito-KV1.3 | Szabo et al, PNAS 105, 14861–14866, 2008 | |
| **Antibodies** | | |
| Anti-human CD4-PerCP | BD | 550631 |
| Anti-human CD25-PECy7 | BD | 557741 |
| Anti-human CCR7-BV421 | BD | 562555 |
| Anti-human CCR7-FITC | R&D | FAB197F |
| Anti-human CD45RO-PE | Thermo Fisher Scientific | 12-0457-42 |
| Anti-human CD8-PE | BD | 345773 |
| Anti-human CD3-PECy7 | BD | 341111 |
| Anti-human CD19-BV421 | BD | 562440 |
| Anti-mouse CD45-APCCy7 | BD | 557659 |
| Anti-mouse CD44-PE | BD | 553134 |
| Anti-mouse CD62L-FITC | BD | 561917 |
| Anti-mouse CD4-PerCP | BD | 553052 |
| Anti-mouse CD25-PECy7 | BD | 552880 |
| Anti-mouse CD3 PE-Cy7 | Sony | 1101100 |
| Anti-mouse CD19 BV421 | BD | 562701 |
| Anti-mouse CD11b PE | BD | 557397 |
| Anti-mouse CD19 FITC | BD | 553785 |
| CD80 PerCp-Cy5 | BioLegend | 104722 |
| Anti-IFN-γ (clone XMG1.2) | Invitrogen/e-Bioscience | 14-7311-85 |
| Anti-mouse CD62L-FITC (clone MEL-14) | Cell Signaling | 76378 S |
| Anti-mouse CD44-APC | BD Pharmingen | 561862 |
| Biotin-conjugated anti-Kv1.3 antibody | Alomone Labs | APC-101B |

| Reagent/Resource | Reference or Source | Identifier or Catalog Number |
|---|---|---|
| Anti-mouse CD45 PE (clone 30-F11) | e-Bioscience | 12-0451-82 |
| Anti-mouse CD44 FITC (clone IM7) | e-Bioscience | 11-0441-82 |
| Anti-mouse CD62L PE (clone MEL14) | BD-Pharmingen | 561918 |
| Anti-mouse CD4 APC (clone GK1.5) | Biolegend | 100412 |
| Anti-mouse Iba1 | Thermo Fisher Scientific | PA5-27436 |
| Anti-mouse GFAP | Dako | Z0334 |
| Goat anti-rabbit IgG (H + L) (HRP polymer) | Abcam | AB214880 |
| **Oligonucleotides and other sequence-based reagents** | | |
| CRISPR/Cas9 construct targeting Kv1.3 | Santa Cruz Inc | Sc-421214 |
| **Chemicals, enzymes and other reagents** | | |
| LymphoSep | Biowest | L0560 |
| Iscove's modified Dulbecco's medium (IMDM) | Gibco | 12440-053 |
| 2 mM L-glutamine | Gibco | 25030024 |
| Penicillin/streptomycin | Gibco | 15140122 |
| HEPES | Gibco | 15630080 |
| AIM-V | Thermo Fisher Scientific | 12055091 |
| Sodium pyruvate | Gibco | 11360070 |
| NEAA | Gibco | 11140050 |
| RPMI-1640 | Gibco | 21875034 |
| FBS | Gibco | A5256701 |
| DNaseI | Thermo Fisher Scientific | AM2222 |
| ShK-F6CA (5-Fam-ShK) | Vivitide | SHK-3746-PI |
| ShK | Alomone Labs | STS-400 |
| Pepsin Digest All | Thermo Fisher Scientific | 003009 |
| BD Cytofix/Cytoperm | BD | 554714 |
| MOG$_{35-55}$ | Espikem | EPK1 |
| MOG$_{35-55}$ | MedchemExpress | HY P1240 |
| Incomplete Freund's Adjuvant | BD | 263910 |
| Incomplete Freund´s adjuvant | Santa Cruz Inc. | sc-24019 |
| Desiccated M. Tuberculosis H37Ra | BD | 231141 |
| Pertussis toxin | Quadratech Diagnostic | QTXAG-108-50 |
| Pertussis toxin | Merck | P7208-5046 |
| RBC cell lysis buffer | Biolegend | 420302 |
| mIL-12 | R&D | 419-ML-010CF |
| mIL-23 | R&D | 1887-ML/CF |
| BD Pharm Lyse™ Lysing Buffer | BD | 555899 |

| Reagent/Resource | Reference or Source | Identifier or Catalog Number |
|---|---|---|
| Ficoll gradient/ Histopaque 1077 | Merck | 10771-500 ML |
| Fc receptor blocking reagent | BioLegend | 101302 |
| Streptavidin-conjugated microbeads | Miltenyi Biotec | 130-048-101 |
| LS columns | Miltenyi Biotec | 130-042-401 |
| Annexin V APC | Thermo Fisher Scientific | A35110 |
| Annexin V FITC | Thermo Fisher Scientific | A13199 |
| CFSE | Thermo Fisher Scientific | C34570 |
| MitoSOX | Thermo Fisher Scientific | M36008 |
| TMRM | Thermo Fisher Scientific | 134361 |
| Mitotracker RedCMXRos | Thermo Fisher Scientific | M7512 |
| Formalin solution | Merk | HT501128-4L |
| DAB Substrate | Abcam | AB64238 |
| Eukitt Quick-hardening mounting medium | Merck | 03989 |
| Cyclosporin H | Merck | SML1575 |
| **Software** | | |
| FlowJo Software v10.10.0 | BD | |
| Ilastik | https://www.ilastik.org/documentation/counting/counting | |
| GraphPad Prism 8 | | |
| ImageJ/Fiji | | |
| Biorender | https://www.biorender.com/ | |
| **Other** | | |
| Pan mouse T cell isolation kit | Miltenyi Biotec | 130-095-130 |
| Pierce™ BCA Protein Assay Kit | Thermo Fisher Scientific | 23225 |
| Bio-Plex Pro Assay | BioRad | 10014905 |
| Luxol Fast Blue | Bio-Optica | 04-200812 |
| Bielschowsky | Bio-Optica | 04-040805 |
| H&E staining kit | Bio-Optica | 04-061010 |
| Mouse to Mouse HRP (DAB) Staining System | Histo-Line Laboratories | MTM001 |
| BD LRS Fortessa X20 flow cytometer. | BD | |
| RotaRod | Ugo Basile, Italy | |
| BTX X2/SC Gemini electroporator | BTX | |
| LEICA TCS SL confocal microscope | Leica | |
| CELL-DYN Emerald | Abbott Laboratories | G4-9513/R06 |
| Tecnai G2 Spirit transmission electron microscope | FEI Electron Microscopes | |
| Leica DM4000 microscope | Leica | |

| Reagent/Resource | Reference or Source | Identifier or Catalog Number |
|---|---|---|
| Leica DM6 B microscope | Leica | |
| S3e™ Cell sorter | BioRad | |
| Leica Stellaris 8 | Leica | |

## Patients

Peripheral blood was collected from patients diagnosed with Relapsing Remitting MS (with age ranging from 15 to 63 years) and who received more than 6 infusions of Natalizumab. Patients were recruited at the Multiple Sclerosis Centre in Padua. The study was approved by the local Ethics Committee (Protocol Number 5326/AO/22) and a written informed consent was obtained from all the participants at the time of blood sampling. The experiments conformed to the principles set out in the WMA Declaration of Helsinki and the Department of Health and Human Services Belmont Report.

## Peripheral blood mononuclear cells isolation

Peripheral blood mononuclear cells (PBMCs) of the patients were isolated by leukapheresis using LymphoSep (Biowest L0560) density gradient centrifugation. Isolated PBMCs were cryopreserved in freezing media containing 10% dimethyl sulfoxide (DMSO) and 90% fetal bovine serum (FBS, Gibco, 10270106) at −196 °C. PBMCs from MS patients treated with Natalizumab were typed for HLADRB1*15:01 using Illumina Sequencing/PacBio Sequencing technology (HistoGenetics, LLC, NY, USA). Subsequently, only samples positive for HLA-DRB1*15:01 underwent assessment in the autoproliferation assay.

## Autoproliferation assay

The autoproliferation assay exclusively utilized PBMC samples that tested positive for the HLA-DRB1*15:01 subtype. This selection was made due to the heightened likelihood of autoproliferation within this specific subtype of individuals with multiple sclerosis. Cryopreserved PBMCs were thawed in Iscove's Modified Dulbecco's Medium (IMDM, GE Healthcare) supplemented with 100 U/mL penicillin/streptomycin (Gibco, 15140122) and 2 mM L-glutamine (Gibco, 25030024). Subsequently, the PBMCs underwent centrifugation and were washed once with serum-free AIM-V medium (Thermo Fisher Scientific, 12055091) supplemented with 2 U/mL Ambion DNaseI (ThermoFisher). This step aimed to prevent cell clumping and was followed by a 15-min incubation at 37 °C. After two washes with Saline Phosphate Buffer (PBS), the cells were resuspended at a concentration of $1 \times 10^6$ cells/mL in RPMI-1640 medium (Gibco, 21875034). Following this, the cells were incubated with 1 µM carboxyfluorescein diacetate N-succinimidyl ester (CFSE, Thermo Fisher, C34570) for 3 min at room temperature. The labeling process was stopped by quenching with a 5x excess volume of RPMI-1640 medium containing 10% FBS, and an aliquot of cells was analyzed by flow cytometry. Following an additional wash step with AIM-V, CFSE-labeled cells were seeded into a 96-well plate at a density of $2 \times 10^5$ PBMCs/200 µL per well in AIM-V medium. The cells were incubated at 37 °C with 5% $CO_2$ for 7 days and, after this period, autoproliferation was assessed by flow cytometry analysis as mean of CFSE dilution. Post-assessment of autoproliferation, cells were treated with either 1 µM or 5 µM PAPTP, 100 nM ShK, or DMSO for 18 h, and subsequent apoptosis was evaluated by flow cytometry. Data were analyzed using the BD LRS Fortessa X20 flow cytometer.

## MitoSOX assay

Cryopreserved unmutated HLA-DRB1*15:01 patient PBMCs, preserved in a solution containing 10% dimethyl sulfoxide (DMSO) and 90% FBS, were thawed and, after an 8-day incubation period, cells were stained with antibodies including CD4-PerCP (BD, 550631), CD25-PECy7 (BD, 557741), and CCR7-BV421 (BD, 562555) for 10 min at room temperature in the dark, and then incubated with 2.5 µM MitoSOX (Thermofisher, M36008) and 2 µM Cyclosporine H (Merk, SML1575) for 30 min at 37 °C in HBSS. Finally, the stained cells were treated with either 1 µM or 5 µM PAPTP, or 100 nM ShK for 30 min, while the control group received no treatment (DMSO). Data were analyzed using the BD LRS Fortessa X20 flow cytometer.

## TMRM assay

Proliferating hPBMCs, after 8/7 days of incubation, were stained with antibodies including CD4-PerCP (BD, 550631) and CCR7-FITC (R&D, FAB197F) for 10 min at room temperature in the dark and then with 20 nM TMRM (Thermofisher, 134361) and 2 µM Cyclosporine H for 30 min at 37 °C in HBSS. Stained cells were treated with either 1 µM PAPTP or 100 nM ShK and analyzed with a S3e™ Cell Sorter (BioRad) after 60 s of treatment. The control group received no treatment (DMSO).

## Determination of Kv1.3 expression

Human PBMCs were harvested after 7 days of autoproliferation and incubated with anti-CD25 PECy7 (BD, 557741), anti-CD4 PerCP (BD, 550631), and anti-CCR7 BV421 (BD, 562555) for 10 min at room temperature in the dark. Cells were washed, centrifuged, and incubated with 10 nM ShK-F6CA for 30 min at room temperature in the dark. Finally, cells were washed twice with PBS, fixed with BD Cytofix/Cytoperm Buffer for 10 min at room temperature, centrifuged, and resuspended in PBS for the FACS analysis.

## Mouse studies

Animal experiments were approved by the local ethical committee and the Italian Ministry of Health (660/2020-PR). Animal experiments of the adoptive transfer model were approved by the University of Cincinnati Ethic Committee and the Institutional Animal Care and Use Committee (IRB 10-05-10-01 and IRB 20-07-07-01). All experiments were performed according to the FELASA regulations and we also followed the ARRIVE guidelines.

## EAE Induction and disease scoring

The animals were randomly distributed into three experimental groups at the beginning of the experiment (control, EAE, and EAE + PAPTP). All the mice were housed at room temperature under the same conditions and had standard food and water ad libitum, under a 12:12 h light-dark cycle. EAE was induced in 8–12-week-old C57BL/6 wild-type females by active immunization using myelin oligodendrocyte glycoprotein p35–55 ($MOG_{35-55}$), following established protocols (Bittner et al, 2014). Briefly, mice received subcutaneous injections of 300 µg $MOG_{35-55}$ (Espikem, EPK1) emulsified in 300 µL of 4 mg/mL Complete Freund's adjuvant (CFA) on each hind flank. The CFA was prepared by dissolving desiccated M. Tuberculosis H37Ra (BD, 231141) in Incomplete Freund Adjuvant (BD; 263910) to achieve a final concentration of 4 mg/mL. Subsequently, mice were intraperitoneally administered 400 ng of pertussis toxin (Quadratech Diagnostic, QTXAG-108-50) in 200 µL of PBS on the day of immunization, followed by 200 ng of pertussis toxin in 200 µL of PBS two days later. Animals were scored daily for clinical symptoms of EAE, according to the following scale: 0, no clinical signs; 1, flaccid tail; 2, hind limb weakness; 3, hind limb paresis; 4, complete bilateral hind limb paralysis; 5, death. PAPTP treatment commenced upon the manifestation of the initial symptoms in mice and was subsequently administered every other day until the mice were sacrificed, totaling 11 injections. The treated group received 4 nmol/gbw of PAPTP via intraperitoneal injection. As a control measure, animals in the control group were intraperitoneally injected with DMSO diluted in PBS. Mice were sacrificed by cervical dislocation 4 h after the last PAPTP injection. Blood was collected from the submandibular vein, and mice were perfused via the left ventricle with PBS followed by 4% paraformaldehyde (buffered in PBS, pH 7.4). The brain and the spinal cords were removed and fixed for Transmission Electron Microscopy.

## Evaluation of motor performance

The coordination capability of EAE mice was measured using the Rotarod apparatus (Ugo Basile, Italy). The animals underwent a two-day training regimen one week before immunization. On the first day, they were required to hang onto the static rod for 5 min and run for 2 min at a speed of 5 rpm. On the subsequent day, the mice hung onto the static rod for 2 min and ran for 5 min at the same 5 rpm speed. During the experiment, the initial speed was set at 5 rpm and gradually increased to 40 rpm within a 5-min period. Each group of animals underwent three trials, with a 20-min recovery period between each trial. The 'latency to fall' was recorded for each trial, defined as the time when an animal either fell from the rod or rotated around it three consecutive times without being able to recover and resume running. The final value for the latency to fall in the experiment was calculated as the mean of the three trials.

## Adoptive transfer EAE model

Six- to eight-week-old female donor wild-type mice were immunized with an emulsion of 1 mg/mL myelin-oligodendrocyte glycoprotein peptide amino acids 35–55 ($MOG_{aa35-55}$) (Medchem-Express, HY P1240) in complete Freund's adjuvant (CFA).

Aliquots of the suspension were subcutaneously injected close to the axillary and inguinal lymph nodes and close to the tail base of the mice. Mice were i.p. injected with PTX (Merck, #P7208-5046) immediately after and 2 days as above. Mice were sacrificed 10 days after immunization and spleens were collected. Single cell suspensions were prepared, and erythrocytes were lysed using a RBC cell lysis buffer (BioLegend 420302) for 5 min on ice. Cells were then washed in 6 volumes PBS, resuspended in RPMI-1640 supplemented with 10% FCS, 10 U/mL penicillin-streptomycin, 2 mM L-glutamine, 1% (v/v) non-essential amino acids, 1 mM sodium pyruvate, 10 mM HEPES (pH 7.4). Cells were adjusted to a density of $4 \times 10^6$ cells/mL in this medium and stimulated with 20 µg/mL MOG, 15 ng/mL mIL-12 (R&D, Wiesbaden, Germany), 5 ng/mL mIL-23 (R&D, Wiesbaden, Germany), and 10 µg/mL anti-IFN-γ (clone XMG1.2; BD Biosciences) for 3 days. Aliquots of the cells were treated with 1 µM PAPTP for 3 days. Antigen-specificity of the cells was confirmed by additional proliferation assays of isolated donor lymphocytes. To this end, 2 days after stimulation, a small sample of the cells was fed with 1 mL fresh medium per well in the presence of [³H]Thymidine (10 µCi/mL) and incorporation of [³H]Thymidine was determined after 24 h.

All other cells were collected after 3 days of stimulation. T-lymphocytes were sorted using the pan mouse T cell isolation kit from Miltenyi Biotec resulting in purification of untouched T cells. An aliquot of the T lymphocytes was stained with FITC-coupled anti-L-selectin antibodies (1:500) and APC-coupled anti-CD44 antibodies (1:500) and analyzed on a FACS Calibur. A second aliquot was stained with FITC-Annexin V (1:1000, Roche), Cy5-coupled anti-L-Selectin (CD62L) antibodies (1:500) and APC-anti-CD44 antibodies. Finally, we also determined overall cell death in the samples by Trypan Blue staining and counting dead cells in 500 cells total and by staining the samples with FITC-Annexin V followed by flow cytometry.

The remaining cells were purified by a Ficoll gradient (Merck) to remove dead cells and the $3 \times 10^6$ cells per mouse were injected intraperitoneally into syngeneic recipient wild-type mice. Assessment of clinical disease activity was performed twice daily and scored as follows: 0, healthy; 0.5, limp tail; 1, hind leg paraparesis; 2, hind leg paraplegia; 3, hind leg paraplegia with incontinence.

## CRISPR deletion of Kv1.3

At least 25 million splenocytes were isolated from $MOG_{aa35-55}$ immunized mice, washed in HEPES-buffered saline (H/S; 132 mM NaCl, 20 mM HEPES [pH 7.4], 5 mM KCl, 1 mM $CaCl_2$, 0.7 mM $MgCl_2$, 0.8 mM $MgSO_4$) and resuspended in 400 µL of H/S. Cells were transferred to electroporation cuvettes, and 2 µg of the CRISPR/Cas9 construct targeting Kv1.3 (Santa Cruz Inc., sc-421214) was added per sample. After a 15-min incubation on ice, electroporation was performed using a BTX X2/SC Gemini device (400 V, 5 pulses, 1 ms per pulse). Cells were incubated on ice for an additional 15 min before being transferred to RPMI-1640 medium supplemented with 10% fetal calf serum, 10 mM HEPES, 2 mM L-glutamine, 1 mM sodium pyruvate, 100 µM non-essential amino acids, 100 U/mL penicillin, and 100 µg/mL streptomycin. Cells were cultured for 2 days, and non-viable cells were removed on day 1 via Ficoll gradient centrifugation.

For sorting, cells were collected, washed with H/S, incubated with Fc receptor blocking reagent (BioLegend, #101302; 1:50

dilution) for 15 min at 4 °C, washed, and labeled with biotin-conjugated anti-Kv1.3 antibody (Alomone Labs, #APC-101B) for 30 min at 4 °C. Following a second wash, cells were incubated with streptavidin-conjugated microbeads (Miltenyi Biotec, #130-048-101) for 30 min at 4 °C. Kv1.3-positive and -negative populations were isolated using LS columns (Miltenyi Biotec, 130-042-401), according to the manufacturer's protocol, washed and resuspended in H/S for transfection with 20 µg of EYFP-mito-Kv1.3 (Szabo et al, 2008) using the same electroporation parameters. After an additional day of culture under the same conditions, cells were stimulated with 1 µM PAPTP. Cell death was assessed by staining with APC-conjugated Annexin V (Thermo Fisher, A35110), followed by flow cytometry (FACSCalibur).

## Confocal microscopy

CD4$^+$CD25$^-$ conventional T cells (Tconv) were isolated from the spleen of wild-type mice using CD4$^+$CD25+ Regulatory T cells isolation kit (Miltenyi, 130-091-041), following manufacturer's instructions. $3 \times 10^6$ Tconv cells were stimulated with $1.5 \times 10^6$ CD3/CD28 mouse Dynabeads (Thermofisher, 11452D) for 24 h in RPMI supplemented with 10% FBS and 1% Pen Strep. Cells were resuspended and plated on µ-slide 4 well ibiTreat chambers (ibidi, 80426-90) coated with polylysine. Cells were centrifuged to make them adhere to the chamber and incubated with 50 nM Mitotracker Red CMXRos (Thermofisher, M7512) and 2 µM cyclosporine H in HBSS at 37 °C for 30 min. Cells were washed and incubated with 100 nM PAPTP-Fluorin HBSS for 30 min. Images were acquired using Leica Stellaris 8 and analyzed with Fiji.

## Brain histology of adoptive EAE experiments

Mice were sacrificed 11 days after transfer immunization when they reached the full immune response, the spleen was removed, the mice were perfused via the left heart with 0.9% NaCl followed by 4% paraformaldehyde (buffered in PBS, pH 7.4), the brain was removed, fixed for 2 days in 4% paraformaldehyde, dehydrated in a series of ethanol, embedded in paraffin. Afterwards, brains were sectioned at 6 µm, dewaxed and rehydrated. Sections were then incubated for 30 min with pepsin (Digest All, Invitrogen, Darmstadt, Germany) at 37 °C, washed 3 times in PBS, and blocked for 10 min with PBS, 0.05% Tween 20, and 5% fetal calf serum (FCS). The samples were washed again in PBS, followed by immunostaining with Cy3-anti-CD45 to detect immune cell invasion into the brain. Alternatively, the sections from the same animal brains were incubated for 45 min with FITC-anti-CD44 (1:250 dilution, clone IM7, e-Bioscience #11-0441-82). The samples were then washed 3 times, each 5 min, in PBS supplemented with 0.05% Tween 20, once in PBS and incubated for 45 min with PE-anti-CD62L-antibodies (1:250 dilution, clone MEL14, BD-Pharmingen #561918), washed again and then finally stained with APC-anti-CD4-antibodies (1:250, clone GK1.5, Biolegend #100412). Samples were washed again 3 times, 5 min each, in PBS supplemented with 0.05% Tween 20 and once in PBS and finally embedded in Mowiol. Samples were analyzed on a LEICA TCS SL confocal microscope using a 40x lens (400-fold magnification). Control staining with isotype-matched control antibodies showed very weak or no signals and served as specificity controls.

## Blood analysis

Peripheral blood (about 500 µL) was collected from the sub-mandibular vein of mice in tubes containing 10 µL of 0.5 M EDTA (pH 7.8). 15 µL of blood were diluted at 1:1 ratio with PBS and analyzed using CELL-DYN Emerald (G4-9513/R06, Abbott Laboratories).

## Transmission electron microscopy

Cerebral cortex and lumbar spinal cords specimens from mice were fixed in 2% formaldehyde, 2.5% (v/v) glutaraldehyde in 0.1 M Na-cacodylate (pH 7.4) for 1 h at room temperature and then overnight at 4 °C. Following washing, post-fixation was performed in a 1% OsO$_4$ solution in 100 mM sodium cacodylate, pH 7.2, at 4 °C. Sections were contrasted with a saturated uranyl acetate solution in 100% ethanol for 15 min, followed by incubation in a 1% (w/v) lead citrate solution in 100% ethanol for 7 min. Thin sections were imaged on a Tecnai G2 Spirit transmission electron microscope (FEI Electron Microscopes) operating at 100 kV. For each brain or spinal cord image, the number of myelin sheets per axon was counted. G-ratio was calculated as in (Wang et al, 2022).

## Flow cytometry

For assessing apoptosis in human PBMCs, $4 \times 10^5$ cells underwent incubation with fluorescent-labeled antibodies for 10 min at room temperature in the dark. Following incubation, cells were washed with PBS, centrifuged, and resuspended in 100 µL of Binding Buffer (10 mM HEPES, 140 mM NaCl, and 2.5 mM CaCl$_2$, pH 7.4). Subsequently, 2 µL of APC-coupled Annexin V (Thermo Fisher, A35110) were added to each tube and incubated for an additional 15 min at room temperature in darkness. Post-incubation, 100 µL of binding buffer was added, and the samples were immediately subjected to Flow Cytometry analysis.

In the case of mouse peripheral blood analysis, 30 µL of blood was incubated with fluorescent-labeled antibodies for 10 min at room temperature in the dark. Red blood cells were lysed by incubating the blood with 3 mL of pre-warmed BD Pharm Lyse™ Lysing Buffer (BD, 555899) for 1 min at room temperature until the solution cleared. Subsequently, the samples were centrifuged at $500 \times g$ for 5 min, and the resulting pellet was resuspended in 100 µL of Binding Buffer for further Annexin V staining, following the previously mentioned protocol. For MitoSox, stained peripheral blood was incubated with 5 µM MitoSox in HBSS for 30 min at 37 °C and then treated in vitro with either 1 µM PAPTP or left untreated (DMSO) in HBSS for 30 min at 37 °C.

For spinal cords, tissues were isolated and homogenized with the gentleMACS Dissociator (Miltenyi), using the Adult Brain Dissociation Kit (Miltenyi, 130-107-677), following manufacturer's instructions. Single cells suspensions were then stained using the Mouse Naive/Effector/Memory T Cell Markers Flow Cytometry Panel (Cell Signaling, #78148), with PE-anti-CD3 (clone 17A2, Cell Signaling #28306S), Cy5-anti-CD44 (clone IM7, Cell Signaling #94170S) and FITC-anti-CD62L (clone MEL14, Cell Signaling #76378S), respectively, used at the following dilutions: 1:40, 1:160, and 1:200.

The detailed list of antibodies used for each panel can be found in Table 1. Sample acquisition was performed using BD LRS Fortessa X20

**Table 1. List of panels of antibodies used for flow cytometry analysis of human PBMCs and mouse peripheral blood.**

| Sample | List of antibodies | Catalogue | Dilution |
|---|---|---|---|
| Human PBMCs | CD45RO-PE | Invitrogen, 12-0457-42 | 1:40 |
| | CD25 PE-Cy7 | BD, 557741 | 1:80 |
| | CD4 PerCP | BD, 550631 | 1:40 |
| | CCR7-BV421 | BD, 562555 | 1:80 |
| | Annexin V APC | Thermo Fisher, A35110 | 1:50 |
| Human PBMCs | CD8-PE | BD, 345773 | 1:40 |
| | CD4-PercP | BD, 550631 | 1:40 |
| | CD3-PECy7 | BD, 341111 | 1:80 |
| | CD19-BV421 | BD, 562440 | 1:80 |
| Mouse Peripheral Blood | CD45-APCCy7 | BD, 557659 | 1:200 |
| | CD44-PE | BD, 553134 | 1:200 |
| | CD62L-FITC | BD, 561917 | 1:200 |
| | CD4-PerCP | BD, 553052 | 1:200 |
| | CD25-PECy7 | BD, 552880 | 1:200 |
| | Annexin V APC | Thermo Fisher, A35110 | 1:50 |
| Mouse Peripheral Blood | CD45-APCCy7 | BD, 557659 | 1:200 |
| | CD3 PE-Cy7 | Sony, 1101100 | 1:200 |
| | CD19 BV421 | BD, 562701 | 1:200 |
| Mouse Peripheral Blood | CD11b PE | BD, 557397 | 1:200 |
| | CD3 PE-Cy7 | BD, 552880 | 1:200 |
| | CD19 FITC | BD, 553785 | 1:200 |
| | CD80 PerCp-Cy5 | BioLegend, 104722 | 1:200 |
| Mouse Peripheral Blood | CD4-PerCP | BD, 553052 | 1:200 |
| | CD62L-BV421 | BD, 562910 | 1:200 |
| | MitoSox | Thermofisher, M36008 | 5 μM |

or with a FACS Calibur for spinal cords, and subsequent data analysis was carried out using FlowJo Software (BD).

## Histological analysis

Mice were transcardially perfused with 4% paraformaldehyde (PFA), and tissues were subsequently harvested. The spinal cords were post-fixed in 4% PFA for 24 h at 4 °C, followed by cryoprotection in 20% sucrose in PBS for 24 h, and then in 30% sucrose in PBS for an additional 48 h at 4 °C. Spinal cords were then embedded in OCT compound and cryosectioned at a thickness of 10 μm. Sections were stained with either Luxol Fast Blue (LFB) (Bio-Optica, 04-200812) or Bielschowsky stain (Bio-Optica, 04-040805), following the manufacturer's protocols. Images were acquired using a Leica DM6 B microscope. For each animal, at least six sections from the lumbar and the lumbar/thoracic regions of the spinal cord were analyzed. Quantification was performed using ImageJ software. The LFB kit employed here uses the Klüver-Barrera dual staining method (LFB + cresyl violet) and detects myelin in turquoise blue, neurons and glial nuclei in violet and Nissl substance in pale pink. The cresyl-violet stained nuclei in the white matter derive mainly from

the immune cell infiltration. For KB LFB staining, the percentages of inflammatory/demyelinated areas in the total white matter areas were calculated as in (Miyauchi et al, 2020). For quantification of the Blieschowsky staining, we used the ImageJ software to identify the percentage of areas without silver accumulation (Fig. 4D and Fig. EV2F left panel) or performed the procedure according to (Theotokis et al, 2022) in Fig. EV2F, right panel: Axonal loss was assessed on ten randomly selected silver-stained spinal cord sections within $120 \times 240$ μm areas along the spinal cord and graded by two independent evaluators in a blinded manner: 0 = normal/even silver stain throughout the white matter compared to unimmunized mice; 1 = small spurious areas in the white matter that lack silver stain; 2 = small, but frequent, areas in the white matter that lack silver stain; and 3 = extensive loss of silver stain throughout the white matter. For histological analysis of spleens and livers, tissues were isolated and fixed in Formalin solution (Merk, HT501128-4L) for 24 h. The samples were then kept in 70% ethanol and stored at 4 °C. Before paraffin embedding, the tissues were dehydrated through a series of graded alcohols and finally embedded in molten paraffin wax. Paraffin sections, 4–5-μm thick, were stained with hematoxylin and eosin (Bio-Optica, 04-061010) according to standard procedures. Images were captured and exported using a Leica DM4000 microscope.

## Brain Immunohistochemistry and Microglia skeleton analysis of EAE model

For mouse brain staining, paraffin-embedded tissues were cut into 5-μm-thick sections. Samples were deparaffinized by washing twice in xylene for 10 min at room temperature, followed by rehydration through a graded ethanol series. Antigen retrieval was performed immersing tissues in 1 mM sodium citrate (pH 6.0) for 20 min, heating using a microwave avoiding boiling, and washing in PBS. For spinal cords, OCT embedded tissues were cut into 10-μm-thick sections and then fixed in 4% PFA for 15 min. For both brain and spinal cords sections, permeabilization was performed using 0.2% Triton X-100 in PBS for 15 min. Subsequently, samples were treated with Peroxide Block (Mouse to Mouse HRP (DAB) Staining System, Histo-Line Laboratories, MTM001) for 10 min, washed three times in PBS, and treated with Super Block (Histo-Line Laboratories, MTM001) for 5 min, followed by three PBS washes. Tissues were incubated overnight at 4 °C in a humidified chamber with anti-Iba-1 (1:250, Invitrogen, PA5-27436) or anti-GFAP (1:250, Dako, Z0334). Samples were then washed three times with PBS at room temperature, and incubated with a secondary goat anti-rabbit IgG (H + L) (HRP polymer) antibody (AB214880, Abcam) for 2 h at room temperature in a dark, humidified chamber. After three more PBS washes, sections were incubated with DAB Substrate (AB64238, Abcam) for 50 s at room temperature, followed by three 5 min washes in PBS. Samples were then dehydrated through an increasing ethanol series and washed twice in xylene for 5 min. Finally, samples were mounted using Eukitt Quick-hardening mounting medium (03989, Sigma-Aldrich). Images were imaged using a Leica DM6 B microscope in bright-field. Skeleton analysis was performed as described by Young and Morrison using ImageJ (Young and Morrison, 2018). For Iba-1, cell counting was conducted using Ilastik, a machine learning-based software, according to the workflow provided by the developers (https://www.ilastik.org/documentation/counting/counting). For GFAP, cell counting was performed using ImageJ.

## Synthesis of fluorescent PAPTP

All reagents and solvents were purchased from Sigma-Aldrich (Milan, Italy) and used without further purification. Flash column chromatography was performed on silica gel (Macherey-Nagel 60, 230–400 mesh, particle size 0.040–0.063 mm) under nitrogen pressure.

$^1$H and $^{13}$C NMR spectra were acquired on a Bruker Avance III HD spectrometer operating at 400 MHz for $^1$H and 101 MHz for $^{13}$C. Chemical shifts ($\delta$) are reported in parts per million (ppm) relative to the residual solvent peak. The following abbreviations denote signal multiplicities: s = singlet, d = doublet, t = triplet, m = multiplet.

Electrospray ionization mass spectrometry (ESI-MS) was performed on an Agilent 1100 Series system in full-scan positive ion mode with the following parameters: nebulizer pressure, 20 psi; drying gas flow, 5 L/min; drying gas temperature, 32 °C; and flow rate, 0.05 mL/min.

Preparative reversed-phase high-performance liquid chromatography (RP-HPLC) was carried out using an Agilent 1290 Prep LC system equipped with dual LC8A pumps and a Prominence SPD-20A UV-Vis detector set to 254 nm. Separation was achieved using a Zorbax Eclipse PrepHT XDB-C18 column (5 μm, 250 × 21.2 mm; Agilent).

Analytical UPLC for purity assessment was performed using an Agilent InfinityLab LC 1260 Series system equipped with a Zorbax Eclipse XDB-C18 column (1.8 μm, 2.1 × 5 mm; Agilent). The mobile phases consisted of water with 0.1% trifluoroacetic acid (TFA) and acetonitrile with 0.1% TFA. The steps of synthesis are shown in Fig. EV2A and are described in the legend of this figure.

## Statistical analysis

All statistical analyses were performed with GraphPad Prism 8 software. Three or more groups were analyzed with one-way ANOVA and Dunnett's posttest. Two-way ANOVA with Bonferroni's post-test was used to compare data with two variables. Unpaired, Student's T-test was used to compare two groups. Additional statistical details can be found in the figure legends. Where individual p-values are not shown for comparisons in the figure, the p-values were >0.05 and therefore differences were not statistically significant.

The sample size planning was based on the results of two-sided Wilcoxon-Mann-Whitney tests. Investigators were blinded to the samples in microscopic studies and to animal identity. Animals were randomly assigned to cages by a technician who was not involved in the experiments, thus the mice were purely randomly assigned for every experiment. Cages were then randomly assigned to the various experimental groups.

## Data availability

All data are available in the main text or the expanded view material. No data presented in the current manuscript requires deposition in a public database.

The source data of this paper are collected in the following database record: biostudies:S-SCDT-10_1038-S44321-025-00307-2.

## Peer review information

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

## Acknowledgements

The authors are very grateful to Professors Roland Martin, Antonio Felipe, and Mario Zoratti for critical reading of the manuscript and Professor Gabriela Constantin and Dr. Gabriele Angelini for useful discussion. The authors also thank Dr. Filippo Severin and Veronica Trimarco for initial help with setting up the panels for FACS analysis and Dr. Valentina Brillo for help with the confocal microscopy. IS would like to dedicate this work to Dr. Ibolya Schmehl. The authors are also grateful to Anson Scientific for language editing. The authors thank the Italian Association of Multiple Sclerosis (FISM grant 2022/R single/ 046 to IS) and DFG (grant GU 335/35-2 to EG) for financial support. BA thanks the Italian Association for Cancer Research for the AIRC fellowship (29677) and TV thanks the Veronesi Foundation for a researcher contract. The authors are grateful for support from the AIRC IG Grant IG 20286 to IS. This study received support also from the MUR-PNRR Progetto finanziato dall'Unione Europea - NextGenerationEU - MUR - Mission 4 "Istruzione e Ricerca" del Piano Nazionale di Ripresa e Resilienza, Componente C2 - Misura 1.4 - National Center for Gene Therapy and Drugs based on RNA Technology - CN00000041 - CUP C93C22002780006 (IS) and from Michael J. Fox Fondation (MJFF-021303 grant to IS).

## Author contributions

**Beatrice Angi**: Conceptualization; Investigation; Visualization; Methodology; Writing—original draft; Writing—review and editing. **Tatiana Varanita**: Conceptualization; Supervision; Investigation; Visualization; Methodology; Writing—original draft; Writing—review and editing. **Marco Puthenparampil**: Conceptualization; Investigation; Methodology; Writing—original draft; Project administration; Writing—review and editing. **Valentina Scattolini**: Conceptualization; Investigation; Visualization; Methodology; Writing—original draft; Writing—review and editing. **Michael Donadon**: Investigation; Methodology; Writing—original draft. **Mitra Tavakoli**: Investigation; Methodology. **Marta Favero**: Investigation; Methodology. **Maguie El Boustani**: Investigation; Visualization; Methodology; Writing—original draft. **Matthias Soddemann**: Investigation; Methodology. **Lucia Biasutto**: Investigation; Methodology. **Diletta Arcidiacono**: Methodology. **Alberto Ongaro**: Methodology. **Andrea Mattarei**: Supervision; Methodology; Writing—original draft. **Livio Trentin**: Supervision. **Gregory Wilson**: Visualization. **Erich Gulbins**: Conceptualization; Funding acquisition; Investigation; Visualization; Writing—original draft; Project administration; Writing—review and editing. **Paolo Gallo**: Conceptualization; Supervision; Project administration; Writing—review and editing. **Ildiko Szabo**: Conceptualization; Supervision; Funding acquisition; Writing—original draft; Project administration; Writing—review and editing.

Source data underlying figure panels in this paper may have individual authorship assigned. Where available, figure panel/source data authorship is listed in the following database record: biostudies:S-SCDT-10_1038-S44321-025-00307-2.

## Disclosure and competing interests statement

The authors declare no competing interests.

# Expanded View Figures

**Figure EV1.  Analysis of the effects of PAPTP on PBMCs from MS patients.**

(A) Representative quantitative results of the Mean Fluorescence Intensity of MitoSox in both $CD4^+$ $CD25^+$ $CCR7^+$ (Naive T cells + $T_{CM}$s) and $CD4^+$ $CD25^+$ $CCR7^-$ ($T_{EM}$ + $T_{EMRA}$) lymphocytes either untreated or treated with 1 μM PAPTP. (B) Representative quantitative results displaying the Mean Fluorescence Intensity of ShK-F6CA in $CD4^+$ $CD25^+$ and $CD4^+$ $CD25^-$ lymphocytes in stained hPBMCs (left) and unstained hPBMCs (right). (C) Gating Strategy for Apoptosis detection in Naive T cells, $T_{EM}$, and $T_{CM}$ cells from hPBMCs. (D) Percentage of $CCR7^-$ $CD45RO^+$ $T_{EM}$ cells in hPBMCs from MS patients treated with indicated PAPTP concentrations. Cells were gated within $CD4^+$ $CD25^+$ $CFSE^-$ autoproliferative lymphocytes ($n = 10$ for untreated and 5 μM treated groups; $n = 8$ for 1 μM PAPTP group). (E) Normalized apoptotic levels of $CCR7^-$ $CD45RO^+$ $T_{EM}$ cells at specified PAPTP concentrations. For each patient, data were normalized based on the untreated sample ($n = 10$ for untreated and 5 μM treated groups; $n = 8$ for 1 μM PAPTP group). (F) Percentage of $CCR7^-$ $CD45RO^-$ $T_{EMRA}$ cells in hPBMCs from MS patients treated with indicated PAPTP concentrations. Cells were gated within $CD4^+$ $CD25^+$ $CFSE^-$ autoproliferative lymphocytes ($n = 9$ for untreated and 5 μM treated groups; $n = 8$ for 1 μM PAPTP group). (G) Normalized apoptotic levels of $CCR7^-$ $CD45RO^-$ $T_{EMRA}$ cells at specified PAPTP concentrations. For each patient, data were normalized based on the untreated sample ($n = 10$ for untreated and 5 μM treated groups; $n = 8$ for 1 μM PAPTP group). (H) Normalized apoptotic levels of $CCR7^-$ effector cells ($T_{EM}$ + $T_{EMRA}$) at specified PAPTP concentrations. For each patient, data were normalized based on the untreated sample ($n = 10$ for untreated and 5 μM treated groups; $n = 8$ for 1 μM PAPTP group). (I) Percentage of $CCR7^+$ $CD45RO^-$ naive T cells in hPBMCs from MS patients treated with indicated PAPTP concentrations. Cells were gated within $CD4^+$ $CD25^+$ $CFSE^-$ autoproliferative lymphocytes ($n = 7$ for untreated and 5 μM PAPTP groups; $n = 5$ for 1 μM PAPTP). (J) Normalized apoptotic levels of $CCR7^+$ $CD45RO^-$ naive T cells at specified PAPTP concentrations. For each patient, data were normalized based on the untreated sample ($n = 7$ for untreated and 5 μM PAPTP groups; $n = 5$ for 1 μM PAPTP). (K) Percentage of $CCR7^+$ $CD45RO^+$ $T_{CM}$ cells in hPBMCs from MS patients treated with indicated PAPTP concentrations. Cells were gated within $CD4^+$ $CD25^+$ $CFSE^-$ autoproliferative lymphocytes ($n = 8$ for untreated and 5 μM PAPTP groups; $n = 6$ for 1 μM PAPTP). (L) Normalized apoptotic levels of $CCR7^+$ $CD45RO^+$ $T_{CM}$ cells at specified PAPTP concentrations. For each patient, data were normalized based on the untreated sample ($n = 8$ for untreated and 5 μM PAPTP groups; $n = 6$ for 1 μM PAPTP). Two outliers were removed from the graph. (M) Percentages of apoptotic cells of the indicated $CD4^+$ $CD25^+$ subpopulations obtained from healthy subjects PBMC ($n = 5$ for each indicated condition). (N) Gating strategy for the identification of $CCR7^+$ and $CCR7^-$ cells for MitoSOX analysis. Mitosox Mean Fluorescence Intensity was evaluated on $CCR7^+$ and $CCR7^-$ populations. The same gating strategy was used for the analysis of Kv1.3 expression, using ShK-F6CA. The ShK-F6CA Mean Fluorescence Intensity of $CD4^+$ $CD25^+$ ShK-F6CA and $CD4^+$ $CD25^-$ cells was evaluated. (D–M) Data represent average ± SEM with superimposed individual data points for each patient. Each data point represents hPBMCs derived from a distinct patient. (D) p-values of one-way ANOVA test. (E, G, H, J, L) p-values of Wilcoxon test. (N) p-value in Friedman test. Where values are not indicated, no significant difference was observed.

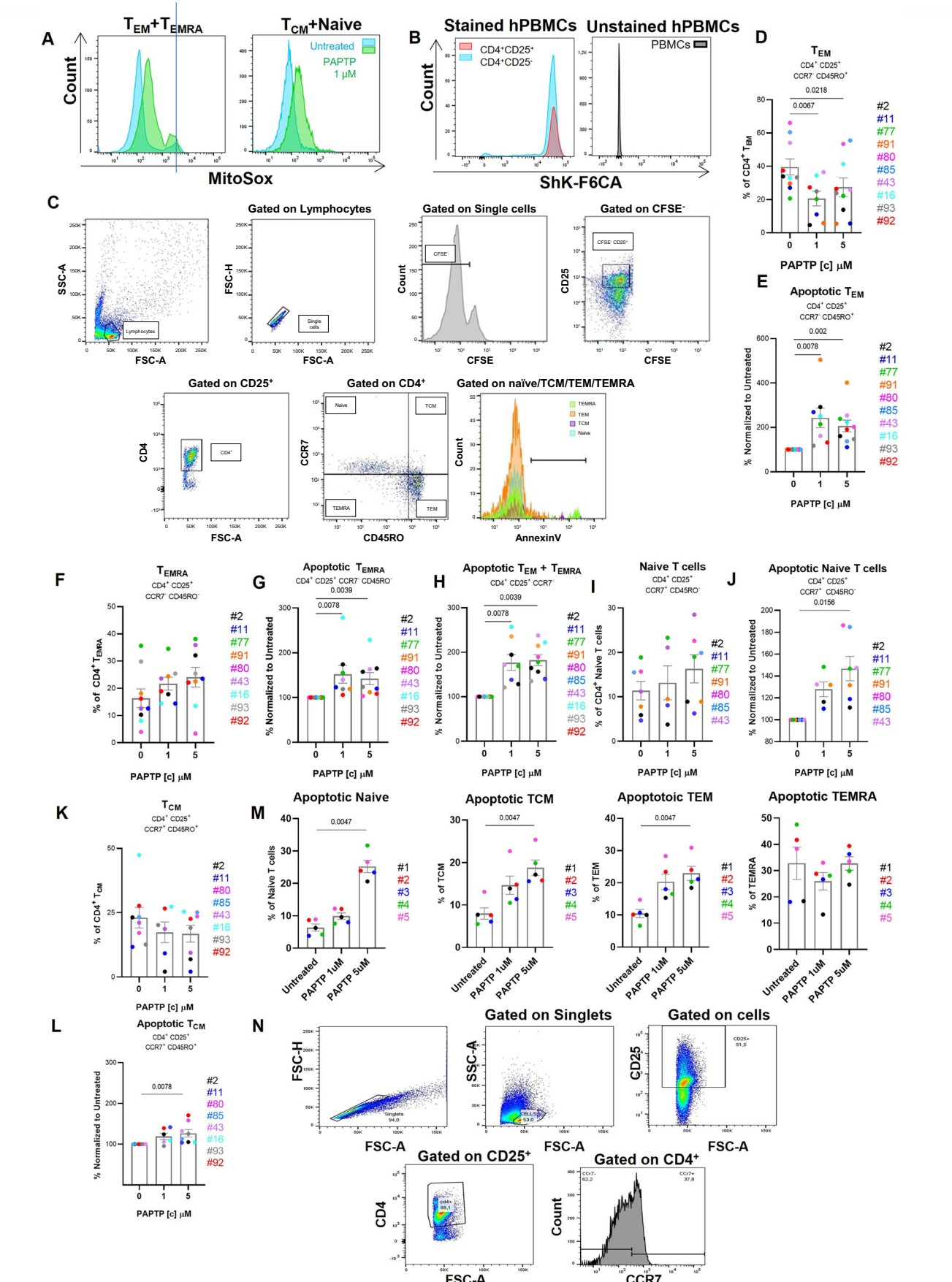

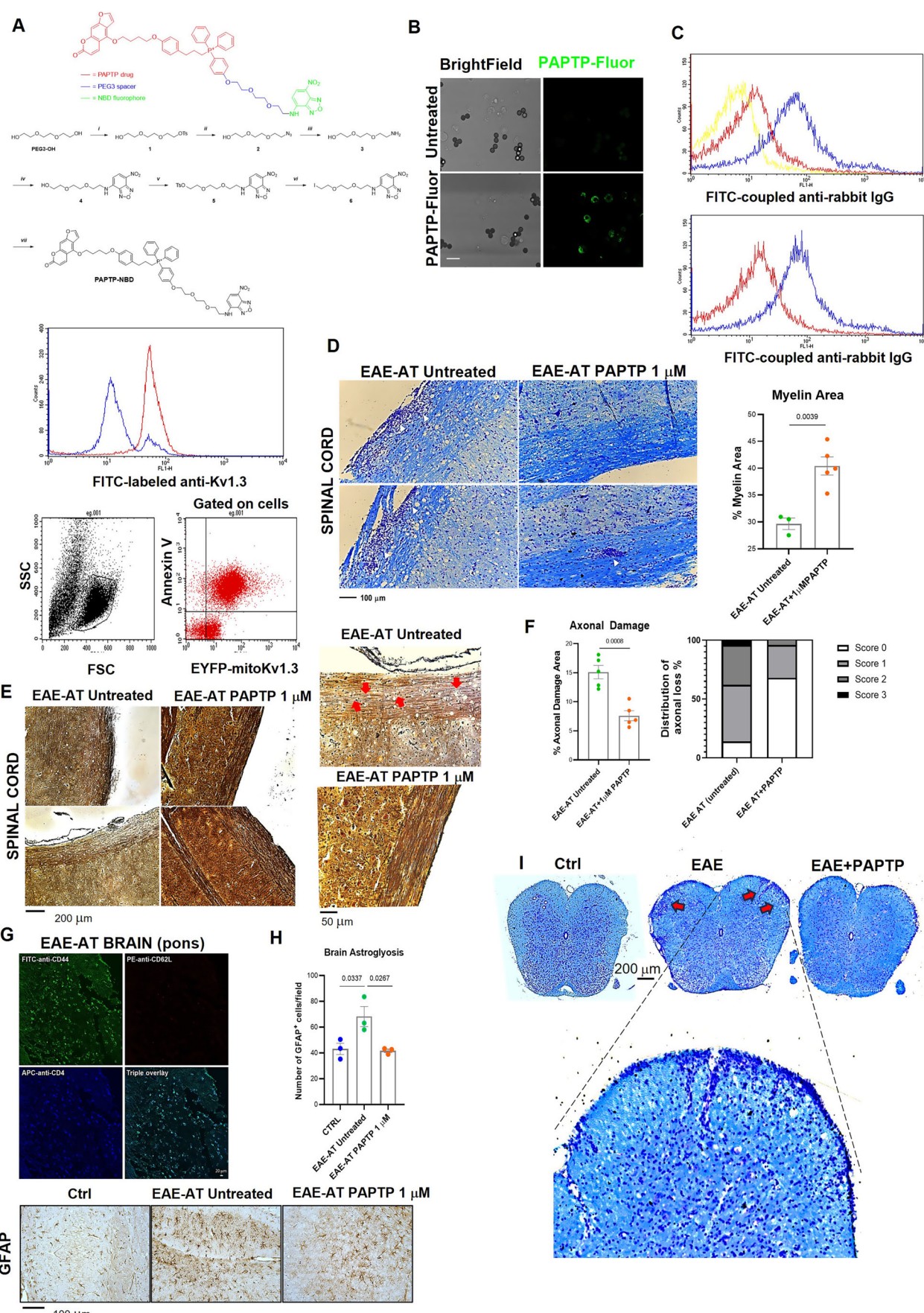

**Figure EV2. PAPTP prevents EAE onset in the Adoptive Transfer Model.**

(A) Upper panel: Chemical Structure of 7-nitrobenz-2-oxa-1,3-diazole (NBD)-labeled PAPTP (PAPTP-NBD). Lower panel: Synthesis of PAPTP-NBD. Reagents and Conditions (i) TsCl, DMAP, pyridine, DCM, r.t., 18 h; (ii) NaN₃, DMF, 90 °C, 3 h; (iii) Pd/C, H2, MeOH, r.t., 16 h; (iv) NBD-Cl, DIPEA, MeOH, r.t., 16 h; (v) TsCl, DMAP, pyridine, DCM, r.t., 5 h; (vi) NaI, acetone, 40 °C, 16 h; (vii) PAPTP-OH, K2CO₃, DMF, r.t., 16 h. Synthesis of 2-(2-(2-hydroxyethoxy)ethoxy)ethyl 4-methylbenzenesulfonate (1). To a solution of triethylene glycol (PEG3-OH, 77.28 g, 514.6 mmol, 8.0 equiv) in dichloromethane (DCM, 255 mL) at 0 °C were added 4-dimethylaminopyridine (DMAP, 15.7 g, 128.6 mmol, 2.0 equiv) and pyridine (10.18 g, 128.6 mmol, 2.0 equiv). After 10 min of stirring, p-toluenesulfonyl chloride (TsCl, 12.3 g, 64.3 mmol, 1.0 equiv), previously dissolved in DCM (165 mL), was added dropwise. The reaction mixture was stirred at room temperature for 18 h. The reaction was then quenched by dilution with 300 mL of 1.0 M HCl, the organic layer separated and the aqueous phase was extracted with DCM (2 × 250 mL). The combined organic layers were dried over anhydrous Na₂SO₄, filtered, and concentrated under reduced pressure. The crude product was purified by flash column chromatography on silica gel using a DCM/acetone mixture (8:2) to afford 1 as a pale yellow oil (13.2 g, 43.4 mmol, yield: 67%). 1H NMR (400 MHz, CDCl3) δ 7.80 (d, $J = 8.4$ Hz, 2H), 7.34 (d, $J = 8.4$ Hz, 2H), 4.18–4.15 (m, 2H), 3.72–3.69 (m, 4H), 3.61 (s, 4H), 3.58–3.56 (m, 2H), 2.44 (s, 3H).13 C NMR (101 MHz, CDCl3) δ 145.01, 133.07, 129.98, 128.11, 72.59, 70.92, 70.43, 69.29, 68.85, 61.89, 21.78. ESI-MS (ion trap): $m/z$ 305 [M + H]+. Synthesis of 2-(2-(2-azidoethoxy)ethoxy)ethan-1-ol (2). To a solution of compound 1 (13.2 g, 43.4 mmol, 1.0 equiv) in anhydrous N,N-dimethylformamide (DMF, 100 mL), sodium azide (NaN₃, 8.5 g, 130.2 mmol, 3.0 equiv) was added. The reaction mixture was stirred at 90 °C for 3 h until thin-layer chromatography (TLC) analysis (EtOAc/PE, 6:4) indicated complete consumption of the starting material. The reaction mixture was then diluted with ethyl acetate (EtOAc, 300 mL) and washed with brine/water 1:1 (5 × 100 mL). The organic layer was dried, and the solvent was removed under reduced pressure. The flask was left under high vacuum overnight to remove residual DMF, yielding compound 2 as a pale yellow oil (6.8 g, 38.7 mmol, yield: 89%). 1H NMR (400 MHz, CDCl3) δ 3.73–3.71 (m, 2H), 3.68–3.64 (m, 6H), 3.61–3.59 (m, 2H), 3.38 (t, $J = 5.0$ Hz, 2H), 2.45 (s, 1H).13 C NMR (101 MHz, CDCl3) δ 72.59, 70.74, 70.48, 70.13, 61.84, 50.74. ESI-MS (ion trap): $m/z$ 176 [M + H]+. Synthesis of 2-(2-(2-aminoethoxy)ethoxy)ethan-1-ol (3). Palladium on carbon (Pd/C, 10% w/w, 0.70 g) was suspended in methanol (30 mL) in a round-bottom flask under nitrogen atmosphere. Compound 2 (6.8 g, 38.7 mmol, 1.0 equiv), previously dissolved in methanol (10 mL), was added to the suspension. The reaction atmosphere was then replaced with hydrogen, and the mixture was stirred at room temperature for 16 h, until thin-layer chromatography (TLC) analysis (DCM/ acetone, 8:2) confirmed complete consumption of the starting material. Hydrogen was removed by nitrogen stream and the reaction mixture was filtered through a celite pad to remove the catalyst. The solvent was evaporated under reduced pressure and the crude product was purified by flash column chromatography on silica gel using a DCM/MeOH/NH3(aq) (87:12:1) as the eluent, affording compound 3 as a colorless oil (2.6 g, 17.4 mmol, yield: 45%). 1H NMR (400 MHz, CDCl3) δ 3.73–3.70 (m, 2H), 3.68–3.62 (m, 4H), 3.61–3.58 (m, 2H), 3.56–3.53 (m, 2H), 2.88 (t, $J = 5.1$ Hz, 2H), 2.55 (s, 3H).13 C NMR (101 MHz, CDCl3) δ 72.81, 72.79, 70.47, 70.24, 61.63, 41.53. ESI-MS (ion trap): $m/z$ 150 [M + H]+. Synthesis of 2-(2-(2-((7-nitrobenzo[c][1,2,5]oxadiazol-4-yl)amino)ethoxy)ethoxy)ethan-1-ol (4). To a solution of NBD-Cl (1.0 g, 5.0 mmol, 1.0 equiv) in methanol (25 mL) at 0 °C were added N,N-diisopropylethylamine (DIPEA, 2.6 g, 20 mmol, 4.0 equiv) and compound 3 (0.82 g, 5.5 mmol, 1.1 equiv). The reaction mixture was allowed to warm to room temperature and stirred for 16 h. The mixture was then diluted with ethyl acetate (150 mL) and washed with saturated NH4Cl solution (3 × 50 mL). The organic layers were combined, dried, and concentrated under reduced pressure. The crude product was purified by flash column chromatography on silica gel using an EtOAc/MeOH mixture (99:1) as the eluent, affording compound 4 as a brown powder (1.0 g, 3.2 mmol, yield: 64%). 1H NMR (400 MHz, MeOD) δ 8.51 (d, $J = 8.9$ Hz, 1H), 6.43 (d, $J = 8.9$ Hz, 1H), 3.84–3.80 (m, 2H), 3.76 (s, 2H, broad signal), 3.71–3.66 (m, 2H), 3.66–3.62 (m, 4H), 3.56–3.53 (m, 2H).13 C NMR (101 MHz, DMSO) δ 154.82, 153.90, 153.60, 147.35, 130.29, 109.01, 81.83, 79.36, 79.22, 77.44, 69.66, 52.88. ESI-MS (ion trap): $m/z$ 313 [M + H]+. Synthesis of 2-(2-(2-((7-nitrobenzo[c][1,2,5]oxadiazol-4-yl)amino)ethoxy)ethoxy)ethyl 4-methylbenzenesulfonate (5). To a solution of compound 4 (100 mg, 0.32 mmol, 1.0 equiv) in dichloromethane (DCM, 1.5 mL) at 0 °C were added pyridine (76 mg, 0.96 mmol, 3.0 equiv), 4-dimethylaminopyridine (DMAP, 78 mg, 0.64 mmol, 2.0 equiv), and p-toluenesulfonyl chloride (TsCl, 122 mg, 0.64 mmol, 2.0 equiv). The reaction mixture was stirred at room temperature for 5 h. The mixture was then diluted with brine (50 mL) and extracted with DCM (3 × 50 mL). The combined organic layers were dried over anhydrous Na₂SO₄ and concentrated under reduced pressure. The crude product was purified by flash column chromatography on silica gel using DCM/MeOH (99:1) as the eluent, affording compound 5 as a brown solid (94 mg, 0.20 mmol, yield: 63%). 1H NMR (400 MHz, (CD3)2CO) δ 8.50 (d, $J = 8.8$ Hz, 1H), 7.77 (d, $J = 8.0$ Hz, 2H), 7.44 (d, $J = 8.0$ Hz, 2H), 6.49 (d, $J = 8.8$ Hz, 1H), 4.19–4.11 (m, 2H), 3.84–3.80 (m, 4H), 3.71–3.66 (m, 2H), 3.64 – 3.60 (m, 2H), 3.60–3.53 (m, 2H), 2.86 (s, 1H), 2.42 (s, 3H).13 C NMR (101 MHz, (CD3)2CO) δ 145.78, 145.45, 145.08, 137.77, 134.22, 130.77, 128.63, 123.50, 99.90, 71.24, 71.07, 70.61, 69.34, 44.68, 21.49. ESI-MS (ion trap): $m/z$ 467 [M + H]+. Synthesis of N-(2-(2-(2-iodoethoxy)ethoxy)ethyl)-7-nitrobenzo[c][1,2,5]oxadiazol-4-amine (6). To a solution of compound 5 (30 mg, 0.064 mmol, 1.0 equiv) in acetone (0.8 mL) was added sodium iodide (NaI, 39 mg, 0.257 mmol, 4.0 equiv). The reaction mixture was stirred at 40 °C for 16 h in a sealed vial. After completion, the mixture was diluted with ethyl acetate (EtOAc, 40 mL) and washed with brine (3 × 10 mL). The organic layer was dried over anhydrous Na₂SO₄, and concentrated under reduced pressure. The crude product was purified by flash column chromatography on silica gel using EtOAc/petroleum ether (1:1) as the eluent, affording compound 6 as a brown solid (14.5 mg, 0.034 mmol, yield: 54%). 1H NMR (400 MHz, (CD3)2CO) δ 8.54 (d, $J = 8.8$ Hz, 1H), 8.13 (s, 1H, broad signal), 6.55 (d, $J = 8.8$ Hz, 1H), 3.93–3.79 (m, 4H), 3.73–3.58 (m, 6H), 3.29 (t, $J = 6.5$ Hz, 2H).13 C NMR (101 MHz, (CD3)2CO) δ 145.81, 145.41, 145.02, 137.67, 123.43, 99.98, 72.35, 71.08, 70.67, 69.32, 44.59, 4.26. ESI-MS (ion trap): $m/z$ 423 [M + H]+ Synthesis of (4-(2-(2-(2-((7-nitrobenzo[c][1,2,5]oxadiazol-4-yl)amino)ethoxy)ethoxy)ethoxy)phenyl) (3-(4-(4-((7-oxo-7H-furo[3,2-g]chromen-4-yl)oxy)butoxy)phenyl) propyl)diphenylphosphonium (PAPTP-NBD) A solution of compound 6 (14.5 mg, 0.034 mmol, 1.1 equiv) in DMF (0.5 mL) was cooled to 0 °C, and PAPTP-OH (25 mg, 0.031 mmol, 1.0 equiv, synthesized as previously reported, 10.3390/ph14020129) and potassium carbonate (K2CO₃, 4 mg, 0.031 mmol, 1.0 equiv) were added. The reaction mixture was stirred at room temperature for 16 h. After completion, the reaction was diluted with EtOAc (30 mL) and washed with 0.5 M HCl (2 × 10 mL) followed by brine (1 × 10 mL). The combined organic layers were dried over anhydrous Na₂SO₄ and concentrated under reduced pressure. The crude product was purified by preparative HPLC and lyophilized to afford PAPTP-NBD as an orange solid (9.6 mg, 0.009 mmol, yield: 29%, purity UPLC > 95%). 1H NMR (400 MHz, (CD3)2CO) δ 8.42 (s, 1H, broad signal), 8.21 (dd, $J = 9.8, 0.7$ Hz, 1H), 7.93–7.68 (m, 13H), 7.26 (dd, $J = 2.4, 1.0$ Hz, 1H), 7.22 (dd, $J = 9.0, 2.6$ Hz, 2H), 7.16–7.08 (m, 3H), 6.82 (d, $J = 8.6$ Hz, 2H), 6.55 (s, 1H, broad signal), 6.19 (d, $J = 9.8$ Hz, 1H), 4.67 (t, $J = 5.8$ Hz, 2H), 4.24–4.21 (m, 2H), 4.09 (t, $J = 5.8$ Hz, 2H), 3.90–3.82 (m, 6H), 3.68 (s, 4H), 3.60–3.48 (m, 2H), 2.84 (t, $J = 5.8$ Hz, 2H), 2.12–1.98 (m, 6H). 13 C NMR (101 MHz, (CD3)2CO) δ 165.18, 159.24, 158.66, 153.86, 150.18, 146.37, 140.10, 136.81 (d, $J = 11.5$ Hz), 135.87 (d, $J = 3.0$ Hz), 134.61 (d, $J = 10.0$ Hz), 133.12, 131.25 (d, $J = 12.6$ Hz), 130.52, 120.83, 119.97, 117.48, 117.35, 115.48, 113.33, 109.39, 108.46, 107.37, 106.47, 94.02, 73.61, 71.46, 71.36, 70.02, 69.73, 69.23, 68.23, 49.07, 36.03 (d, $J = 16.9$ Hz), 27.59, 26.72, 25.49 (d, $J = 3.7$ Hz), 22.25 (d, $J = 52.6$ Hz). ESI-MS (ion trap): $m/z$ 363 [M]+. (B) Confocal microscopy images showing the fluorescent signal in untreated cells and in those treated with fluorescent PAPTP (PAPTP-fluor) in mitochondria of CD4+CD25- Tconv cells isolated from healthy mice before and after treatment with 100 nM PAPTP-Fluor for 30 min. The scale bar is 10 µm. Control experiment for Fig. 3A. The same cells shown in this representative image are also shown in Fig. 3A at higher magnification. (C) Left panels: Upper panel: Downregulation of Kv1.3 in cells treated with CRIPSR/Cas9 and Kv1.3 staining in sorted cells. Yellow: CRISPR/Cas transfected, unstained. Blue: CRISPR/Cas transfected, cells positively sorted for Kv1.3. This is the fraction, which was positively sorted and then stained with FITC-anti-Kv1.3 antibodies.For sorting, cells were collected, washed with H/S, incubated with Fc receptor blocking reagent (BioLegend, #101302; 1:50 dilution) for 15 min at 4 °C, washed, and labeled with biotin-conjugated anti-Kv1.3 antibody (Alomone Labs, #APC-101B) for 30 min at 4 °C. Following a second wash, cells were incubated with streptavidin-conjugated microbeads (Miltenyi Biotec, #130-048-101) for 30 min at 4 °C. Kv1.3-positive and -negative populations were isolated using LS columns (Miltenyi Biotec, 130-042-401). For the flow cytometry, Fc-receptors were blocked with True stain (1:50 dilution, cells were collected, washed with H/S, incubated with Fc receptor blocking reagent (Clone S17011E, BioLegend, #156604) and then an aliquot of the samples was stained with a FITC-coupled anti-rabbit IgG (1:500, Jackson Immunoresearch 711-096-152) to detect the anti-Kv1.3, which was already bound to the cells. Red: CRISPR/Cas transfected, cells negatively sorted for Kv1.3. This is the fraction, which was negatively sorted and then stained exactly as the blue fraction. The staining was done on aliquots just before retransfection of the mito-Kv1.3 construct to confirm downregulation. Lower panel: Same as above, but with aliquots that were fixed for 10 min at room temperature in 1% buffered PFA, then washed and permeabilized for 8 min with 0.1% Triton X-100 at room temperature, washed again and then stained. This confirms downregulation of extra- and intracellular Kv1.3. Right upper panel: Flow cytometry of sorted Kv1.3-negative cells that were re-transfected. Blue: Control-transfected (empty vector). Aliquots were fixed for 10 min at room temperature in 1% buffered PFA, then washed and permeabilized for 8 min with 0.1% Triton X-100 at room temperature, washed again, Fc-receptors

were blocked with True stain (1:50, Biolegend, #156604) and then stained with FITC-coupled anti-Kv1.3 (Alamone, #APC-101-F). Red: EYFP-Mito-Kv1.3 transfected cells, as above. Right lower panel: Representative dot plot showing the gating strategy for the identification of EYFP-mitoKv1.3$^+$ Annexin$^+$ and EYFP-mitoKv1.3$^-$ Annexin$^+$ cells after PAPTP treatment. Kv1.3$^-$ cells were transfected with EYFP-mitoKv1.3 construct and subsequently treated with PAPTP for 48 h. Cells were then analyzed for apoptosis using flow cytometry. (**D**) Representative images of longitudinal spinal-cord sections stained with luxol fast blue from mice receiving MOG$_{35-55}$ activated untreated or 1 μM PAPTP-treated lymphocytes. Scale bar, 100 μm. Demyelinated are indicated with white arrows. Right: Average ± SEM of myelin area per field ($n = 3$ for mice receiving untreated lymphocytes, $n = 5$ for mice receiving 1 μM PAPTP-treated lymphocytes). *p*-value of Student's t test. (**E**) Representative images of longitudinal spinal-cord sections stained with Bielschowsky staining from mice receiving MOG$_{35-55}$ activated untreated or 1 μM PAPTP-treated lymphocytes. Scale bar, 200 μm. Right upper panel: enlarged image from the lower EAE AT untreated sample shown on the left. Right lower panel: enlarged image from the upper EAE AT + PAPTP sample shown on the left. Please note damaged axon fibers in the EAE. (**F**) Quantification of axon damage from longitudinal sections of Blieschowsky-stained spinal cords from EAE AT and EAE AT + PAPTP animals. Quantification was performed following the method used for Fig. 4D (left panel) and according to (Theotokis et al, 2022). (**G**) Representative images of a brain section of a wild-type mouse injected with MOG$_{35-55}$ activated untreated lymphocytes, stained with the indicated antibodies. The sections are from the same experiment shown in Fig. 3J. (**H**) Average ± SEM of the number of GFAP$^+$ cells per field in brain slices of healthy animals (Ctrl) and mice receiving untreated or 1 μM PAPTP-treated lymphocytes ($n = 3$ for each group). On the right, representative immunohistochemical images of GFAP$^+$ in brain slices from mice of the indicated groups. The images were taken from the same region for each animal. The scale bar corresponds to 100 μm. *p*-value from one-way ANOVA. (**I**) Additional examples of Klüver-Barrera dual staining performed as in Fig. 4C. Red arrows indicate infiltrated/demyelinated zones. Please note also vacuolation, as e.g., in (Morales et al, 2006) in the enlarged image.

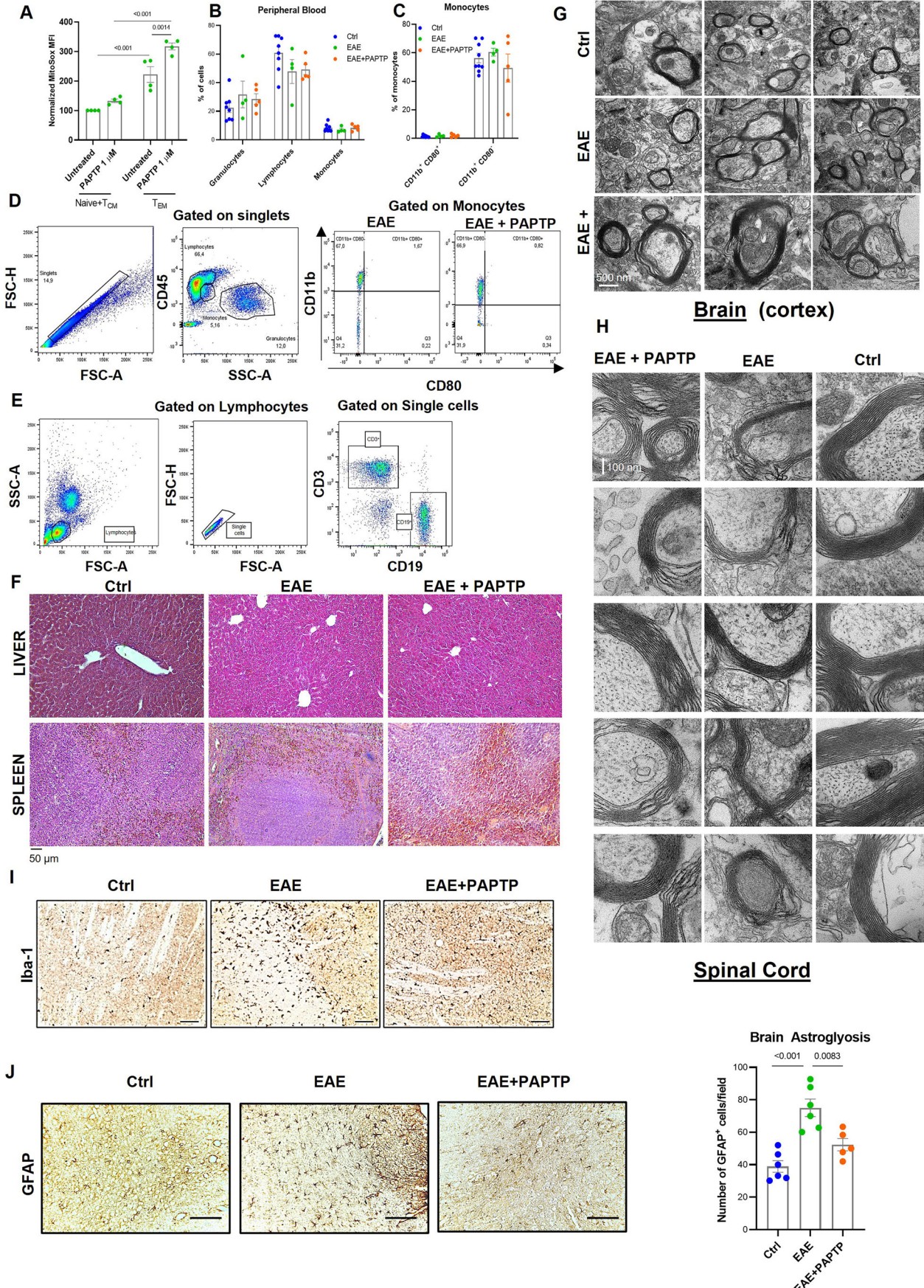

◀

**Figure EV3. Effects of PAPTP in EAE model.**

(A) Mean Fluorescence intensity of MitoSox in CD4$^+$ CD62L$^+$ (Naive+T$_{CM}$) and CD4$^+$ CD62L$^-$ (T$_{EM}$) lymphocytes in peripheral blood from EAE mice before (Untreated) and after 30 min treatment with 1 μM PAPTP. For each animal, data were normalized on the Mitosox Mean Fluorescence Intensity of CD4$^+$ CD62L$^+$ untreated cells. Data represent average ± SEM ($n = 4$). *p*-values of two-way ANOVA are indicated. (B) Percentages of lymphocytes, monocytes, and granulocytes in peripheral blood of mice of the indicated group at the endpoint of the experiment, evaluated using flow cytometry. Populations were gated on total single cells. Data represent average ± SEM ($n = 8$ for controls; $n = 4$ for EAE group and $n = 5$ for EAE + PAPTP group). (C) Percentages of CD11b$^+$ CD80$^+$ and CD11b$^+$ CD80$^-$ monocytes in peripheral blood of mice of the indicated group at the experimental endpoint. Cells were gated on C11b$^+$ monocytes. Data represent average ± SEM ($n = 9$ for controls; $n = 4$ for EAE group and $n = 5$ for EAE + PAPTP group). (D) Gating Strategy for identification of monocytes, lymphocytes and granulocytes (top). Representative dot plot showing CD11b$^+$ CD80$^+$ and CD11b$^+$ CD80$^-$ monocytes in peripheral blood of mice of the indicated group at the experimental endpoint (bottom). (E) Gating Strategy for identification of CD3$^+$ and CD19$^+$ lymphocytes in peripheral blood of mice. (F) Representative H&E images of livers and spleens of mice of the indicated groups. Scale bar indicated in the figure. Representative Transmission Electron Microscopy images showing neuronal (G) and spinal cords (H) myelination of mice from the indicated groups at the experimental endpoint. Scale bar indicated in the figure. (I) Representative immunohistochemical images of Iba-1$^+$ cells in brain slices from mice of the indicated group, sacrificed at 30 dpi. The scale bar corresponds to 100 μm. See enlarged images on Fig. 5K. (J) Average ± SEM of the number of GFAP$^+$ cells per field in brain slices of mice of the indicated groups ($n = 6$ for Ctrl and EAE, $n = 5$ for EAE + PAPTP groups). One outlier, defined with GraphPad was removed from the EAE + PAPTP group. At least 5 sections per animal were analyzed. On the right, representative immunohistochemical images of GFAP$^+$ in brain slices from mice of the indicated groups. The images were taken from the same region for each animal. The scale bar corresponds to 100 μm. (G–J) *p*-values of one-way ANOVA test are shown.

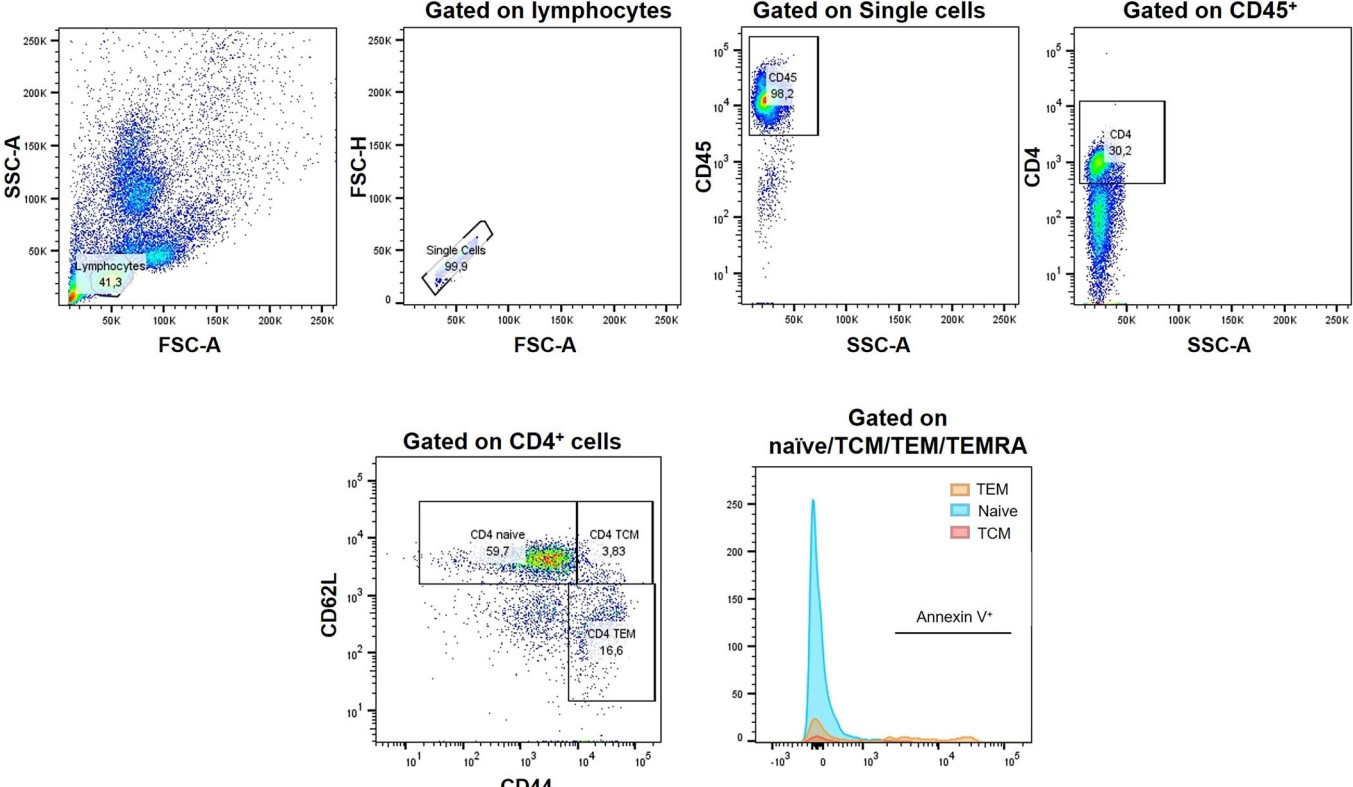

**Figure EV4. Gating strategy in EAE mice blood.**

Gating strategy used for the evaluation of apoptosis in naive, $T_{EM}$ and $T_{CM}$ lymphocytes in peripheral blood of mice.

