## [Peer Review File · EMBO Molecular Medicine]

Selective inhibition of the mitochondrial Kv1.3 prevents and alleviates multiple sclerosis in vivo

Beatrice Angi, Tatiana Varanita, Marco Puthenparampil, Valentina Scattolini, Michael Donadon, Mitra Tavakoli, Marta Favero, Maguie El Boustani, Matthias Soddemann, Lucia Biasutto, Diletta Arcidiacono, Alberto Ongaro, Andrea Mattarei, Livio Trentin, Gregory Wilson, Erich Gulbins, Paolo Gallo, and Ildiko Szabo

Corresponding authors: Ildiko Szabo (ildi@civ.bio.unipd.it) , Erich Gulbins (erich.gulbins@uni-due.de)

Review Timeline:

Submission Date:	30th Dec 24
Editorial Decision:	6th Feb 25
Revision Received:	11th Jun 25
Editorial Decision:	8th Jul 25
Revision Received:	6th Aug 25
Accepted:	11th Aug 25

Editor: Jingyi Hou

Transaction Report:

6th Feb 2025

Dear Prof. Szabo,

Thank you again for submitting your work to EMBO Molecular Medicine. We have now received the reports from the three reviewers and as you will see below, the reviewers think that the study is potentially interesting. They raise however a series of concerns, which we would ask you to convincingly address in a revision.

The reviewers' recommendations are relatively clear, so there is no need for me to reiterate the points listed below. All the raised issues need to be carefully addressed. In particular, Reviewers #2 and #3 had overlapping concerns regarding the need for a spinal cord pathology analysis and improved imaging data. Additionally, Reviewer #3 emphasized the need for convincing evidence that PAPTP specifically targets mitoKv1.3 in TEMs.

We would welcome the submission of a revised version within three months for further consideration. As you may already know, our editorial policy allows in principle a single round of major revision, and it is therefore essential to provide responses to the referees' comments that are as complete as possible.

Please also contact us as soon as possible if similar work is published elsewhere. If other work is published, we may not be able to extend the revision period beyond three months.

I look forward to seeing a revised form of your manuscript as soon as possible.

Kind regards,
Jingyi

Jingyi Hou
Senior Editor
EMBO Molecular Medicine

We require:

- 1) A .docx formatted version of the manuscript text (including legends for main figures, EV figures and tables). Please make sure that the changes are highlighted to be clearly visible.
- 2) Individual production quality figure files as .eps, .tif, .jpg (one file per figure). For guidance, download the 'Figure Guide PDF': (<https://www.embopress.org/page/journal/17574684/authorguide#figureformat>).
- 3) A .docx formatted letter INCLUDING the reviewers' reports and your detailed point-by-point responses to their comments. As part of the EMBO Press transparent editorial process, the point-by-point response is part of the Review Process File (RPF), which will be published alongside your paper.
- 4) A complete author checklist, which you can download from our author guidelines (<https://www.embopress.org/page/journal/17574684/authorguide#submissionofrevisions>). Please insert information in the checklist that is also reflected in the manuscript. The completed author checklist will also be part of the RPF.
- 5) Please note that all corresponding authors are required to supply an ORCID ID for their name upon submission of a revised

manuscript.

6) It is mandatory to include a 'Data Availability' section after the Materials and Methods. Before submitting your revision, primary datasets produced in this study need to be deposited in an appropriate public database, and the accession numbers and database listed under 'Data Availability'. Please remember to provide a reviewer password if the datasets are not yet public (see <https://www.embopress.org/page/journal/17574684/authorguide#dataavailability>).

12) Author contributions: You will be asked to provide CRediT (Contributor Role Taxonomy) terms in the submission system. These replace a narrative author contribution section in the manuscript.

13) A Conflict of Interest statement should be provided in the main text.

14) Every published paper now includes a 'Synopsis' to further enhance discoverability. Synopses are displayed on the journal

webpage and are freely accessible to all readers. They include a short stand first (maximum of 300 characters, including space) as well as 2-5 one-sentences bullet points that summarizes the paper. Please write the bullet points to summarize the key NEW findings. They should be designed to be complementary to the abstract - i.e. not repeat the same text. We encourage inclusion of key acronyms and quantitative information (maximum of 30 words / bullet point). Please use the passive voice. Please attach these in a separate file or send them by email, we will incorporate them accordingly.

Please also suggest a visual abstract to illustrate your article as a PNG file 550 px wide x 300-600 px high.

15) All Materials and Methods need to be described in the main text using our 'Structured Methods' format. According to this format, the Methods section includes a Reagents and Tools Table (listing key reagents, experimental models, software and relevant equipment and including their sources and relevant identifiers) followed by a Methods and Protocols section describing the methods, ideally using a step-by-step protocol format. The aim is to facilitate adoption of the methodologies across labs.

Please download and fill our Reagents and Tools Table template (.docx), which you can find in our author guidelines: <https://www.embopress.org/page/journal/17574684/authorguide#structuredmethods>

**** Reviewer's comments ****

Referee #1 (Comments on Novelty/Model System for Author):

The manuscript shows very clearly the benefit of using a mitoKv1.3 inhibitor, PAPTP, in the management of EAE in mice and clearly shows that the results could be translated to humans. The authors showed that PAPTP targets the Tem cells in the periphery, and reduces the number of harmful cells that could invade the CNS. There is data on humans (in vitro) and very clear in vivo results for EAE in mice. For humans, it is shown clearly that the PAPTP induces apoptosis Tem and T(EMRA) cells without harming T(CM) and naïve cells. For mice, the prevention and the treatment of EAE is clearly shown. The in vivo data are supported by extensive neuropathological data (immunofluorescence, electron microscopy). Finally, it is convincingly shown that the target is in the peripheral pool of the TEM cells.

I am convinced that PAPTP has a great therapeutic potential

Referee #1 (Remarks for Author):

Szabo and colleagues have performed an impressive set of experiments to demonstrate the suitability of PAPTP, their membrane permeant mitochondrial Kv1.3 specific inhibitor, in the prevention and treatment of multiple sclerosis. The study is very extensive, the authors have applied multiple techniques to convince the reader that i) PAPTP induces apoptosis in TEMs from MS patients, ii) that adoptive transfer of EAE can be prevented and iii) treated in mice. Their key observation is that PAPTP selectively induces apoptosis in both human and murine effector memory T cells (TEM) without compromising TCM and naïve T cell populations. In addition to classical immunological methods (T cell subset separation, proliferation and apoptosis assay, etc) they used immunofluorescence of neural tissues, electron microscopy, image analysis of microglia, functional mouse tests (motor performance, disease progression scoring, etc). As mentioned above, the methodological palette is very impressive. The conclusion is well supported by experimental data, key findings are clearly summarized in Fig 5 as a model for the action of PAPTP: the induction of TEM apoptosis in the periphery thereby reducing the infiltration of the brain. PAPTP is superior to other drugs targeting Kv1.3 since PAPTP is causing apoptosis whereas peptide toxins targeting Kv1.3 simply reduce proliferation.

There are a few minor comments to the paper:

1. Line 99: Probably therapy naïve than naïve patients, the ones who receive NAT treatment have higher lymphocyte autoprolieration than those that did not receive this treatment.
2. TEMRA cells are termed as effector memory re-expressing CD45RO T cells in the text (line 105) but are shown in Fig. 1 G as CD4+, CD25+ CCR7- CD45RO- TEMRAs.
3. Fig. 1 C shows the autoprolieration and apoptosis assay, rather than the gating strategy. Please fix the reference. Also, the structure of PAPTP on this panel seems to be oddly displayed here. The molecule was already introduced in Fig. 1A
4. Fig. 1D: % of TEM on CD4+ cells axis label: I think that : % of CD4+ TEM cells would be a better axis label- The same Fig. 1 F,

Also normalized to untreated would be a better label for Fig. 1G, H, J, L

5. Lines 147..149 are the repetitions of Lines 126..127.

6. Please add some explanation in the legend to Fig. 3C about how to read the WBC count, RBC count and platelet count. Do I understand correctly that WBC count is $\sim 8^*$ (10 to the power of 3)/microliter?

7. Line 221: "we also confirmed in this study where in the brain of EAE mice we did not find PAPTP at detectable levels (i.e., its concentration, if any, was < 0.1 nmoles/g tissue) at the end point of the experiment." Please add: data not shown.

Referee #2 (Comments on Novelty/Model System for Author):

N/A

Referee #2 (Remarks for Author):

The manuscript by Angi et al. presents intriguing findings regarding the role of chronically activated autoreactive effector memory T (TEM) lymphocytes in multiple sclerosis (MS), focusing on their high expression of the mitochondrial Kv1.3 potassium channel. Utilizing the small molecule inhibitor PAPTP, which selectively targets this channel, the authors demonstrated effective reduction of these TEM cells in blood samples from relapsing-remitting MS (RRMS) patients, without affecting other T cell populations. In a mouse model, pre-treatment with PAPTP during adoptive transfer was shown to specifically eliminate TEMs, thereby preventing disease onset. Moreover, administering PAPTP to mice prior to disease manifestation appeared to inhibit the progression of experimental autoimmune encephalomyelitis (EAE). Collectively, these findings propose that targeting the mitochondrial Kv1.3 channel in autoreactive TEMs could represent a potential therapeutic strategy for managing RRMS.

To solidify the conclusions of this study, further evidence is necessary:

Enhancing the TEM data in figures 3G-H with luxol fast blue staining for demyelination could substantiate the findings.

Performing this analysis on spinal cord sections (lumbar, thoracic, and cervical regions) from both the general EAE and adoptive transfer EAE models would be beneficial. Furthermore, examining neuronal degeneration using Bielschowsky staining in both models may strengthen the claim that PAPTP intervention preserves neuronal integrity and prevents degeneration.

Quantifying astrogliosis in both adoptive transfer and general EAE mice should be incorporated. Including IBA-1 quantification in adoptive transfer EAE mice would provide additional insight into the inflammatory landscape.

The rationale for conducting cytokine profiling on homogenized brain samples requires clarification, considering that EAE primarily affects the spinal cord in mice. Repeating these analyses with spinal cord homogenates might uncover further changes in cytokine profiles.

Conducting a MitoSox assay on isolated TEMs from untreated and PAPTP-treated EAE mice, similar to figure 1A, would help ascertain whether the effects of PAPTP on TEMs are consistent across human and murine models.

Minor suggestions for refinement:

There appears to be a missing reference to supplementary figure 2 on lines 179-180. Additionally, the staining information for CD44, CD62L, and CD4 in PAPTP-treated mice seems absent and should be addressed.

On line 231, the text lacks a reference to figure 4C, which should be added for consistency.

These recommendations aim to enhance the manuscript's clarity and provide rigorous support for the proposed mechanism and therapeutic potential.

Referee #3 (Comments on Novelty/Model System for Author):

the primary manifestations of EAE occur in the lumbar spinal cord, the authors should evaluate inflammation and tissue damage (including demyelination, oligodendrocyte death, and axonal degeneration) in the lumbar spinal cord of EAE mice.

Referee #3 (Remarks for Author):

The authors aimed to demonstrate that selective inhibition of the mitoKv1.3 channel leads to TEM apoptosis and subsequently reduces EAE severity. While the topic is important, the conclusions drawn are premature. There is no clear evidence that PAPTP selectively inhibits the mitoKv1.3 channel in TEMs. Additionally, the clinical and histological data from the EAE experiments appear rough and unconvincing. There are several major concerns.

1. The authors must provide convincing evidence that PAPTP induces TEM apoptosis specifically by inhibiting the mitoKv1.3 channel. To demonstrate selectivity, the authors should: 1) conduct experiments confirming that PAPTP inhibits mitoKv1.3 in TEMs; 2) generate mitoKv1.3 knock-out TEMs; 3) show that PAPTP induces apoptosis in wild-type TEMs but not in mitoKv1.3 knock-out TEMs.

2. Figure 2G, the authors should present daily EAE clinical scores to provide a more detailed assessment of disease progression.

3. Since the primary manifestations of EAE occur in the lumbar spinal cord, the authors should evaluate inflammation and tissue damage (including demyelination, oligodendrocyte death, and axonal degeneration) in the lumbar spinal cord of adaptive EAE mice.

4. Similarly, for active EAE, the authors should assess inflammation and tissue damage (including demyelination, oligodendrocyte death, and axonal degeneration) in the lumbar spinal cord of affected mice.

5. The presented EM data are rough and unconvincing. They fail to clearly demonstrate demyelination and axonal degeneration in the brains of both untreated and treated EAE mice. More detailed and higher-quality images are needed.
6. The interpretation of figure 3K-J is unclear. During acute EAE, Iba1+ cells in the CNS comprise a mixed population of microglia and macrophages.
7. Oligodendrocytes and neurons also express mitoKv1.3 channel. Does PAPTP affect the viability of oligodendrocytes and neurons in the EAE models?

***** Reviewer's comments *****

Referee #1 (Comments on Novelty/Model System for Author):

The manuscript shows very clearly the benefit of using a mitoKv1.3 inhibitor, PAPTP, in the management of EAE in mice and clearly shows that the results could be translated to humans. The authors showed that PAPTP targets the Tem cells in the periphery, and reduces the number of harmful cells that could invade the CNS. There is data on humans (in vitro) and very clear in vivo results for EAE in mice. For humans, it is shown clearly that the PAPTP induces apoptosis Tem and T(EMRA) cells without harming T(CM) and naïve cells. For mice, the prevention and the treatment of EAE is clearly shown. The in vivo data are supported by extensive neuropathological data (immunofluorescence, electron microscopy). Finally, it is convincingly shown that the target is in the peripheral pool of the TEM cells. I am convinced that PAPTP has a great therapeutic potential

We are very grateful to the Reviewer for the positive evaluation of our work and for the useful suggestions.

Referee #1 (Remarks for Author):

Szabo and colleagues have performed an impressive set of experiments to demonstrate the suitability of PAPTP, their membrane permeant mitochondrial Kv1.3 specific inhibitor, in the prevention and treatment of multiple sclerosis. The study is very extensive, the authors have applied multiple techniques to convince the reader that i) PAPTP induces apoptosis in TEMs from MS patients, ii) that adoptive transfer of EAE can be prevented and iii) treated in mice. Their key observation is that PAPTP selectively induces apoptosis in both human and murine effector memory T cells (TEM) without compromising TCM and naïve T cell populations. In addition to classical immunological methods (T cell subset separation, proliferation and apoptosis assay, etc) they used immunofluorescence of neural tissues, electron microscopy, image analysis of microglia, functional mouse tests (motor performance, disease progression scoring, etc). As mentioned above, the methodological palette is very impressive. The conclusion is well supported by experimental data, key findings are clearly summarized in Fig 5 as a model for the action of PAPTP: the induction of TEM apoptosis in the periphery thereby reducing the infiltration of the brain. PAPTP is superior to other drugs targeting Kv1.3 since PAPTP is causing apoptosis whereas peptide toxins targeting Kv1.3 simply reduce proliferation.

There are a few minor comments to the paper:

1. Line 99: Probably therapy naïve than naïve patients, the ones who receive NAT treatment have higher lymphocyte autoproliiferation than those that did not receive this treatment.

We corrected this sentence, as suggested.

2. TEMRA cells are termed as effector memory re-expressing CD45RA T cells in the text (line 105) but are shown in Fig. 1 G as CD4+, CD25+ CCR7- CD45RO- TEMRAs.

Indeed, CD45RA and CD45RO are the longer and shorter isoforms of CD45, respectively. CD45RA is expressed predominantly on naïve T cells, while CD45RO is characteristic of memory and activated T cells. CD45RA and CD45RO are used therefore to distinguish between naïve and memory T cells, respectively, while the terminally differentiated effector cells, named TEMRAs, re-express CD45RA

(TEMRA are defined by this characteristic) and lack both CD45RO and CCR7 (Carrasco J, Godelaine D, Van Pel A, Boon T, Van Der Bruggen P. CD45RA on human CD8 T cells is sensitive to the time elapsed since the last antigenic stimulation. *Blood*. 2006;108:2897–2905). Therefore, in our study we used CD45RO and CCR7 as markers to distinguish the different T cell subpopulations. This concept is now explained better in the main text: “Upon antigenic restimulation T_{CM} lose the chemokine receptor CCR7 expression and differentiate into T_{EM} and finally into T_{EMRA}, which are considered to be terminally differentiated. T_{EMRA} lack both CCR7 and CD45RO, a short isoform of CD45 and re-express the longer isoform, CD45RA (35). “

3. Fig. 1 C shows the auto proliferation and apoptosis assay, rather than the gating strategy. Please fix the reference. Also, the structure of PAPTP on this panel seems to be oddly displayed here. The molecule was already introduced in Fig.1A

Thanks for this comment. We changed Figure 1 and show the gating strategy in the Extended View figure 1C. We removed the structure of PAPTP, as it is now shown in the materials and methods section, given that PAPTP was the starting material for the synthesis of its fluorescent version, now included in the revised version to show the accumulation of the drug in mitochondria.

4. Fig. 1D: %of TEM on CD4+ cells axis label: I think that : % of CD4+ TEM cells would be a better axis label- The same Fig. 1 F, Also normalized to untreated would be a better label for Fig. 1G, H, J, L

We have complied.

5. Lines 147..149 are the repetitions of Lines 126..127.

We apologize for this mistake, now it has been fixed.

6. Please add some explanation in the legend to Fig. 3C about how to read the WBC count, RBC count and platelet count. Do I understand correctly that WBC count is $\sim 8 \times 10^3$ (10 to the power of 3)/microliter?

We added the explanation. Indeed, the WBC count is 8×10^3 /microliter, a value that is similar to that reported in other studies when taking the blood from the facial submandibular vein (e.g. Frohlich et al, Comparison of Serial Blood Collection by Facial Vein and Retrobulbar Methods in C57BL/6 Mice , *J Am Assoc Lab Anim Sci*. 2018 Jul 1;57(4):382-391).

7. Line 221: "we also confirmed in this study where in the brain of EAE mice we did not find PAPTP at detectable levels (i.e., its concentration, if any, was < 0.1 nmoles/g tissue) at the end point of the experiment." Please add: data not shown.

We have complied.

Referee #2 (Comments on Novelty/Model System for Author):

N/A

Referee #2 (Remarks for Author):

The manuscript by Angi et al. presents intriguing findings regarding the role of chronically activated autoreactive effector memory T (TEM) lymphocytes in multiple sclerosis (MS), focusing on their high expression of the mitochondrial Kv1.3 potassium channel. Utilizing the small molecule inhibitor PAPTP, which selectively targets this channel, the authors demonstrated effective reduction of these TEM cells in blood samples from relapsing-remitting MS (RRMS) patients, without affecting other T cell populations. In a mouse model, pre-treatment with PAPTP during adoptive transfer was shown to specifically eliminate TEMs, thereby preventing disease onset. Moreover, administering PAPTP to mice prior to disease manifestation appeared to inhibit the progression of experimental autoimmune encephalomyelitis (EAE). Collectively, these findings propose that targeting the mitochondrial Kv1.3 channel in autoreactive TEMs could represent a potential therapeutic strategy for managing RRMS. To solidify the conclusions of this study, further evidence is necessary: Enhancing the TEM data in figures 3G-H with luxol fast blue staining for demyelination could substantiate the findings. Performing this analysis on spinal cord sections (lumbar, thoracic, and cervical regions) from both the general EAE and adoptive transfer EAE models would be beneficial.

First of all, we would like to thank the Reviewer for the positive evaluation of our work and for the useful suggestions. As requested, in the revised version we included new data showing that demyelination, detected using luxol fast blue staining, is largely prevented in the animals treated with PAPTP. To obtain the spinal cord section from the general EAE model at the disease peak, we performed a new series of experiments, further confirming that PAPTP very quickly and significantly reduces clinical score and strongly reduces demyelination well detectable in the EAE mice in the lumbar and thoracic regions. The new data showing luxol fast blue staining are reported in figures 4C for general EAE and Extended View figure 2D for the adoptive transfer EAE. Quantifications were performed as previously described in Miyauchi et al, Gut microorganisms act together to exacerbate inflammation in spinal cords, 2020, Nature, 585(7823):102-106 for the spinal cord cross-sections and in Theotokis et al for the longitudinal spinal cord sections (Brain Pathol. 2022 Jul;32(4):e13040).

Furthermore, examining neuronal degeneration using Bielschowsky staining in both models may strengthen the claim that PAPTP intervention preserves neuronal integrity and prevents degeneration. Quantifying astrogliosis in both adoptive transfer and general EAE mice should be incorporated. Including IBA-1 quantification in adoptive transfer EAE mice would provide additional insight into the inflammatory landscape.

We have complied and performed the Bielschowsky staining and quantification of neuronal integrity following EAE and PAPTP treatment (figures 4D and Expanded View figure 2E of the revised version). Furthermore, immunohistochemistry using anti-GFAP antibodies revealed the astrogliosis-reducing effect of PAPTP in EAE spinal cord (revised Figure 4E) and brain (EV2G). Quantification in each sample (for all luxol fast blue, Bielschowsky staining and GFAP detection) was performed blinded to the investigator by two independent persons. Consistent with the findings shown in these new figures is the result of the rotarod test (Figure 5F in the revised version) which shows a complete recovery of motor function in almost all animals.

The rationale for conducting cytokine profiling on homogenized brain samples requires clarification, considering that EAE primarily affects the spinal cord in mice. Repeating these analyses with spinal cord homogenates might uncover further changes in cytokine profiles.

The main reason for performing this assay was to prove that EAE occurred also in the PAPTP-treated animals, through determination of the eosinophil chemoattractant Eotaxin levels, since Eotaxin-1 is produced by various cell types, including lymphocytes, fibroblasts and endothelial cells and is transported from the blood to the brain through the Blood-Brain Barrier. However, the Reviewer is correct, that a detailed cytokine profiling in the spinal cord at different time points following disease induction might give a more clear insight into the cytokine profiling-disease progression relationship. This set of experiments will be performed in a follow up study, therefore we deleted the cytokine profiling from the current manuscript.

Conducting a MitoSox assay on isolated TEMs from untreated and PAPTP-treated EAE mice, similar to figure 1A, would help ascertain whether the effects of PAPTP on TEMs are consistent across human and murine models.

We have complied and measured the mitochondrial ROS level in TEMs isolated from the animals of the experiment shown in Figure 4B. These experiments, now reported in EV 3A confirm that indeed, the effect and the mechanism of action of PAPTP are consistent between human MS patients' cells and the cells isolated from the murine EAE model: TEM showed a higher basal ROS level with respect to TCM and naïve T cells and only the TEM's ROS was higher after treatment with PAPTP.

Minor suggestions for refinement:

There appears to be a missing reference to supplementary figure 2 on lines 179-180. Additionally, the staining information for CD44, CD62L, and CD4 in PAPTP-treated mice seems absent and should be addressed.

Thanks for this suggestion, we have complied and cited the EV figure. In accordance with the lack of CD45+ cell infiltration, CD4⁺CD44⁺CD62L⁻ cells were absent in PAPTP-treated mice (data not shown).

On line 231, the text lacks a reference to figure 4C, which should be added for consistency. These recommendations aim to enhance the manuscript's clarity and provide rigorous support for the proposed mechanism and therapeutic potential.

Many thanks for these suggestions.

Referee #3 (Comments on Novelty/Model System for Author):

the primary manifestations of EAE occur in the lumbar spinal cord, the authors should evaluate inflammation and tissue damage (including demyelination, oligodendrocyte death, and axonal degeneration) in the lumbar spinal cord of EAE mice.

Please see detailed response below.

Referee #3 (Remarks for Author):

The authors aimed to demonstrate that selective inhibition of the mitoKv1.3 channel leads to TEM apoptosis and subsequently reduces EAE severity. While the topic is important, the conclusions drawn are premature. There is no clear evidence that PAPTP selectively inhibits the mitoKv1.3 channel in TEMs. Additionally, the clinical and histological data from the EAE experiments appear rough and unconvincing. There are several major concerns.

First of all, we would like to thank the Reviewer for the very useful suggestions. We addressed all points, allowing to substantially improve our study.

1. The authors must provide convincing evidence that PAPTP induces TEM apoptosis specifically by inhibiting the mitoKv1.3 channel. To demonstrate selectivity, the authors should: 1) conduct experiments confirming that PAPTP inhibits mitoKv1.3 in TEMs; 2) generate mitoKv1.3 knock-out TEMs; 3) show that PAPTP induces apoptosis in wild-type TEMs but not in mitoKv1.3 knock-out TEMs.

To address this point in both human and murine cells, we performed a series of new experiments and report in the revised version 3 independent lines of evidence regarding the action of PAPTP on mitochondrial Kv1.3. First, we synthesized a fluorescent version of PAPTP and followed its accumulation and localization in living T cells. As shown in Figure 3A, PAPTP-fluo perfectly co-localized with Mitotracker, a lipophilic cationic dye that is taken up by the mitochondria. Please note that the observed donut shape is due to the exclusion of Mitotracker from the round nucleus and the signal observed for mitochondria was similar to those reported in other studies (e.g. Baixauli F, et al, 2011, The mitochondrial fission factor dynamin-related protein 1 modulates T-cell receptor signalling at the immune synapse. *Embo j* 30: 1238-1250 and Buck MD, et al, 2016, Mitochondrial Dynamics Controls T Cell Fate through Metabolic Programming. *Cell* 166: 63-76. This experiment thus confirms that PAPTP specifically accumulates in mitochondria. Second, we obtained genetic evidence that expression of mitochondrial Kv1.3 is sufficient to trigger PAPTP-induced apoptosis. It is not possible to silence exclusively mitochondrial Kv1.3, as the same gene codifies the plasmamembrane and the mitochondria-located isoforms. Since the transfection of primary lymphocytes is most efficient through electroporation, we silenced Kv1.3 using a CRISPR-Cas9 construct through electroporation of cells obtained from the spleen of MOG₃₃₋₅₅-immunized mice (to have a suitable number of cells) and separated the Kv1.3+ cells from those silenced for Kv1.3. Next, we transfected the Kv1.3-negative cells with mitochondria-targeted EYFP-Kv1.3 construct (previously used in Szabo et al, PNAS, 2008) and left the cells either untreated or treated them with 1 μ M PAPTP (Figure 3B). These experiments clearly show that in contrast to mitoKv1.3-negative cells, where PAPTP induced death only in 2% of the cells, in mitoKv1.3-positive cells cell death exceeded 75% as assessed by Annexin V staining and flow cytometry (Figure 3C and extended view figure 2C). Finally, we assessed the effect of the membrane-impermeant potent Kv1.3 inhibitor Shk (Stichodactyla toxin) and of PAPTP on changes in mitochondrial membrane potential (TMRM), mitochondrial ROS production (MitoSox) and apoptosis (Annexin binding) in autoproliiferative cells from humans (Figure 1 B-F and Extended view 1B, 1C). We have previously shown in several cellular models that PAPTP causes a rapid hyperpolarization of the mitochondrial membrane potential (e.g. Leanza et al, *Cancer Cell*, 2017), as it blocks the influx of depolarizing positively charged K⁺ ions into the matrix. Thus, the observed rapid hyperpolarization within 1 minutes after addition of PAPTP clearly shows that PAPTP is acting on mitochondrial Kv1.3, as such a rapid effect on mitochondrial membrane potential is fully compatible with the action of the drug on an ion channel whose modulation is known to affect rapidly the membrane potential in general (of the PM and of mitochondria). Furthermore, Shk that blocks the plasmamembrane-located channel did not induce hyperpolarization, indicating again that PAPTP does not induce hyperpolarization by acting on the plasmamembrane Kv1.3. In accordance with the known effect of hyperpolarization on ROS release, we observed ROS production only after treatment with PAPTP but not Shk. Finally, Shk was not able to induce apoptosis of these autoproliiferative T cells from MS patients.

2. Figure 2G, the authors should present daily EAE clinical scores to provide a more detailed assessment of disease progression.

Former Figure 2G did not describe clinical scores. The former Figure 2E already showed the daily clinical scores for AT-EAE. In the former Figure 3B which reports EAE, the clinical scores were assessed every second day since PAPTP was administered every second day. For the revision we repeated the EAE experiment in order to perform histology at the peak of the disease and in this experiment we assessed the clinical score every day (Figure 4A and B of revised version), confirming again that administration of PAPTP upon disease onset drastically reduced EAE clinical score already within a few days.

3. Since the primary manifestations of EAE occur in the lumbar spinal cord, the authors should evaluate inflammation and tissue damage (including demyelination, oligodendrocyte death, and axonal degeneration) in the lumbar spinal cord of adoptive EAE mice.

4. Similarly, for active EAE, the authors should assess inflammation and tissue damage (including demyelination, oligodendrocyte death, and axonal degeneration) in the lumbar spinal cord of affected mice.

Answer for 3 and 4: As requested, in the revised version we included new data showing that demyelination, detected using luxol fast blue staining, is largely prevented in the animals treated with PAPTP. To obtain the spinal cord section from the general EAE model at the disease peak, we performed a new series of experiments, further confirming that PAPTP very quickly and significantly reduces clinical score and strongly reduces demyelination that is well detectable in the EAE mice in the lumbar and thoracic regions. The new data showing luxol fast blue staining are reported in figures 4C for general EAE and Extended View figure 2D for the adoptive transfer EAE. Quantifications were performed as previously described in Miyauchi et al, Gut microorganisms act together to exacerbate inflammation in spinal cords, 2020, Nature, 585(7823):102-106 for the spinal cord cross-sections and in Theotokis et al for the longitudinal spinal cord sections (Brain Pathol. 2022 Jul;32(4):e13040). We also performed the Bielschowsky staining and quantification of neuronal integrity following EAE and upon PAPTP treatment (figures 4D and Extended View figure 2E of the revised version). Furthermore, immunohistochemistry using anti-GFAP antibodies revealed the astrogliosis-reducing effect of PAPTP in EAE spinal cord (revised Figure 4E) and brain (EV2G). Quantification in each sample (for all luxol fast blue, Bielschowsky staining and GFAP detection) was performed blinded to the investigator by two independent persons. Consistent with the findings shown in these new figures is the result of the rotarod test (Figure 5F in the revised version) which shows a complete recovery of motor function in almost all animals.

5. The presented EM data are rough and unconvincing. They fail to clearly demonstrate demyelination and axonal degeneration in the brains of both untreated and treated EAE mice. More detailed and higher-quality images are needed.

We substituted the images with those acquired at higher magnification. The images are very similar to those published by other groups (e.g. Dupree JL et al (2022) Lanthionine Ketimine Ethyl Ester Accelerates Remyelination in a Mouse Model of Multiple Sclerosis. ASN Neuro 14: 17590914221112352, Rodrigues-Amorim et al, Nature Comm. (2024) 15(1):6744) and highlight a decrease of the number of myelin layers in EAE animals and a recovery upon PAPTP treatment.

6. The interpretation of figure 3K-J is unclear. During acute EAE, Iba1+ cells in the CNS comprise a mixed population of microglia and macrophages.

We thank the Reviewer for pointing out this issue. Indeed, both activated microglia and brain-infiltrated macrophages are positive for Iba1 and these cells contribute to the inflammatory state that is so crucial in MS. We have modified the text accordingly.

7. Oligodendrocytes and neurons also express mitoKv1.3 channel. Does PAPTP affect the viability of oligodendrocytes and neurons in the EAE models?

As stated in the original version of the paper, PAPTP does not cross the BBB in the EAE model (and in general) as its concentration was below the detection limit using HPLC. The result is consistent also with our previous finding described in Parrasia et al, An Angiopep2-PAPTP Construct Overcomes the

Blood-Brain Barrier. New Perspectives against Brain Tumors Pharmaceuticals (Basel) 2021 Feb 6;14(2):129. doi: 10.3390/ph14020129 where we have also shown that PAPTP does not cross BBB.

8th Jul 2025

Dear Prof. Szabo,

Thank you for submitting the revised version of your manuscript. We have now received feedback from the two reviewers who re-evaluated your study. As you will see below, Reviewer #2 is generally satisfied with the revisions you have made, whereas Reviewer #3 has raised several remaining concerns.

During our pre-decision cross-commenting process, Reviewer #2 provided additional input on how to address Reviewer #3's concerns. In particular, regarding points #2, #3, and #4 from Reviewer #3, Reviewer #2 stated:

"I think that quantifying Bielchowsky and LFB staining will suffice, as this will help understanding quantitatively the impact on axons and myelin. Would not be myself too picky on the EM aspect, which is not easy to apply and standardise in EAE (could write forever in support of this). I think providing quantification of Bielchowsky and LFB staining is acceptable."

In light of this, we ask that you address Reviewer #3's points #2-4 in line with Reviewer #2's suggestion. In addition, please address the minor issues raised by Reviewer #2 and point #1 raised by Reviewer #3.

On a more editorial level:

1. Please remove the Authors' contribution section from the manuscript file.
2. Please provide up to five keywords.
3. Add the following missing funding information to the submission system: DFG (grant GU 335/35-2 to E.G), the Veronesi Foundation; MUR-PNRR Progetto finanziato dall'Unione Europea - NextGenerationEU - MUR - Mission 4 "Istruzione e Ricerca" del Piano Nazionale di Ripresa e Resilienza, Componente C2 - Misura 1.4.
4. Please provide a visual abstract to illustrate your article as a PNG file 550 px wide x 300-600 px high.
5. Please provide a 'Synopsis' to further enhance discoverability. Synopses are displayed on the journal webpage and are freely accessible to all readers. They include a short stand first (maximum of 300 characters, including space) as well as 2-5 one-sentences bullet points that summarizes the paper. Please write the bullet points to summarize the key NEW findings. They should be designed to be complementary to the abstract - i.e. not repeat the same text. We encourage inclusion of key acronyms and quantitative information (maximum of 30 words / bullet point). Please use the passive voice. Please attach these in a separate file, we will incorporate them accordingly.
6. Please remove the one sentence summary from the manuscript file.
7. "Data and materials availability" should be renamed to " Data availability"
8. "Competing interests" should be renamed to "Disclosure and Competing Interests Statement".
9. EV figures:
 - The figure titles and legends for the EV figures need to be revised for consistency. Currently, there is a mix of labels such as "Figure S2" and "EV3," "EV4." These should be updated to follow the correct format: "Figure EV2," "Figure EV3," and "Figure EV4".
 - Each EV figure must be uploaded as a separate file.
 - The term "Supplementary Information" should not be used.
10. Please add callouts for Figure 7 in the main text and include specific callouts for each panel of Figure 6.
11. Table A should be relabeled as Table 1 and positioned between the main and EV figure legends. Additionally, the legend should be formatted as plain text rather than placed in a text box.
12. The label "Scheme 1" should be removed, as this nomenclature is not used in our format. The corresponding synthesis figure could be labeled as an EV figure, with an appropriate callout in the main text and a legend provided.
13. in Methods, include a statement that informed consent was obtained from all subjects and that the experiments conformed to the principles set out in the WMA Declaration of Helsinki and the Department of Health and Human Services Belmont Report.
14. Please provide "The paper explained": EMBO Molecular Medicine articles are accompanied by a summary of the articles to

emphasize the major findings in the paper and their medical implications for the non-specialist reader. Please provide a draft summary of your article highlighting

15. During our routine image checks, we noticed that the microscopy panels across the figure set appear pixelated. This is a common result of converting original 16-bit TIFF images to RGB format for publication, and while not a cause for concern, it can sometimes give the impression of image alteration to critical readers. Please provide higher resolution microscopy images for - Figure 3A & Figure EV3B.

16. The reuse of cells between Figure 3A and Figure EV3A is not mentioned in the figure legends and should be clearly stated.

17. The manuscript sections should be in the following order: Title page - Abstract & Keywords - Introduction - Results - Discussion - Methods - Data Availability - Acknowledgments - Disclosure Statement & Competing Interests - References - Figure Legends - (Main Tables with legends if applicable) - Expanded View Figure Legends

18. Please address the following comments related to figure legends:

- Please note that the exact p values are not provided in the legends of figures 1A, B, D, E, F; 2A, B, D, E; 3C, E, F, G, H, I; 4B-E; 5B, C, D, E, F, G, I, J, L; 6A, B, C, G; EV1 D, E, G, H, J, L, M; EV2 D, G; EV3 A, J.
- Please note that information related to n is missing in the legend of figure EV1 M

I look forward to seeing a revised form of your manuscript as soon as possible.

Kind regards,
Jingyi Hou

Jingyi Hou
Senior Editor
EMBO Molecular Medicine

***** Reviewer's comments *****

Referee #2 (Remarks for Author):

I am satisfied with the authors' revisions and recommend the work for publication in EMBO Molecular Medicine. Please address the following minor comments before publication:

Textual Revisions:

Lines 124-125: Clarify if PBMCs are from both treatment-naïve RRMS patients and those undergoing NAT treatment, or only the latter.

Line 133: Define "TEMRA" upon its first use.

Line 141: Correct the sentence structure and remove the redundant "(Figure 1A)".

Lines 155-157: This rationale is redundant given the explanation in Lines 124-125 and can be removed.

Lines 212-216: Revise statements to reflect that Annexin V staining alone indicates increased apoptosis, not necessarily

"inducing death," without concomitant PI staining.

Lines 307-327: Update the figure reference from Figure 4 to Figure 6.

Lines 335-336: Rephrase "greater axonal remyelination" to "preserved myelin" or "decreased demyelination," as LFB staining alone cannot determine remyelination in this context.

Figure Revisions:

Figure 1E: Include the untreated condition in the overlaid graph.

Figure EV2E: Clarify why Bielchowsky staining was not quantified.

Referee #3 (Remarks for Author):

The authors add some new data to address the reviewer's concerns. However, many of the major concerns are not addressed.

1. The authors state: "Kv1.3 was silenced using CRISPR-Cas9 in cells obtained from the spleen of MOG33-55-immunized mice and the Kv1.3-expressing cells were separated from those silenced/negative for Kv1.3 (see Materials and methods for details). Next, the Kv1.3-negative cells were transfected with mitochondria-targeted EYFP-Kv1.3 construct (Szabo et al, 2008) and the cells were either left untreated or treated with 1 μ M PAPTP (Figure 3B)". However, the authors do not present any evidence showing that Kv1.3 is silenced using CRISPR-Cas9 in cells, and that mitoKv1.3 is expressed in the Kv1.3-negative cells transfected with mitochondria-targeted EYFP-Kv1.3 construct. The authors should perform western blot or immunostaining to verify Kv1.3 deletion and mitoKv1.3 expression in these cells.

2. Figure EV 2D and 2E Blieschowsky staining looks odd, which does not show any evidence of axon damage. In fact, axons look intact in these EAE mice.

3. Figure 4C, luxol fast blue staining shows that myelination is significantly enhanced in the EAE mice, as compared to control mice and EAE + PAPTP.

4. Figures 5G-I and EV3G, H: Electron microscopy (EM) is the gold standard for assessing demyelination and axonal degeneration in EAE mice. Demyelination refers to the disruption of compact myelin and the clearance of myelin debris, resulting in denuded (naked) axons. Although a recent study suggests that thinning of pre-existing mature myelin may represent a non-destructive form of myelin loss in EAE (Front Cell Neurosci. 2025 Mar 11;19:1565913. doi: 10.3389/fncel.2025.1565913), the authors should provide convincing evidence demonstrating that axon and myelin damage occurs. Specifically, they should quantify the increase in the number of degenerating axons, the number of demyelinated (naked) axons, the number of axons wrapped by damage myelin, and/or the number of axons ensheathed by abnormally thin myelin in EAE mice.

***** Reviewer's comments *****

Referee #2 (Remarks for Author):

I am satisfied with the authors' revisions and recommend the work for publication in EMBO Molecular Medicine. Please address the following minor comments before publication:

Textual Revisions:

Lines 124-125: Clarify if PBMCs are from both treatment-naïve RRMS patients and those undergoing NAT treatment, or only the latter.
Line 133: Define "TEMRA" upon its first use.
Line 141: Correct the sentence structure and remove the redundant "(Figure 1A)".
Lines 155-157: This rationale is redundant given the explanation in Lines 124-125 and can be removed.
Lines 212-216: Revise statements to reflect that Annexin V staining alone indicates increased apoptosis, not necessarily "inducing death," without concomitant PI staining.
Lines 307-327: Update the figure reference from Figure 4 to Figure 6.
Lines 335-336: Rephrase "greater axonal remyelination" to "preserved myelin" or "decreased demyelination," as LFB staining alone cannot determine remyelination in this context.

Many thanks for the useful suggestions, we have complied for all points.

Figure Revisions:

Figure 1E: Include the untreated condition in the overlaid graph.
Figure EV2E: Clarify why Bielchowsky staining was not quantified.

We included the requested overlaid graph in Figure 1E and performed quantification of the Bielschowsky staining, as requested also by Reviewer 3.

Referee #3 (Remarks for Author):

The authors add some new data to address the reviewer's concerns. However, many of the major concerns are not addressed.

1. The authors state: "Kv1.3 was silenced using CRISPR-Cas9 in cells obtained from the spleen of MOG33-55-immunized mice and the Kv1.3-expressing cells were separated from those

silenced/negative for Kv1.3 (see Materials and methods for details). Next, the Kv1.3-negative cells were transfected with mitochondria-targeted EYFP-Kv1.3 construct (Szabo et al, 2008) and the cells were either left untreated or treated with 1 μ M PAPTP (Figure 3B)". However, the authors do not present any evidence showing that Kv1.3 is silenced using CRISPR-Cas9 in cells, and that mitoKv1.3 is expressed in the Kv1.3-negative cells transfected with mitochondria-targeted EYFP-Kv1.3 construct. The authors should perform western blot or immunostaining to verify Kv1.3 deletion and mitoKv1.3 expression in these cells.

In the previous revised version we have provided evidence that mitochondria-targeted EYFP-Kv1.3 construct is expressed in the Kv1.3-downregulated cells in Figure EV2C, where we showed representative FACS data of EYFP-mitoKv1.3⁺ Annexin⁺ cells after PAPTP treatment. In this figure it is shown that EYFP-mitoKv1.3⁺ cells (on the x scale EFYP signal is detected due to the presence of a population that expresses EFYP in fusion with mitochondria-targeted Kv1.3) are the ones that prevalently undergo apoptosis. Following the Reviewer's request, now we included additional control data in Figure EV2C, showing 1) Downregulation of Kv1.3 in cells treated with CRISPR/Cas9 and Kv1.3 staining in sorted cells; 2) Flow cytometry of sorted Kv1.3-negative cells that were re-transfected with mitochondria-targeted EYFP-Mito-Kv1.3.

Please find here below the figures along with the explanations that were included into the new version of the manuscript, as parts of Figure EV2C.

Figure 1. Downregulation of Kv1.3 in cells treated with CRISPR/Cas9 and Kv1.3 staining in sorted cells;

Upper panel:

Yellow curve: CRISPR/Cas transfected, unstained

Blue curve: CRISPR/Cas transfected, cells positively sorted for Kv1.3 (see details in mat and methods section). Briefly: For sorting, cells were collected, washed, incubated with Fc receptor blocking reagent and labelled with biotin-conjugated anti-Kv1.3 antibody (Alomone Labs, #APC-101B). Following a second wash, cells were incubated with streptavidin-conjugated microbeads. Then, Kv1.3-positive and -negative populations were isolated using LS columns (Miltenyi Biotec, 130-042-401). For the flow cytometry, Fc-receptors were blocked, and then an aliquot of the cells was stained with a FITC-coupled

anti-rabbit IgG to detect the anti-Kv1.3, which was already bound to the cells. Please note the logarithmic scale of the X axis.

Red curve: CRISPR/Cas transfected, cells negatively sorted for Kv1.3. This is the fraction, which was negatively sorted and then stained exactly as the blue fraction. The staining was done on aliquots just before retransfection of the mito-Kv1.3 construct to confirm down-regulation.

Lower panel: Same as above, but with aliquots that were fixed, permeabilized and then stained. This confirms downregulation of extra- and intracellular Kv1.3.

Figure 2: Flow cytometry of sorted Kv1.3-negative cells that were re-transfected with EYFP-Mito-Kv1.3

Left upper panel: Blue: Permeabilized cells stained with FITC-coupled anti-Kv1.3 (Alamone, #APC-101-F). Most cells are negative for Kv1.3, a few are positive, which might be due to re-expression in some cells, since the CRISPR/CAS construct may not delete both genes in all cells. This is consistent with the data on cell death in the paper, which also shows some death in the fraction, which is not mito-Kv1.3 transfected.

Red: EYFP-Mito-Kv1.3 transfected cells, as above. Almost all cells express Mito-Kv1.3. The transfection efficacy is very high, because the electroporation is under very stringent conditions. Here the cells were stained with FITC-anti-Kv1.3, because otherwise we could not see Kv1.3 in untransfected cells.

Lower left panel: for comparison is shown Fig EV2C, where Kv1.3- cells were transfected with EYFP-mitoKv1.3 construct and subsequently treated with PAPTP for 48

hours. Cells were then analyzed for apoptosis using flow cytometry and Annexin V binding. Here, we detected EYFP-mitoKv1.3 (detecting EYFP) in order to detect the transfectants.

2. Figure EV 2D and 2E Blieschowsky staining looks odd, which does not show any evidence of axon damage. In fact, axons look intact in these EAE mice.

Figure EV 2E indeed shows the 2E Blieschowsky staining, while Fig EV2D shows Luxol fast blue/cresyl violet staining. In order to better highlight the differences between untreated and PAPTP-treated spinal cord, we show in the revised version images with higher magnification (in Figure EV2E) (and with improved sharpness of the images), where disruption of the axons is well visible and some examples are indicated with red arrows on the longitudinal sections.

For comparison, we report here the result of Blieschowsky staining on longitudinal sections of spinal cord from Theotokis et al, Brain Pathology 2022. Left: control EAE; Right: treated EAE

Quantification of our data was performed as suggested also by Reviewer 2, according to the procedure described in Theotokis et al 2022, Brain Pathology): Axonal loss was assessed on ten randomly selected silver-stained spinal cord sections within 120 x 240 μm areas along the spinal cord and graded by two independent evaluators in a blinded manner: 0 = normal/even silver stain throughout the white matter compared to unimmunized mice; 1 = small spurious areas in the white matter that lack silver stain; 2 = small, but frequent, areas in the white matter that lack silver stain; and 3 = extensive loss of silver stain throughout the white matter. In addition, we also performed quantification using Fiji algorithm as for the cross-sections of the spinal cord shown in Figure 4D. In this quantification we measured the areas (using Fiji) lacking silver staining with respect to the total examined area.

Please note that our specimens of the cross sections (copied below from figure 4D) with Blieschowsky staining are also compatible with those reported in the literature, e.g. Pyka-Fosciak et al, J. of Physiology and Pharmacology 2023, 74, 4, 465-472):

3. Figure 4C, luxol fast blue staining shows that myelination is significantly enhanced in the EAE mice, as compared to control mice and EAE + PAPTP.

We apologize for not having provided in the methods section all the necessary information in order to correctly interpret the result. The LFB staining kit we used was the Klüber Barrera Luxol fast blue staining

kit which contains LFB and then counterstaining is performed using cresyl violet. This kit detects myelin in turquoise blue, and nuclei in of neurons and glia as well as nuclei of infiltrating mononuclear cells in violet, finally Nissl substance in pink. This kit is recommended for staining of myelin and phospholipids.

The intensified staining the Reviewer refers to and observable in the EAE model spinal cord (see Figure 4c) corresponds to the infiltrating leukocytes, whose presence correlates with the myelin loss, as observed also in the following images by other groups and reported here for comparison.

The example below is from: Miyauchi et al, Nature, 585, page 102, 3 September 2020

Legend of the above figure: Mice that were orally treated with ampicillin were protected from demyelination of the spinal cord. B) Left, representative images of spinal-cord sections stained with luxol fast blue, from control (n = 7) and ampicillin-treated (n = 7) groups. Scale bar, 500 μ m. Right, the demyelinated area in the white matter was calculated. From Mat. and methods: The demyelinated regions in the white matter were quantified using ImageJ (v.1.8.0, NIH), and the percentages of demyelinated areas in the total white matter areas were calculated. Demyelination of spinal cords was assessed at day 20 after immunization.

We also contacted the authors of the Nature paper who confirmed that they measured the area of the dark blue zones with respect to the area of the white matter to quantify demyelination, as the cresyl-violet stained nuclei (included in the Klüver Barrera Luxol fast blue staining kit) in the white matter derive from the immune cell infiltration and indeed correspond to the areas of the demyelination, as also stated and shown for example in Ucciferri et al, *J. Vis. Exp.* (204), e65738, doi:10.3791/65738, 2024.

A further example of staining with the Klüver Barrera Luxol fast blue staining is reported in Morales LB et al, *J. of Neuroscience* 2006. Here the authors assessed demyelination at the peak of the disease (like us) and used the same LFB+cresyl violet staining. On the right images where EAE was induced, the authors detected massive leukocyte infiltration and demyelination indicated by pale areas and vacuolation. The same kind of outcome is shown in Figure 4C of our manuscript, where some of the pale areas are indicated with red arrows in the revised version.

We have now explained better this point in the methods section and refer to inflammatory/demyelinating lesions in the text and figure legend and cited the above-mentioned papers.

4. Figures 5G-I and EV3G, H: Electron microscopy (EM) is the gold standard for assessing demyelination and axonal degeneration in EAE mice. Demyelination refers to the disruption of compact myelin and the clearance of myelin debris, resulting in denuded (naked) axons. Although a recent study suggests that thinning of pre-existing mature myelin may represent a non-destructive form of myelin loss in EAE (Front Cell Neurosci. 2025 Mar 11;19:1565913. doi: 10.3389/fncel.2025.1565913), the authors should provide convincing evidence demonstrating that axon and myelin damage occurs. Specifically, they should quantify the increase in the number of degenerating axons, the number of demyelinated (naked) axons, the number of axons wrapped by damage myelin, and/or the number of axons ensheathed by abnormally thin myelin in EAE mice.

As pointed out by Reviewer 2, quantitative analysis of the TEM images is not straightforward and furthermore the images shown here are of support to the other methods we used which all gave consistent results (score, rotarod, microglia activation, astrogliosis, demyelination, axon degeneration in histology). Indeed, Reviewer 3 did not ask for the above-mentioned analysis during the first round of review. In any case, taking into account the kind feedbacks from both Reviewers, we performed a well-accepted evaluation of the axon demyelination as assessed by TEM. In particular, we measured and reported in the revised version in Figure 5I lower panel, the g-ratio. This method of quantification was performed as suggested by a STAR protocol paper (Wang et al, STAR protocols 3, 101304, June 17, 2022) (shown here below):

Legend from the cited paper:

(D) Representative pictures of myelin at higher magnification (36700) in the corticospinal tract. Scale bar, 1 mm.

(E) Myelinated axon density (mean number of myelinated axons per square millimeter) in WT and gene KO mice. Data are represented as mean G s.e.m. * $p < 0.05$.

(F) Diagram and calculation of g-ratio of myelinated axons.

(G) Diagram and calculation of g-ratio of elongated myelinated axons.

(H) Quantification of g-ratio. Data are represented as mean G s.e.m. * $p < 0.05$.

Our results show that, compatibly with the change of the number of myelin layer, the g ratio increases in EAE animals' spinal cords but is decreased to normal levels upon treatment with PAPTP.

We hope that the revision clarified the issues raised by the Reviewer and we thank for the useful suggestions that helped us to improve our manuscript.

11th Aug 2025

Dear Prof. Szabo,

Please find enclosed the final reports on your manuscript. We are pleased to inform you that your manuscript is accepted for publication and is now being sent to our publisher to be included in the next available issue of EMBO Molecular Medicine.

Yours sincerely,
Jingyi

Jingyi Hou
Senior Editor
EMBO Molecular Medicine

Referee #2 (Comments on Novelty/Model System for Author):

The authors have successfully addressed by final minor concerns, as well as provided a much clearer assessment of axons and myelin, as other reviewers had asked. I am happy for this paper to be published in EMBO Molecular Medicine.

Referee #2 (Remarks for Author):

The authors have successfully addressed by final minor concerns, as well as provided a much clearer assessment of axons and myelin, as other reviewers had asked. I am happy for this paper to be published in EMBO Molecular Medicine.
